# WHEN MACHINE LEARNING GETS PERSONAL: EVALUATING PREDICTION AND EXPLANATION

**Louisa Cornelis**[1,*]    **Guillermo Bernárdez**[1]    **Haewon Jeong**[1]    **Nina Miolane**[1]
[1]University of California, Santa Barbara, USA
[*]Correspondence to: `louisacornelis@ucsb.edu`

## ABSTRACT

In high-stakes domains like healthcare, users often expect that sharing personal information with machine learning systems will yield tangible benefits, such as more accurate diagnoses and clearer explanations of contributing factors. However, the validity of this assumption remains largely unexplored. We propose a unified framework to quantify how *personalizing a model* influences both prediction and explanation. We show that its impacts on prediction and explanation can diverge: a model may become more or less explainable even when prediction is unchanged. For practical settings, we study a standard hypothesis test for detecting personalization effects on demographic groups. We derive a finite-sample lower bound on its probability of error as a function of group sizes, number of personal attributes, and desired benefit from personalization. This provides actionable insights, such as which dataset characteristics are necessary to test an effect, or the maximum effect that can be tested given a dataset. We apply our framework to real-world tabular datasets using feature-attribution methods, uncovering scenarios where effects are fundamentally untestable due to the dataset statistics. Our results highlight the need for joint evaluation of prediction and explanation in personalized models and the importance of designing models and datasets with sufficient information for such evaluation.

## 1 INTRODUCTION

In critical domains like healthcare and education, machine learning models are increasingly personalized by incorporating input attributes that encode personal characteristics. These attributes can be sensitive and linked to historical bias, such as sex or race, or costly, for example requiring expert-administered medical assessments. When users provide personal attributes to a model, they implicitly expect improved predictions, but does personalization consistently meet that expectation?

Personalization can indeed enhance predictive accuracy. For instance, cardiovascular risk prediction models often perform better when including sex (Paulus et al., 2016; Huang et al., 2024; Mosca et al., 2011) and race (Paulus et al., 2018). This is because men, women, and different racial groups exhibit different heart disease patterns. For example, hypertension is more common in African American populations (Flack et al., 2003). Hence, personalization enhances clinical predictions by capturing meaningful biological and sociocultural variation.

However, personalization can also pose risks. Including sensitive attributes such as race, gender, or age can amplify biases in machine learning and perpetuate damaging inequality. For example, Obermeyer et al. (2019) showed that a health algorithm relying on health care costs, an attribute shaped by racial inequities, systematically underestimated illness in Black patients compared to equally sick white patients. This reduced their access to extra care by over half.

Generally, personalization may benefit overall accuracy while harming specific groups, making such risks harder to detect. In sleep apnea classification, adding age and sex improved overall performance but increased errors for older women and younger men (Suriyakumar et al., 2023). Similar group disparities have been observed in explainable machine learning, where some users receive less faithful or reliable explanations than others (Balagopalan et al., 2022; Dai et al., 2022). These gaps matter: when explanations are less faithful to the true model logic, they can give users an inaccurate picture of how the model makes decisions, leading to misplaced trust or missed warning signs. Dai et al.

(2022) illustrate this in a healthcare setting where explanations for men correctly reveal the model's reliance on a spurious feature, helping doctors override bad predictions, while explanations for women hide the spurious reasoning and instead highlight clinically relevant cues, causing doctors to trust incorrect predictions and resulting in higher misdiagnosis rates. However, these studies did not examine whether personalization itself contributes to explanation disparities, making it critical to assess whether model personalization may reduce explanation quality for some users. Hence, before personalizing a model, practitioners must consider if it delivers consistent gains across demographic groups in both prediction and explanation—see Figure 1.

This showcases the need for a quantitative framework to rigorously and fairly assess the benefits and risks of personalization. We focus on two key goals of machine learning models in high-stakes settings like healthcare: *(i)* making accurate predictions and *(ii)* providing explanations for them. Our central question is: *how reliably can we evaluate whether personalization improves prediction accuracy and explanation quality, both overall and across groups?*

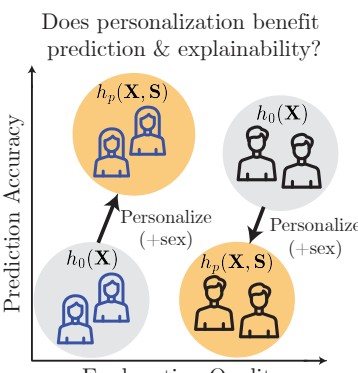

Figure 1: Impact of personalization on prediction and explanation: some groups benefit, others are harmed. $h_0$ is a generic model, $h_p$ is a personalized model that takes an additional group attribute, $\mathbf{S}$.

**Contributions.** We propose a comprehensive study of the impact of personalization for prediction accuracy and explanation quality in machine learning models. Specifically:

1. We show that even when personalization does not improve prediction, it can enhance or degrade explainability in terms of how sufficient and comprehensive an explanation is. This highlights the need to evaluate both independently to ensure fairness in settings where accuracy and interpretability are critical (Section 4).

2. We derive distribution-aware limits on when personalization cannot be reliably tested, showing how many attributes or samples are needed in finite datasets. Our theory extends prior work beyond binary classification to general supervised learning, revealing key differences between evaluating prediction and explanation in classification versus regression (Section 5).

3. We apply our proposed framework to real-world tabular datasets on classification and regression tasks, finding empirical evidence that personalization can affect explanation and prediction differently (Table 2). We illustrate how group-level gains from personalization are fundamentally untestable, thereby precluding statistical justification across different scenarios (Section 6).

Overall, we offer a cautionary perspective on the promise of personalized medicine and the personalization of machine learning in other critical domains. Even when personalizing a machine learning model could be beneficial, it might be impossible to reliably prove it—thus limiting its practical use.

## 2 RELATED WORK

Studies that investigate how personalizing machine learning models influences group outcomes (Suriyakumar et al., 2023) are limited to a narrow subset of performance measures and do not address explanation quality as described next. Extended related works are in Appendix A.

**Theory.** Few works theoretically characterize the impact of personalization. Monteiro Paes et al. (2022) define the Benefit of Personalization (BoP) as the smallest gain in performance that any group can expect from personalization. While the *definition* of BoP applies to any supervised learning task and "performance" measure, the theory supporting its use is confined to binary performance measures, such as accuracy in binary classification (0/1 loss) or false negative and positive rates (Bernoulli variables). Hence, it does not extend to continuous metrics like regression accuracy or explanation quality for regression and fails to provide a complete framework. Moreover, the theorems make unrealistic assumptions about dataset statistics (e.g., demographic groups of equal size) that further restrict their applicability in real-world settings. The general impact of personalization therefore remains theoretically uncharacterized.

Table 1: **Costs of model $h$ for group $s$ used to evaluate the impact of personalization** on data $(\tilde{\mathbf{X}}, \mathbf{Y})$ where $\tilde{\mathbf{X}} = \mathbf{X}$ for a generic model $h_0$, $\tilde{\mathbf{X}} = (\mathbf{X}, \mathbf{S})$ for a personalized model $h_p$, while $\tilde{\mathbf{X}}_{\setminus J}$ denotes the input when removing the most important features and $\tilde{\mathbf{X}}_J$ is its complement (see Section 4). Personalization benefits group $s \in \mathcal{S}$ if $C(h_0, s) - C(h_p, s) > 0$ and harms if $C(h_0, s) - C(h_p, s) < 0$. Incomprehensiveness is abbreviated as Incomp.

| | $C(h, s)$ | **Classification** | **Regression** |
|---|---|---|---|
| **Predict** | **Loss** | $\Pr(h(\tilde{\mathbf{X}}) \neq \mathbf{Y} \mid \mathbf{S} = s)$ | $\mathbb{E}\left[\|h(\tilde{\mathbf{X}}) - \mathbf{Y}\|^2 \mid \mathbf{S} = s\right]$ |
| | **Evaluation metric** | $-\mathrm{AUC}(h(\tilde{\mathbf{X}}), \mathbf{Y} \mid \mathbf{S} = s)$ | $-R^2(h(\tilde{\mathbf{X}}), \mathbf{Y} \mid \mathbf{S} = s)$ |
| **Explain** | **Sufficiency** | $\Pr(h(\tilde{\mathbf{X}}) \neq h(\tilde{\mathbf{X}}_J) \mid \mathbf{S} = s)$ | $\mathbb{E}\left[\|h(\tilde{\mathbf{X}}) - h(\tilde{\mathbf{X}}_J)\|^2 \mid \mathbf{S} = s\right]$ |
| | **Incomp.** | $-\Pr\left(h(\tilde{\mathbf{X}}) \neq h(\tilde{\mathbf{X}}_{\setminus J}) \mid \mathbf{S} = s\right)$ | $-\mathbb{E}\left[\|h(\tilde{\mathbf{X}}) - h(\tilde{\mathbf{X}}_{\setminus J})\|^2 \mid \mathbf{S} = s\right]$ |

**Empirical Evidence.** While the impact of personalization on explanation quality has never been measured, a few empirical studies have evaluated the fairness of explanations.

Specifically, Balagopalan et al. (2022) train a human-interpretable model to imitate a black-box model, and quantify explanation quality using fidelity, defined as agreement with the black-box predictions. They find that fidelity (and thus explanation quality and reliability) varies across groups, but their experiments are restricted to binary classifiers, and to fidelity as the only explanation method. By contrast, Dai et al. (2022) evaluate various post hoc explanation methods across different evaluation metrics. They show that explanations can vary in quality across demographic groups, leading to fairness concerns, though their experiments are also restricted to binary classifiers. Neither work considers regression tasks or examines how personalization would affect differences in explanation quality across groups. These constraints limit the practical relevance of existing empirical results, as real-world scenarios do not always align with such settings.

**Link to Fairness.** Fairness in machine learning aims to mitigate biased outcomes affecting individuals or groups (Mehrabi et al., 2022). Past works have defined individual fairness, which seeks similar performance for similar individuals (Dwork et al., 2011), or group fairness (Dwork & Ilvento, 2019; Hardt et al., 2016), which seeks similar performance across different groups. Within this literature, most methods, metrics, and analyses are intended for classification tasks (Pessach & Shmueli, 2022). As for the fair regression literature, authors focus on designing fair learning methods (Hebert-Johnson et al., 2018; Berk et al., 2017; Fukuchi et al., 2013; Pérez-Suay et al., 2017; Calders et al., 2013), such as multicalibration, or defining fairness criteria for regression tasks (Gursoy & Kakadiaris, 2022; Agarwal et al., 2019). By contrast, our approach does not require equal performance across individuals or groups. Instead, we study a relaxed fairness notion: ensuring that no group is systematically harmed by personalization. We propose a framework to evaluate whether this weaker fairness criterion is satisfied, both theoretically and empirically, rather than proposing corrective algorithms.

## 3 BACKGROUND: BENEFIT OF PERSONALIZATION FRAMEWORK

Let $\mathcal{X}, \mathcal{S}, \mathcal{Y}$ denote, respectively, the input feature, group attribute, and outcome spaces. A *personalized model* $h_p : \mathcal{X} \times \mathcal{S} \to \mathcal{Y}$ aims to predict an outcome variable $y \in \mathcal{Y}$ using both an input feature vector $x \in \mathcal{X}$ and a vector of group attributes $s \in \mathcal{S}$. In contrast, a *generic model* $h_0 : \mathcal{X} \to \mathcal{Y}$ does not use group attributes. We consider that a fixed data distribution $P = P_{\mathbf{X}, \mathbf{S}, \mathbf{Y}}$ is given, and that $h_0$ and $h_p$ are trained to minimize a loss over a training dataset $\mathcal{D}_{train}$.

**Cost.** We first evaluate how a model $h$ (generic or personalized) performs for a given group.

**Definition 3.1** (Expected Group Cost). The expected cost of model $h$ for the group $s \in \mathcal{S}$ as measured by the cost function cost is defined as: $C(h, s) \triangleq \mathbb{E}_P[\text{cost}(h, \tilde{\mathbf{X}}, \mathbf{Y}) \mid \mathbf{S} = s]$, where $\tilde{\mathbf{X}} = \mathbf{X}$ for a generic model $h_0$, and $\tilde{\mathbf{X}} = (\mathbf{X}, \mathbf{S})$ for a personalized model $h_p$.

In what follows, we use cost and expected cost interchangeably, with the convention that lower cost means better performance. In practice, the cost is evaluated over a set, $\mathcal{D}$, that is independent from the train set. Costs of interest are shown in Table 1: top rows focus on prediction accuracy (loss and evaluation metrics), while bottom ones address explanation quality (sufficiency and incomprehensiveness). As explanation metrics are less common than accuracy metrics, we review them next.

**Cost for Explainability.** We assume access to an *auxiliary explanation method* that assigns importance scores to input features—e.g., based on the magnitude of input gradients. Then, the *explanation quality metric* measures whether the features with the highest importance scores are actually meaningful (see Nauta et al. (2023) for a review). We use *sufficiency* and *incomprehensiveness* as explanation quality metrics to illustrate our framework and apply our framework using Integrated Gradients, DeepLIFT and Shapley Value Sampling. These metrics quantify the change in prediction when the most important features are removed or retained. For a comprehensive discussion of our rationale in selecting these metrics, see Appendix A. We emphasize that importance is defined relative to the explanation method, not to any ground truth. This is by design: the goal is not to assume a known set of truly important features, but to assess how well a given explanation method identifies features that meaningfully affect the model's prediction.

**Benefit of Personalization.** We can quantify the impact of a personalized model in terms of the benefit of personalization, defined next:

**Definition 3.2** (Group Benefit of Personalization (G-BoP) (Monteiro Paes et al., 2022))**.** The gain from personalizing a model can be measured by $\text{G-BoP}(h_0, h_p, s) \triangleq C(h_0, s) - C(h_p, s)$, comparing the costs of the generic $h_0$ and personalized models $h_p$ for group $s \in \mathcal{S}$. By convention, $\text{G-BoP} > 0$ if the personalized model performs better than the generic one.

We use $\text{G-BoP}_P$ and $\text{G-BoP}_X$ to refer to G-BoP for prediction and explanation respectively – see Appendix B Table 3 for concrete examples. For example, for prediction evaluation in regression using MSE G-BoP is: $\mathbb{E}\left[\|h_0(\mathbf{X}) - \mathbf{Y}\|^2 \mid \mathbf{S} = s\right] - \mathbb{E}\left[\|h_p(\mathbf{X}, s) - \mathbf{Y}\|^2 \mid \mathbf{S} = s\right]$, and for incomprehensiveness $\mathbb{E}\left[\|h_p(\mathbf{X}, s) - h_p(\mathbf{X}_{\setminus J}, s_{\setminus J})\|^2 \mid \mathbf{S} = s\right] - \mathbb{E}\left[\|h_0(\mathbf{X}) - h_0(\mathbf{X}_{\setminus J})\|^2 \mid \mathbf{S} = s\right]$. To evaluate whether all groups benefit from personalization, or if any are harmed, we use the following definition as our final assessment metric:

**Definition 3.3** (Benefit of Personalization (BoP) (Monteiro Paes et al., 2022))**.** The BoP is defined as: $\gamma(h_0, h_p) \triangleq \min_{s \in \mathcal{S}}(\text{G-BoP}(h_0, h_p, s))$, i.e., the minimum group BoP value across groups $s \in S$ to capture the worst group improvement, or degradation, resulting from personalization.

A positive $\gamma$ indicates that all groups receive better performance with respect to the cost function. Contrary to this, a negative $\gamma$ reflects that at least one group is disadvantaged by personalization. When $\gamma$ is small or negative, the practitioner might want to reconsider the use of personalized attributes in terms of fairness with respect to all groups. When $\gamma$ is used to evaluate improvement in prediction and explanation, it is referred to as $\gamma_P$ and $\gamma_X$, respectively.

*Remark.* The definitions of G-BoP and $\gamma$ were originally introduced in Monteiro Paes et al. (2022). While formally applicable to any cost function, these definitions have only been studied and used with binary costs—such as 0-1 classification loss or false positive/negative rates—due to a theoretical gap that prevents their use with continuous costs, including an analysis of prediction and explanation for regression tasks. Since a holistic analysis of prediction and explanation across machine learning tasks is our primary focus, addressing this gap is central to our contribution in Section 5.

## 4 Impact of Personalization on Prediction And Explainability

This section provides the first formal analysis showing that personalization's effect on prediction does not determine its effect on explainability, highlighting the need to evaluate both. A common intuition in machine learning is that if personalization improves prediction, it should also improve the quality of explanations derived from the model. This intuition is reflected in the XAI literature, where evaluation practices often conflate model accuracy with explanation correctness. For example, a recent survey notes that "these commentaries relate to the inherent coupling of evaluating the black box' predictive accuracy with explanation quality. As pointed out by Robnik-Šikonja and Bohanec (Robnik-Sikonja & Bohanec, 2018), the correctness of an explanation and the accuracy of the predictive model may be orthogonal" (Nauta et al., 2023). This assumption also appears implicitly in many high-stakes applications, where explanations from high-performing models are used to draw insights about real-world structure (Elmarakeby et al., 2021; Chereda et al., 2021). Despite its prevalence, this presumed connection between predictive performance and explanation quality has not been formally analyzed in the context of personalization.

Theorems 4.1 and 4.2 prove that prediction gains and explanation gains can diverge, demonstrating that gains in prediction performance (measured by $\text{BoP}_P$) and gains in explanation quality (measured

by $\text{BoP}_X$) need not align. Theorem 4.4 provides a partial converse, identifying an additive setting where the two align. Though idealized, this boundary case clarifies when practitioners can trust prediction and explanation to align. Proofs are in Appendix C.

**No Prediction Benefit Does Not Imply No Explainability Benefit.** The following theorem shows that a personalized model may match a generic model in accuracy, yet offer better explanation. Thus, focusing only on prediction can overlook significant interpretability gains.

**Theorem 4.1.** *There exists a data distribution $P_{\mathbf{X},\mathbf{S},\mathbf{Y}}$ such that the Bayes optimal classifiers $h_0$ and $h_p$ satisfy $\gamma_P(h_0, h_p) = 0$ (with $\gamma_P$ measured by 0-1 loss) and $\gamma_X(h_0, h_p) > 0$ (with $\gamma_X$ measured by sufficiency and incomprehensiveness).*

**Example 1.** We illustrate Theorem 4.1 with a real-world example. Consider a model with many input features that are partially redundant, for instance, a loan approval model that uses credit score, income, and debt-to-income ratio. Adding a personal feature that is highly correlated with existing features may not change the predictions. However, it can alter the explanation if that feature is the most direct or informative input. For example, adding a binary feature like "pre-approved by another bank", which is strongly correlated with existing features, may leave predictions unchanged, but an explainer might now assign most importance to this new feature because it provides a clearer justification. Figure 5 illustrates the construction behind the proof for sufficiency, where both generic $h_0$ and personalized $h_p$ models predict perfectly (left side), yet only keeping the most important feature for each (right side) shows that the personalized model is more explainable. For this distribution, $\text{G-BoP}_P(h_0, h_p, s) = 0$ and $\text{G-BoP}_X(h_0, h_p, s) > 0$ for each group $s$, so all groups are impacted similarly by personalization. Figure 6 illustrates the proof for incomprehensiveness.

**No Prediction Harm Does Not Imply No Explainability Harm.** A personalized model may match a generic model in accuracy yet offer worse explanations. Thus, focusing only on predictive performance can obscure significant harms to explainability.

**Theorem 4.2.** *There exists a data distribution $P_{\mathbf{X},\mathbf{S},\mathbf{Y}}$ such that the Bayes optimal classifiers $h_0$ and $h_p$ satisfy $\gamma_P = 0$ (with $\gamma_P$ measured by 0-1 loss) and $\gamma_X < 0$ (with $\gamma_X$ measured by incomprehensiveness).*

**Example 2.** To illustrate Theorem 4.2, consider a pneumonia detection model using chest X-ray findings that perfectly predict outcomes. Adding white blood cell count leaves accuracy unchanged, but the personalized model now splits importance between X-ray findings and white blood cell count. The explanation is worse because it's now split across two features, making it less clear which feature drives the decision, even though the X-ray alone was already perfectly predictive.

Additionally, Theorem 4.3 proves this phenomenon for both sufficiency and incomprehensiveness by showing how personalization might not alter predictive accuracy across groups, but it might affect explainability differently for different groups.

**Theorem 4.3.** *There exists a data distribution $P_{\mathbf{X},\mathbf{S},\mathbf{Y}}$ such that the Bayes optimal classifiers $h_0$ and $h_p$ satisfy $\text{G-BoP}_P(h_0, h_p, s) = 0$ (measured by 0-1 loss) for all groups $s$, but some groups have $\text{G-BoP}_X(h_0, h_p, s) > 0$ while others have $\text{G-BoP}_X(h_0, h_p, s) < 0$ (measured by sufficiency and incomprehensiveness).*

Figure 7 illustrates the proof for sufficiency, where both generic $h_0$ and personalized $h_p$ models predict perfectly (left), yet only keeping the most important feature for each (right) shows that the personalized model is more explainable for the group $(s' = 1, s = 0)$, and less explainable for group $(s' = 0, s = 1)$. Figure 8 illustrates the proof for incomprehensiveness.

Together, Theorems 4.1, 4.2 and 4.3 show that knowing $\gamma_P = 0$ provides no information about $\gamma_X$. This motivates the need to evaluate both prediction and explainability, as we offer to do in Section 5.

**No Explainability Benefit Can Imply No Prediction Benefit.** We now ask the converse: can a lack of explainability benefit imply no predictive benefit? We show this is true for a simple additive model, as long as two notions of explainability measures –sufficiency and incomprehensiveness– do not see any benefit.

**Theorem 4.4.** *Assume that $h_0$ and $h_p$ are Bayes optimal regressors and $P_{\mathbf{X},\mathbf{S},\mathbf{Y}}$ follows an additive model, i.e., $\mathbf{Y} = \alpha_1 \mathbf{X_1} + \cdots + \alpha_t \mathbf{X_t} + \alpha_{t+1} \mathbf{S_1} + \cdots + \alpha_{t+k} \mathbf{S_k} + \epsilon$, where $\mathbf{X_1}, \cdots, \mathbf{X_t}$ and $\mathbf{S_1}, \cdots, \mathbf{S_k}$ are independent, and $\epsilon$ is independent random noise. Then, if for $s \in \mathcal{S}$ we have $\text{G-BoP}_{suff}(h_0, h_p, s) = \text{G-BoP}_{incomp}(h_0, h_p, s) = 0$, then $\text{G-BoP}_P(h_0, h_p, s) = 0$. Consequently, if, for all groups $s$, $\text{G-BoP}_{suff}(h_0, h_p, s) = \text{G-BoP}_{incomp}(h_0, h_p, s) = 0$, then $\gamma_P = 0$.*

This theorem demonstrates that under an additive model, if there is no benefit in explanation quality, then there is also no benefit in prediction accuracy. Figure 9 illustrates this proof. Additionally, we get the following corollary:

**Corollary 4.5.** *Under the assumptions of Theorem 4.4, if for $s \in \mathcal{S}$, we have G-BoP$_P(h_0, h_p, s) \neq 0$, then it also holds that G-BoP$_{suff}(h_0, h_p, s) \neq 0$ or G-BoP$_{incomp}(h_0, h_p, s) \neq 0$. Consequently, if $\gamma_P \neq 0$, then there exists a group $s \in \mathcal{S}$ such that G-BoP$_{suff}(h_0, h_p, s) \neq 0$ or G-BoP$_{incomp}(h_0, h_p, s) \neq 0$.*

This theorem shows that if personalization affects prediction, it must also affect explanation for at least one explainability measure and one demographic group. This result establishes a rare direct link between explanation and prediction, in a simplified linear setting. Proving this for general models remains an open question.

## 5 Testing Personalization's Impact on Prediction and Explanation

Having emphasized the importance of evaluating both prediction and explainability, we now introduce a methodology to assess them in practice. The true BoP $\gamma$, defined over the whole data distribution, is inaccessible and needs to be estimated from finite samples. Then, if its estimate $\hat{\gamma}$ is positive, one must consider whether the true $\gamma$ is also likely to be positive. In scenarios where personalization incurs a price—such as requesting sensitive user information—one should determine how large $\hat{\gamma}$ must be to ensure that the true benefit exceeds a desired threshold $\gamma \geq \epsilon$. This section analyzes the validity of BoP hypothesis testing and provides guidelines for its application.

### 5.1 Validity of Hypothesis Tests

**Hypothesis Tests.** Given an audit dataset $\mathcal{D}$ with $k$ binary group attributes, we want to know whether personalization improves each group by at least $\epsilon > 0$. We formalize the null and the alternative hypotheses using a standard framework for the BoP (Monteiro Paes et al., 2022):

$$H_0 : \gamma(h_0, h_p; \mathcal{D}) \leq 0 \Leftrightarrow \text{Personalized } h_p \text{ does not bring any gain for at least one group,}$$
$$H_1 : \gamma(h_0, h_p; \mathcal{D}) \geq \epsilon \Leftrightarrow \text{Personalized } h_p \text{ yields at least } \epsilon \text{ improvement for all groups.}$$

Importantly, $H_0$ and $H_1$ are not complementary, because we want to reject the null in favor of the alternative if the impact is both positive *and* practically meaningful, i.e., $\geq \epsilon$. With these hypotheses, we ask: can we rule out that there is no harm *and* assert a meaningful benefit of at least $\epsilon$?

The improvement $\epsilon$ is in cost function units and represents the improvement for the group that benefits the least from the personalized model. The value $\epsilon$ is domain-specific and should be chosen by the practitioner. For example, in healthcare, if personalization requires time-intensive and sensitive inputs—like mental health assessments—it may only be justified if it improves diagnostic accuracy by at least a few points, making $\epsilon$ a clinically and ethically meaningful threshold. In such cases, $\epsilon$ becomes a threshold for balancing speed and clinical value.

Once $\epsilon$ is chosen, the practitioner may run the hypothesis test by computing the estimate $\hat{\gamma}$ on $\mathcal{D}$ and follow the rule: $\hat{\gamma} \geq \epsilon \Rightarrow$ *Reject $H_0$: Conclude that personalization yields at least $\epsilon$ improvement for all groups*. We note that different testing strategies could also be used. To capture this generality, we define a decision function $\Psi : (h_0, h_p, \mathcal{D}, \epsilon) \to \{0, 1\}$, where $\Psi = 1$ indicates rejection of $H_0$. In our case, $\Psi(h_0, h_p, \mathcal{D}, \epsilon) = (\hat{\gamma} \geq \epsilon)$. Regardless of its specific form, our goal is to assess the validity of *any* test aiming to evaluate the impact of personalization $\gamma$.

**Invalidity of the Tests: Probability of Error.** We quantify the (in)validity of a test in terms of its probability of error, defined as: $P_e = \frac{1}{2}\left(\Pr(\text{Rejecting } H_0 | H_0 \text{ is true}) + \Pr(\text{Failing to reject } H_0 | H_1 \text{ is true})\right)$. The probability of error is a composite measure that weights Type I and Type II errors, under the assumption that $H_0$ and $H_1$ are equiprobable with probability $1/2$.

We propose to derive a minimax lower bound on the probability of error $P_e$. This involves considering the worst-case data distributions that maximize $P_e$ and the best possible decision function $\Psi$ that minimizes it. Notably, a high lower bound guarantees a high error probability for *any* test with $H_0$ and $H_1$ on the BoP, flagging settings where testing the impact of personalization is unreliable.

Figure 2: **Testing personalization for prediction and explanation depends on learning task.** Lower bound on the probability of error $P_e$ with respect to number of personal attributes $k$, for dataset sizes $N = 10^2$, $10^3$, and $10^4$ with $\epsilon = 0.01$. In classification (orange), the bound is fixed by the categorical nature of the individual BoP (iBoP) and is identical for prediction and explanation. In regression (green and blue), $P_e$ depends on the spread of individual BoPs—parameterized by variance $\sigma^2$ (Gaussian) or scale $b$ (Laplace). Smaller variance or scale allows more attributes before testing becomes unreliable ($P_e \geq 0.25$). Computed for $m = \lfloor N/d \rfloor$ samples per group with $d = 2^k$ groups.

**Theorem 5.1.** *Consider $k$ binary group attributes, $\mathcal{S} \triangleq \{0, 1\}^k$, that specify $d \triangleq |\mathcal{S}| = 2^k$ groups, each containing $m_j$ individuals, $j = 1, .., d$. Let $H_0$ (resp. $H_1$) denote the data distributions under which the generic model $h_0$ (resp. the personalized model $h_p$) performs better, i.e., $\gamma \leq 0$ (resp. $\gamma \geq \epsilon$). Denote $H'_0$ (resp. $H'_1$) the data distributions that are mixtures of components in $H_0$ (resp. $H_1$). Then, there exists $P \in H'_0$ (resp. $Q \in H'_1$), for which the individual benefit of personalization $\mathbf{B} = cost(h_0, \tilde{\mathbf{X}}, \mathbf{Y}) - cost(h_p, \tilde{\mathbf{X}}, \mathbf{Y})$, follows a probability density $p$ (resp. $p^\epsilon$ for one group), where $\mathbb{E}_p[\mathbf{B}] = 0$, and $\mathbb{E}_{p^\epsilon}[\mathbf{B}] = \epsilon$, such that:*

$$\min_\Psi \max_{\substack{P_0 \in H'_0 \\ P_1 \in H'_1}} P_e \geq \frac{1}{2} \left( 1 - \frac{1}{2\sqrt{d}} \left[ \frac{1}{d} \sum_{j=1}^d \left( \mathbb{E}_{p^\epsilon} \left[ \frac{p^\epsilon(\mathbf{B})}{p(\mathbf{B})} \right] \right)^{m_j} - 1 \right]^{\frac{1}{2}} \right). \tag{1}$$

The proof for Theorem 5.1 is in Appendix D.1. We note that the lower bound cannot exceed $1/2$, which is expected. In a binary hypothesis test with equal priors, a decision rule corresponding to random guessing would yield $P_e = 1/2$. Consequently, the optimal probability of error (the minimum over all decision rules $\Psi$) must lie in the interval $[0, 1/2]$.

Crucially, this lower bound can be tailored to the practitioner's specific use case, i.e., to the distribution of the individual benefit $\mathbf{B}$ under $H_0$ and $H_1$. For example, if $\mathbf{B}$ is known or observed to follow a Laplace distribution with scale $b$, the practitioner should choose $p = \mathrm{Laplace}(0, b)$ and $p^\epsilon = \mathrm{Laplace}(\epsilon, b)$. Figure 3 shows the expression of the lower bound for the Laplace distribution (proof provided in Appendix D.8). If none of these standard distributions provided in the appendix are a good match for the BoP distribution, Theorem 5.1 remains valid for any distribution, as long as its probability density function is known. In such cases, practitioners may use flexible density estimation tools, such as normalizing flows, to approximate the PDF from data and apply Theorem 5.1 directly. The next corollary expresses it for distributions in the exponential family, which we use to find a bound for when $\mathbf{B}$ follows a Gaussian distribution (see Appendix D.5).

**Corollary 5.2.** *The lower bound in Th. 5.1 for distributions $p, p^\epsilon$ in the exponential family (parameter $\theta$, moment generating function $M$) is:* $\min_\Psi \max_{\substack{P_0 \in H'_0 \\ P_1 \in H'_1}} P_e \geq$

$\frac{1}{2} \left( 1 - \frac{1}{2\sqrt{d}} \left[ \frac{1}{d} \sum_{j=1}^d \left( \frac{M_p(2\Delta\theta)}{M_p(\Delta\theta)^2} \right)^{m_j} - 1 \right]^{\frac{1}{2}} \right)$ *with $\Delta\theta = \theta^\epsilon - \theta$.*

The proof for Corollary 5.2 is in Appendix D.2. These results generalize and tighten an existing bound for categorical distribution only (Monteiro Paes et al., 2022) (see Appendix D.3) and provide the first general framework to evaluate the (in)validity of hypothesis tests on personalization for prediction and explanation, and across supervised machine learning tasks.

*Remark.* These bounds apply to any metric that can be formulated as an evaluation score and used to compare model performance across subgroups (across classification and regression)–i.e., our statistical testing tools are not tied to any particular explainability or performance measure or method.

**Experimental Design: Group Attributes, Sample Size, and Detectable Gain.** We investigate how probability of error depends on the dataset, and how it determines the practitioner's ability to test the impact of personalization. For example, with a fixed number of individuals $N$, a larger number of personal attributes $k$ increases the number of groups $d = 2^k$, reducing the number of samples per group, which increases the risk of error. Accordingly, if the practitioner commits to a fixed $k$ to test a desired gain $\epsilon$ (resp. fixed $k$ and $N$), they need a minimum group size $m$ to keep the error bound below a desired level, as shown next.

**Corollary 5.3.** *To ensure* $\min \max P_e \leq v$ *for a chosen threshold* $v$, *equal group sizes must satisfy* $m \geq m_{\min}$, *where:* $m_{min} = \frac{\log\left(4 \cdot 2^k (1-2v)^2 + 1\right)}{\log(1+\epsilon^2)}$ *for a categorical BoP,* $m_{min} = \frac{\sigma^2}{\epsilon^2} \log\left(1 + 4 \cdot 2^k (1-2v)^2\right)$ *for a Gaussian BoP of variance* $\sigma^2$, *and* $m_{\min} = \frac{b}{\epsilon} \log\left(1 + 2^{2+k} (1-2v)^2\right)$ *for a Laplace BoP of scale b.*

App. E provides practitioners with another dataset-specific feasibility check: Corollary E.1 bounds the maximum number of attributes that can be used before the lower bound error exceeds 25%.

## 5.2 PRACTICAL CONSIDERATIONS WHEN TESTING PREDICTION AND EXPLANATION

We examine how the lower bound in Theorem 5.1 depends on the distribution of individual BoPs $\mathbf{B}$, and how this determines the practitioner's ability to test for prediction or explanation gains.

**Testing Prediction and Explanation in Classification Tasks.** When the task is classification with 0-1 loss, the individual BoPs follow categorical distributions with values in $\{-1, 0, 1\}$:

$$\mathbf{B}_P = (h_0(\mathbf{X}) \neq \mathbf{Y}) - (h_p(\mathbf{X}, \mathbf{S}) \neq \mathbf{Y}), \quad \mathbf{B}_X = (h_0(\mathbf{X}) \neq h_0(\mathbf{X}_J)) - (h_p(\mathbf{X}, \mathbf{S}) \neq h_p(\mathbf{X}_J, \mathbf{S}_J))$$

for prediction and explanation (e.g., sufficiency), respectively –see costs in Table 1. In this setting, the lower bound in Theorem 5.1 is identical for prediction and explanation (see Figure 3, bottom): either both are testable, or neither is.

Figure 2 shows the lower bound on the probability of error $P_e$ as a function of $k$, for typical dataset sizes in medical settings $N \in \{10^2, 10^3, 10^4\}$. In classification (orange curves), even a small number of personal attributes $k$ leads to high error lower bounds. For instance, at $N = 100$ and $k = 1$, the bound already exceeds 40%, making reliable testing impossible for both prediction and explanation.

**Testing Prediction and Explanation in Regression Tasks.** In regression, the situation is more nuanced. For instance, with MSE loss, we have continuously valued individual BoP random variables:

$$\mathbf{B}_P = |h_0(\mathbf{X}) - \mathbf{Y}|^2 - |h_p(\mathbf{X}, \mathbf{S}) - \mathbf{Y}|^2, \quad \mathbf{B}_X = |h_0(\mathbf{X}) - h_0(\mathbf{X}_J)|^2 - |h_p(\mathbf{X}, \mathbf{S}) - h_0(\mathbf{X}_J, \mathbf{S}_J)|^2,$$

for prediction and explanation, respectively. Suppose these follow Laplace distributions with scales $b_P$ and $b_X$. Then, the lower bounds will differ for prediction and explanation (Figure 3, bottom): one could be testable while the other is not, highlighting an asymmetry absent in the classification case.

As illustrated in Figure 2, smaller scale values ($b$) allow for a larger number of personal attributes $k_{\max}$ to be tested without theoretical barriers. Unlike classification, there is no proof that regression tasks cannot support reliable testing of personalization for dataset sizes encountered in medical settings $N \in \{10^2, 10^3, 10^4\}$, even with many personal attributes $k$.

## 6 CASE STUDIES: EVALUATING PERSONALIZATION ON REAL DATASETS

We illustrate how to use our results to investigate the impact of personalization on prediction and explanation, to reveal the many cases where reliable testing is in fact impossible. This section focuses on one real-world healthcare scenario, while other scenarios are provided in Appendix G. **Remark.** Across these hypothesis tests we always evaluate if there is a benefit of personalization, i.e. $\gamma > \epsilon > 0$, but interested practitioners may want to evaluate whether an existing machine learning model could harm one group. In that case the hypothesis test should be flipped, i.e. $\gamma < \epsilon < 0$.

**Healthcare Scenario.** Consider MIMIC-III (Medical Information Mart for Intensive Care) (Johnson et al., 2016), a dataset of patients admitted to critical care units at a large tertiary hospital –containing vital signs, medications, lab results, diagnoses, imaging reports, and outcomes such as length of stay.

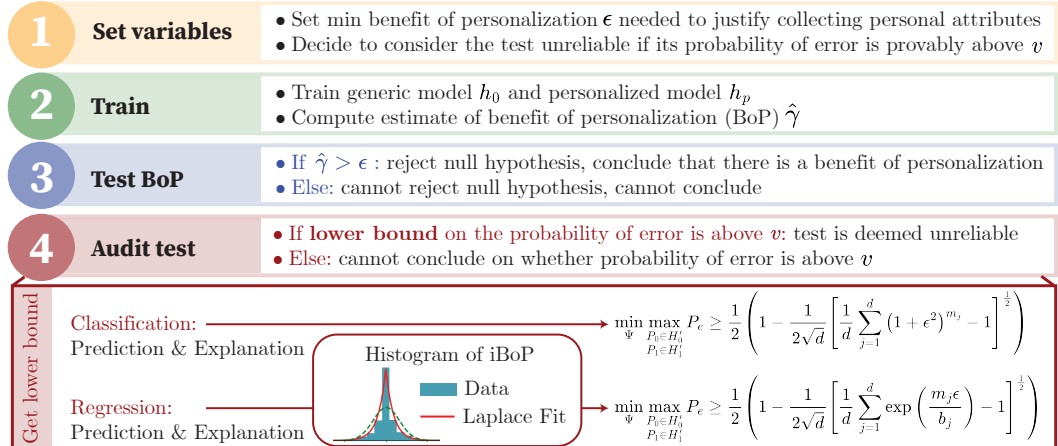

| 1 | Set variables | • Set min benefit of personalization $\epsilon$ needed to justify collecting personal attributes
• Decide to consider the test unreliable if its probability of error is provably above $v$ |
|---|---|---|
| 2 | Train | • Train generic model $h_0$ and personalized model $h_p$
• Compute estimate of benefit of personalization (BoP) $\hat{\gamma}$ |
| 3 | Test BoP | • If $\hat{\gamma} > \epsilon$ : reject null hypothesis, conclude that there is a benefit of personalization
• Else: cannot reject null hypothesis, cannot conclude |
| 4 | Audit test | • If **lower bound** on the probability of error is above $v$: test is deemed unreliable
• Else: cannot conclude on whether probability of error is above $v$ |

Figure 3: **Summary of the steps to test BoP for prediction and explanation.**

Suppose that a practitioner has developed a deep learning model to predict a patient's length of stay (regression) or whether the length of stay exceeds 3 days (classification) – see details in Appendix F.1. They are wondering whether their model should be personalized by including (or not) two personal attributes: Age × Race $\in \{18 - 45, 45+\} \times \{\text{White(W)}, \text{NonWhite(NW)}\}$. However, they are concerned this could disadvantage some groups, not only by reducing prediction accuracy but also by limiting the ability to uncover factors that explain critical care duration. We provide a step-by-step procedure to use our framework to evaluate the benefit of personalization (summarized in Figure 3).

① **Select $\epsilon$ and $v$.** The practitioner first chooses the minimum improvement they expect from personalization—$\epsilon_P$ for prediction and $\epsilon_X$ for explanation (e.g., $\epsilon_P = \epsilon_X = 0.002$). They then set a tolerance threshold $v$ for the probability of error beyond which they will not trust the hypothesis test (e.g., $v = 25\%$).

② **Report empirical benefits of personalization.** The practitioner trains $h_0$ and $h_p$ (with additional attributes age and race) and reports empirical personalization benefits in Table 2 (0–1 loss for classification, MSE for regression). They utilize the Integrated Gradients explainer method and evaluate it using the sufficiency and incomprehensiveness metrics. Across tasks, some groups seem to show benefits for prediction but harm for explanation, and vice versa. This should not be surprising given the results of Section 4, which show that prediction and explanation gains can diverge.

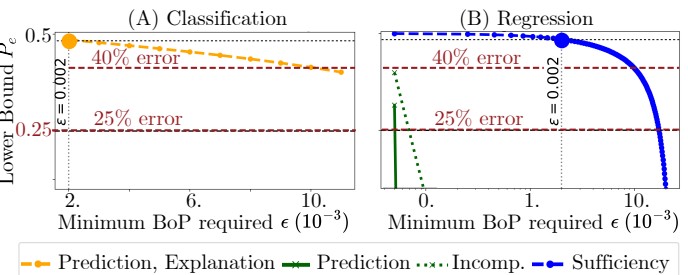

Figure 4: **Lower bound of $P_e$ vs. $\epsilon$ on MIMIC-III**: classification (A) and regression with Laplace (green) and Gaussian (blue) models for the individual BoPs (B). At the minimum BoP set in this case study ($\epsilon = 0.002$), testing personalization for prediction and explanation is impossible for classification (same for sufficiency for regression) as $P_e \geq 40\%$ regardless of the hypothesis test.

③ **Perform hypothesis test.** The practitioner assesses whether $\hat{\gamma}$ exceeds $\epsilon_P$ or $\epsilon_X$. It does for all metrics with a positive $\hat{\gamma}$, hence they can reject the null hypothesis for these cases.

④ **Assess reliability of the results.** Next, the practitioner assesses whether the empirical results are statistically meaningful using the framework from Section 5. *For the classification model*, the lower bound on the probability of error exceeds 40% (Figure 4, $\epsilon = 0.002$), indicating that it is not even possible to test whether personalization helps or harms performance. As a result, the practitioner would likely retain the generic classifier. *For the regression model*, they examine the distributions of individual BoPs, $\mathbf{B}_P$ and $\mathbf{B}_X$ (Figure 3, bottom, and Appendix F.1). Sufficiency is best fit by Gaussians with varying variances; prediction and incomprehensiveness align with Laplace distributions of different scales. The corresponding lower bounds on error exceed 40%

Table 2: Benefits of personalization ($\hat{C}(h_0) - \hat{C}(h_p)$) on the MIMIC-III test set for predicting length of stay (LOS): regression or classification (LOS > 3 days). Incomprehensiveness is abbreviated as incomp. and population as pop. Values that are worsened by $h_p$ are colored red.

| Group | | Classification | | | | Regression | | |
|---|---|---|---|---|---|---|---|---|
| | $n$ | Prediction | Incomp. | Sufficiency | $n$ | Prediction | Incomp. | Sufficiency |
| White, 45+ | 8443 | 0.0063 | -0.0226 | 0.0053 | 8379 | 0.0021 | -0.0906 | 0.1914 |
| White, 18–45 | 1146 | 0.0044 | 0.0489 | 0.0244 | 1197 | 0.0023 | 0.1219 | 0.2223 |
| NonWhite, 45+ | 3052 | -0.0026 | -0.0023 | 0.0029 | 3044 | 0.0108 | -0.0501 | 0.3494 |
| NonWhite, 18–45 | 696 | -0.0216 | 0.0560 | 0.0072 | 717 | 0.0212 | 0.0441 | 0.3293 |
| All Pop. | 13337 | 0.0026 | -0.0077 | 0.0065 | 13337 | 0.0051 | -0.0550 | 0.2376 |
| **Minimal BoP** | 13337 | -0.0216 | -0.0226 | 0.0029 | 13337 | 0.0021 | -0.0906 | 0.1914 |

for sufficiency—making it untestable—but fall below 5% for prediction and incomprehensiveness (Figure 4, $\epsilon = 0.002$). Now, we provide insights that were gained from applying our framework to this scenario, and others in Appendix G.

**Insight: A high empirical benefit of personalization $\hat{\gamma}$ can be misleading.** As shown in Table 2 and lower bounded in ④, the regression experiment reports the largest apparent benefit for sufficiency ($\hat{\gamma} = 0.1914$), yet the data did not permit a valid test, making the result inconclusive. Prediction showed a much smaller benefit ($\hat{\gamma} = 0.0021$), but our analysis found no barriers to testing, and the null was rejected. This shows that large $\hat{\gamma}$ does not guarantee a valid conclusion; empirical values must be paired with our framework to assess validity.

**Insight: The choice of improvement threshold $\epsilon$ is key.** Increasing $\epsilon$ reduces the lower bound on the probability of error $P_e$, making hypothesis testing potentially less unreliable (Figure 4), but also raises the bar for rejecting the null, requiring a larger $\hat{\gamma}$. Thus, $\epsilon$ trades off test validity against ability to detect effects. In real-world applications, $\epsilon$ reflects the minimum performance gain a practitioner needs to justify collecting costly personal data (e.g., genetic markers). For instance, $\epsilon = 0.002$ in length-of-stay prediction accuracy may be worthwhile. Converting this back to the original units, the improvement is approximately 0.06 days per patient. Although small at the individual level, across 100 patients per day this amounts to a cumulative gain of about 6 patient-days of hospital stay estimated more accurately each day. This illustrates that $\epsilon$ should be set based on practical value, not statistical convenience, even though higher $\epsilon$ values tend to make hypothesis tests more reliable.

**Insight: Results do not depend on the explanation method.** Table 2 reports results with Integrated Gradients (Sundararajan et al., 2017). Since our framework applies to any explanation method, we test whether this choice affects the evaluation of the impact of personalization. Appendix G analyzes Shapley Value Sampling (Štrumbelj & Kononenko, 2010) and DeepLIFT (Shrikumar et al., 2017), finding substantial agreement across the methods—though effect sizes differ.

**Insight: Personalization is hard to evaluate across medical datasets.** To show the practicality of the framework, we also include experiments on the UCI Heart Dataset (Janosi et al., 1989) and the MIMIC-III Kidney injury cohort (Suriyakumar et al., 2023), again utilizing a range of explanation methods (see App. G). Using the same $\epsilon$ as above, no test is valid for the S.V.S explainer on the Heart dataset (see caption of Table 4). This shows the difficulty of reliably evaluating personalization. Generally, this analysis points to a limitation of personalized medicine and healthcare: while personalization may yield improvements, demonstrating them reliably can be infeasible—restricting applicability.

## CONCLUDING REMARKS

We present a unified framework for evaluating the benefits of personalization with respect to both prediction accuracy and explanation quality, facilitating nuanced decisions regarding the use of personal attributes. Our analysis shows that in many practical settings, particularly classification tasks, the statistical conditions required to validate personalization are often unmet. As a result, even when personalization shows empirical gains, meaningful validation may not be feasible.

**Limitations & Future Work.** While we relax several assumptions relative to prior work, our theoretical results still rely on assumptions not always met in practice; further reducing them remains an important direction. Additionally, while we focused on explanation quality due to its importance in clinical adoption, our results in Section 5 extend to other goals. Future work can build on this framework to evaluate additional desiderata such as fairness, robustness, and uncertainty calibration.

ACKNOWLEDGMENTS

Nina Miolane acknowledges support from NSF Grant 2313150 and the NSF CAREER Award 240158. Guillermo Bernárdez acknowledges support from the Chan Zuckerberg Initiative, the UC Noyce Foundation, and Arlequin AI. Louisa Cornelis acknowledges support from the Chan Zuckerberg Initiative, the UC Noyce Foundation, and the NSF Graduate Research Fellowship Program (NSF GRFP) Grant 2139319.

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

## A    EXTENDED RELATED WORKS

We provide additional extended works about explainability methods and fairness of recourse below.

**Explainability** Typical approaches to model explanation involve measuring how much each input feature contributes to the model's output, highlighting important inputs to promote user trust. This process often involves using gradients or hidden feature maps to estimate the importance of inputs (Simonyan et al., 2014; Smilkov et al., 2017; Sundararajan et al., 2017; Yuan et al., 2022). For instance, gradient-based methods use backpropagation to compute the gradient of the output with respect to inputs, with higher gradients indicating greater importance (Sundararajan et al., 2017; Yuan et al., 2022). We focus on feature-attribution explanations as they remain the most widely used form of post hoc interpretability in practice (Nauta et al., 2023). To reflect a range of underlying assumptions, we employ three distinct and widely adopted explainers: Integrated Gradients (gradient-based), DeepLIFT (backpropagation-based), and Shapley value sampling (perturbation-based).

The quality of these explanations is often evaluated using the principle of *faithfulness* (Lyu et al., 2024; Dasgupta et al., 2022; Jacovi & Goldberg, 2020), which measures how accurately an explanation represents the reasoning of the underlying model. Two key aspects of faithfulness are *sufficiency* and *comprehensiveness* (DeYoung et al., 2020; Yin et al., 2022); the former assesses whether the inputs deemed important are adequate for the model's prediction, and the latter examines if these features capture the essence of the model's decision-making process. We selected these metrics as they are widely-adopted, model-agnostic measures that directly assess explanation faithfulness through standard perturbation-based evaluation (Serrano & Smith, 2019), aligning with established principles of correctness and completeness in the explainability literature (Nauta et al., 2023).

**Explanations in Practice: Medical Domain** Explainable AI methods are widely deployed in the medical domain, and clinicians routinely interact with explanations when interpreting AI outputs. A recent review identified 454 medical AI articles published between 2018–2022, with 93 analyzed in depth, showing extensive use of explainable AI techniques across diagnostic and clinical decision-support applications (Ali et al., 2023; Salih et al., 2024). A growing body of work shows that explainable AI is already shaping consequential medical decisions across multiple clinical domains. In obstetrics, explainable decision-support systems for gestational diabetes significantly influence clinicians' choices and advice-taking behavior, demonstrating that explanations directly affect medical judgment (Du et al., 2022). In dermatology, domain-specific explanations increase diagnostic accuracy, confidence, and trust, highlighting clinicians' willingness to adopt explainable AI systems in practice (Chanda et al., 2024). In radiology, physicians achieve their highest diagnostic accuracy when receiving AI advice paired with visual explanatory annotations, with non-experts benefiting most from explainable guidance (Gaube et al., 2023). In cardiology, explainable AI methods are used to select and justify heart-failure survival prediction models, with explainability explicitly enabling clinicians to understand model reasoning and make more informed treatment decisions (Moreno-Sánchez, 2023). Together, these studies demonstrate that explanations influence diagnosis, trust, and decision pathways in real clinical environments—underscoring the importance of evaluating whether explanations faithfully reflect model behavior.

**Fairness of Recourse** A related line of work examines fairness of algorithmic recourse, which studies whether different demographic groups face unequal effort to obtain favorable outcomes from a predictive model. Ustun et al. (2019) show that recourse burden can vary sharply across groups, even when recommended actions look formally identical, either because the recourse itself differs or because the real-world effort required to carry it out is unequal. This line of work demonstrates that fair prediction does not guarantee fair recourse. Our framework offers a complementary perspective: instead of analyzing post-hoc interventions, we study when personalization produces unequal benefits or harms across groups in prediction and explanation. Like the recourse literature, our results highlight that different desiderata, here, prediction benefit and explanation benefit, can diverge and therefore must be evaluated jointly.

## B    BOP

In the following table, we show how these abstract definitions can be used to measure BoP for both predictions and explanations, each across both classification and regression tasks. The empirical

population and group BoP are defined as: $\hat{\mathrm{BoP}}(h_0, h_p) = \hat{C}(h_0) - \hat{C}(h_p)$ and $\hat{\mathrm{BoP}}(h_0, h_p, s) = \hat{C}(h_0, s) - \hat{C}(h_p, s)$, respectively.

Table 3: Formal definitions of the benefit of personalization for prediction and explanation metrics, evaluated for subgroup $s$. The generic model $h_0$ takes input $\mathbf{X}$, while the personalized model $h_p$ takes input $(\mathbf{X}, \mathbf{S})$; this corresponds to the quantity previously denoted as $\tilde{\mathbf{X}}$ when referring to an unspecified model $h$ in Table 1. For the explanation metrics, $\mathbf{X}_{\setminus J}$ denotes the input obtained when removing the most important features, and $\mathbf{X}_J$ denotes the complementary set of features that are kept. Likewise, $\mathbf{s}_{\setminus J}$ and $\mathbf{s}_J$ denote the removed and retained subsets of the personalized attribute $s$. Higher BoP values in each row indicate a greater benefit of personalization for subgroup $s$.

| Evaluation Type | Benefit of personalization for group $s$ |
|---|---|
| Predict (Classification, 0-1 loss) | $\Pr(h_0(\mathbf{X}) \neq \mathbf{Y} \mid \mathbf{S} = s) - \Pr(h_p(\mathbf{X}, s) \neq \mathbf{Y} \mid \mathbf{S} = s)$ |
| Predict (Regression, MSE) | $\mathbb{E}\left[\|h_0(\mathbf{X}) - \mathbf{Y}\|^2 \mid \mathbf{S} = s\right] - \mathbb{E}\left[\|h_p(\mathbf{X}, s) - \mathbf{Y}\|^2 \mid \mathbf{S} = s\right]$ |
| Explain (Sufficiency, classification, 0-1 loss) | $\Pr(h_0(\mathbf{X}) \neq h_0(\mathbf{X}_J) \mid \mathbf{S} = s) - \Pr(h_p(\mathbf{X}, s) \neq h_p(\mathbf{X}_J, s_J) \mid \mathbf{S} = s)$ |
| Explain (Sufficiency, regression, MSE) | $\mathbb{E}\left[\|h_0(\mathbf{X}) - h_0(\mathbf{X}_J)\|^2 \mid \mathbf{S} = s\right] - \mathbb{E}\left[\|h_p(\mathbf{X}, s) - h_p(\mathbf{X}_J, s_J)\|^2 \mid \mathbf{S} = s\right]$ |
| Explain (Incomprehensiveness, classification, 0-1 loss) | $\Pr\left(h_p(\mathbf{X}, s) \neq h_p(\mathbf{X}_{\setminus J}, s_{\setminus J}) \mid \mathbf{S} = s\right) - \Pr\left(h_0(\mathbf{X}) \neq h_0(\mathbf{X}_{\setminus J}) \mid \mathbf{S} = s\right)$ |
| Explain (Incomprehensiveness, regression, MSE) | $\mathbb{E}\left[\|h_p(\mathbf{X}, s) - h_p(\mathbf{X}_{\setminus J}, s_{\setminus J})\|^2 \mid \mathbf{S} = s\right] - \mathbb{E}\left[\|h_0(\mathbf{X}) - h_0(\mathbf{X}_{\setminus J})\|^2 \mid \mathbf{S} = s\right]$ |

## C  COMPARISON BoP FOR PREDICTION AND BoP FOR EXPLAINABILITY PROOFS

In this section, we present the full proofs comparing the impact of personalization on prediction accuracy versus explanation quality, highlighting situations under which their effects diverge or align.

### C.1  PROOF FOR THEOREM 4.1

We provide the proof for theorem 4.1 for two metrics of explanation quality: sufficiency and incomprehensiveness, from Table 1. The proof for sufficiency is illustrated in Figure 5. The proof for incomprehensivess is illustrated in Figure 6

*Proof.* Let $\mathbf{X} = (\mathbf{X_1}, \mathbf{X_2})$ where $\mathbf{X_1}$ and $\mathbf{X_2}$ are independent and each follows $\mathrm{Unif}(-\frac{1}{2}, \frac{1}{2})$. Let us define one binary personal attribute $s \in \{0, 1\}$ as $\mathbf{S} = \mathbb{1}(\mathbf{X_1} + \mathbf{X_2} > 0)$ and assume that we seek to predict $\mathbf{Y} = \mathbf{S}$. Then, $h_0(x) = \mathbb{1}(\mathbf{X_1} + \mathbf{X_2} > 0)$ and $h_p(x) = \mathbb{1}(\mathbf{S} > 0)$ are the generic and personalized classifiers of interest.

**Prediction.**  Both classifiers achieve perfect accuracy. Therefore, $\mathrm{BoP}_P(h_0, h_p) = 0$.

In particular, they also achieve perfect accuracy when we restrict the input $\mathbf{X}$ to any subgroup, subgroup $s = 0$ or subgroup $s = 1$, such that:

$$\mathrm{G\text{-}BoP}_P(h_0, h_p, s = 0) = \mathrm{G\text{-}BoP}_P(h_0, h_p, s = 1) = \mathrm{BoP}_P(h_0, h_p) = 0,$$
$$\Rightarrow \gamma_P(h_0, h_p) = \min_{s \in \{0, 1\}} \mathrm{G\text{-}BoP}_P(h_0, h_p, s) = 0.$$

**Explanation (sufficiency).**  We now test sufficiency by evaluating the accuracy of classifiers using only the important feature.

For model $h_0$, its important feature set $J_0$ is either $\{\mathbf{X_1}\}$ or $\{\mathbf{X_2}\}$. Without loss of generality, let $J_0 = \{\mathbf{X_1}\}$. For the personalized model, $J_p = \{\mathbf{S}\}$.

For sufficiency, we compute:

$$\begin{aligned} \Pr(h_0(\mathbf{X}) \neq h_0(\mathbf{X_{J_0}})) &= \Pr(\mathbf{X_1} + \mathbf{X_2} \leq 0 | \mathbf{X_1} > 0)\Pr(\mathbf{X_1} > 0) \\ &\quad + \Pr(\mathbf{X_1} + \mathbf{X_2} > 0 | \mathbf{X_1} \leq 0)\Pr(\mathbf{X_1} \leq 0) \\ &= \frac{1}{4}, \end{aligned} \qquad (2)$$

where the computation per group also gives:

$$\Pr(h_0(\mathbf{X}) \neq h_0(\mathbf{X_{J_0}}) | s = 0) = \Pr(h_0(\mathbf{X}) \neq h_0(\mathbf{X_{J_0}}) | s = 1) = \frac{1}{4}.$$

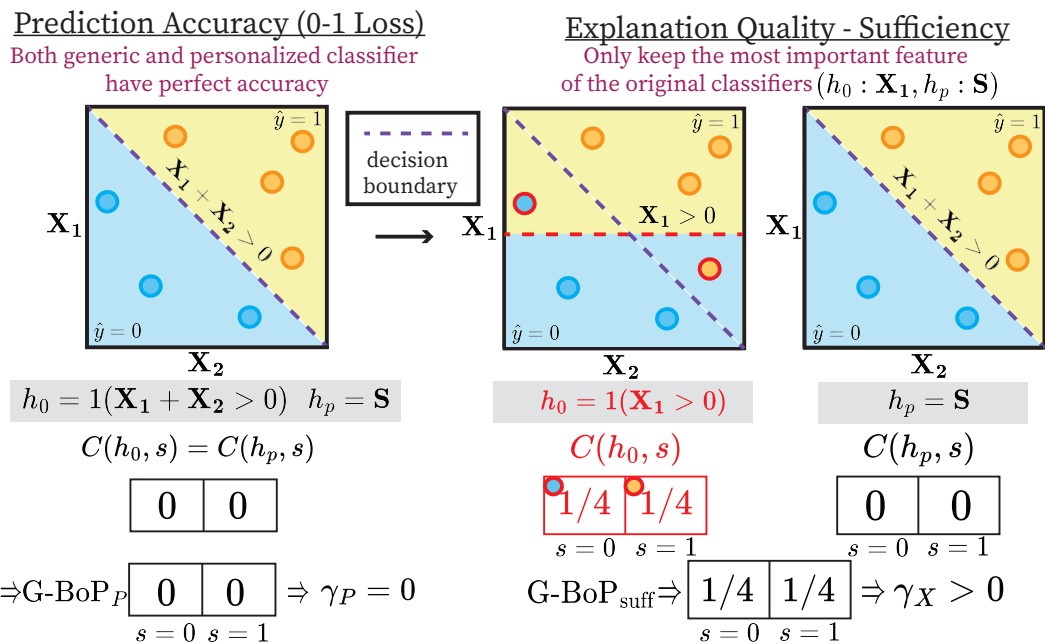

Figure 5: Comparing a generic model ($h_0$) and a personalized model ($h_p$) on prediction and explanation (sufficiency). Top-left: The generic model $h_0$ uses both $\mathbf{X_1}$ and $\mathbf{X_2}$ for predictions, with its decision boundary defined by $\mathbf{X_1} + \mathbf{X_2} > 0$. The personalized model, $h_p$, has access to the group attribute $\mathbf{S}$ (defined as $\mathbf{S} = \mathbb{1}(\mathbf{X_1} + \mathbf{X_2} > 0)$), and its prediction rule is to output $\mathbf{S}$. Bottom-left: Since both classifiers achieve perfect accuracy (on both groups $s = 0$ and $s = 1$), the Group Benefit of Personalization ($G - BoP_P$) is 0 on both groups, and thus: $\gamma_P = 0$. Top-right: In the sufficiency evaluation, where only the most important feature is kept, $h_p$ achieves perfect prediction since it relies solely on $\mathbf{S}$, reaching a sufficiency cost of 0 for each group. In contrast, $h_0$, using only $\mathbf{X_1}$, now makes prediction errors and has a worst sufficiency cost of $\frac{1}{4}$ for each group. Bottom-right: Since the personalized model has better sufficiency than the generic model, the G-BoP is positive and equal to $\frac{1}{4}$ for both groups, and hence $\gamma_x = \frac{1}{4} > 0$. Hence, personalization can enhance explainability even though prediction accuracy remains the same.

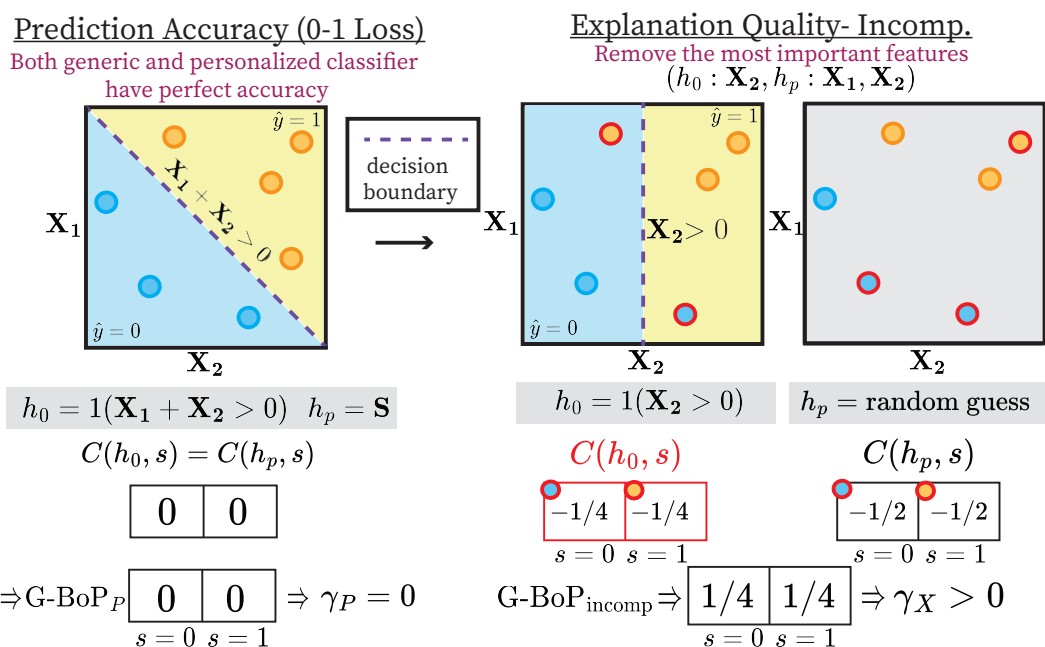

Figure 6: Comparing a generic model ($h_0$) and a personalized model ($h_p$) on prediction and explanation (incomprehensiveness). Both achieve perfect accuracy, but $h_p$ relies solely on $\mathbf{S} = 1(\mathbf{X_1} + \mathbf{X_2} > 0)$, yielding higher incomprehensiveness. Hence, personalization can improve explainability even when accuracy is unchanged: here, $\gamma_P = 0$ and $\gamma_X > 0$.

On the other hand, the sufficiency for $h_p$ is

$$\Pr(h_p(\mathbf{X}, \mathbf{S}) \neq h_p(\mathbf{X_{J_p}}, \mathbf{S_{J_p}})) = 0,$$

as $J_p = \{\mathbf{S}\}$ is sufficient to make a prediction for $h_p$. The computation per group also gives 0, since the model makes perfect predictions independently of the value taken by $\mathbf{S}$.

Thus, $\text{BoP}_X$ in terms of sufficiency is also $\frac{1}{4}$. Computing this quantity per group gives:

$$\text{G-BoP}_X(h_0, h_p, s = 0) = \text{G-BoP}_X(h_0, h_p, s = 1) = \frac{1}{4},$$

$$\Rightarrow \gamma_{\text{suff}}(h_0, h_p) = \min_{s \in \{0,1\}} \text{G-BoP}_X(h_0, h_p, s) = \frac{1}{4}. \tag{3}$$

**Explanation (incomprehensiveness)** Incomprehensiveness is the opposite of comprehensiveness. For clarity, we provide the computations for comprehensiveness first.

Comprehensiveness of $h_0$ is

$$
\begin{aligned}
\Pr(h_0(\mathbf{X}) \neq h_0(\mathbf{X}_{\backslash \mathbf{J_o}})) &= \Pr(\mathbf{X_1} + \mathbf{X_2} \leq 0 | \mathbf{X_2} > 0)\Pr(\mathbf{X_2} > 0) \\
&\quad + \Pr(\mathbf{X_1} + \mathbf{X_2} > 0 | \mathbf{X_2} \leq 0)\Pr(\mathbf{X_2} \leq 0) \\
&= \Pr(\mathbf{X_1} + \mathbf{X_2} \leq 0 | \mathbf{X_2} > 0) \cdot \frac{1}{2} + \Pr(\mathbf{X_1} + \mathbf{X_2} > 0 | \mathbf{X_2} \leq 0) \cdot \frac{1}{2} \\
&= \Pr(\mathbf{X_1} + \mathbf{X_2} \leq 0 | \mathbf{X_2} > 0) \quad \text{(due to symmetry of the distribution)} \\
&= \int_{x_2 > 0, x_1 + x_2 \leq 0} \Pr(x_1, x_2) dx_1 dx_2 / \Pr(\mathbf{X_2} > 0) \\
&= 2 \cdot \int_{x_2 = 0}^{\frac{1}{2}} \Pr(x_2) \int_{x_1 \leq -x_2} \Pr(x_1) dx_1 dx_2 \\
&= 2 \cdot \int_{x_2 = 0}^{\frac{1}{2}} \Pr(x_2)(-x_2 + \frac{1}{2}) dx_2 \\
&= 2 \cdot \left[ -\frac{1}{2} x_2^2 + \frac{1}{2} x_2 \right]_0^{\frac{1}{2}} \\
&= \frac{1}{4}.
\end{aligned}
\tag{4}
$$

Hence, incomprehensiveness of $h_0$ is $-\frac{1}{4}$.

Computing this quantity per group gives, by symmetry of the problem:

$$
\begin{aligned}
\Pr(h_0(\mathbf{X}) \neq h_0(\mathbf{X}_{\backslash \mathbf{J_o}}) \mid s = 0) &= \Pr(h_0(\mathbf{X}) \neq h_0(\mathbf{X}_{\backslash \mathbf{J_o}}) \mid s = 1) \\
&= \frac{1}{2} \Pr(h_0(\mathbf{X}) \neq h_0(\mathbf{X}_{\backslash \mathbf{J_o}})) \\
&= \frac{1}{4}.
\end{aligned}
\tag{5}
$$

Hence, incomprehensiveness per group is also $-\frac{1}{4}$.

For $h_p$, comprehensiveness is:

$$
\Pr(h_p(\mathbf{X}, \mathbf{S}) \neq h_p(\mathbf{X}_{\backslash \mathbf{J_p}}, \mathbf{S}_{\backslash \mathbf{J_p}})) = \frac{1}{2},
$$

as without $\mathbf{S}$, $h_p$ can only make a random guess. Hence, incomprehensiveness for each group is $-\frac{1}{2}$.

Computing this quantity per group also gives $\frac{1}{2}$ since $h_p$ makes a random guess independently of the subgroup considered:

$$
\begin{aligned}
\Pr(h_p(\mathbf{X}) \neq h_p(\mathbf{X}_{\backslash \mathbf{J_p}}) \mid s = 0) &= \Pr(h_p(\mathbf{X}) \neq h_p(\mathbf{X}_{\backslash \mathbf{J_p}}) \mid s = 1) \\
&= \Pr(h_p(\mathbf{X}) \neq h_p(\mathbf{X}_{\backslash \mathbf{J_p}})) \\
&= \frac{1}{2}.
\end{aligned}
\tag{6}
$$

while the incomprehensiveness per group is therefore $-\frac{1}{2}$.

Hence, $\text{BoP}_X$ in terms of incomprehensiveness is $\frac{1}{4}$.

Computing this quantity per group gives:

$$
\begin{aligned}
\text{G-BoP}_X(h_0, h_p, s = 0) = \text{G-BoP}_X(h_0, h_p, s = 1) &= \frac{1}{4}, \\
\Rightarrow \gamma_{\text{incomp}}(h_0, h_p) = \min_{s \in \{0,1\}} \text{G-BoP}_X(h_0, h_p, s) &= \frac{1}{4}.
\end{aligned}
\tag{7}
$$

$\square$

C.2   PROOF FOR THEOREM 4.2:

We provide the proof for Theorem 4.2, for explainability incomprehensiveness.

*Proof.* Let $\mathbf{X} = (\mathbf{X})$ where $\mathbf{X}$ follows $\mathrm{Unif}(-\frac{1}{2}, \frac{1}{2})$. Define one binary personal attribute $s \in \{0, 1\}$ as $\mathbf{S} = \mathbf{X}$ and assume that the true label that we seek to predict is $\mathbf{Y} = \mathbf{X} > 0$. We define the classifiers of interest as:

$$h_0(\mathbf{X}) = \mathbb{1}(\mathbf{X} > 0), h_p(\mathbf{X}, \mathbf{S}) = \frac{1}{2}(\mathbf{X} + \mathbf{S}).$$

**Prediction.**   Both $h_0$ and $h_p$ are perfectly aligned with the ground truth and yield $\hat{y} = \mathbf{Y}$. Therefore, they achieve perfect accuracy. In particular, they also achieve perfect accuracy when we restrict the input $\mathbf{X}$ to any subgroup, subgroup $s = 0$ or subgroup $s = 1$, such that:

$$\text{G-BoP}_P(h_0, h_p, s = 0) = \text{G-BoP}_P(h_0, h_p, s = 1) = \text{BoP}_P(h_0, h_p) = 0,$$
$$\Rightarrow \gamma_P(h_0, h_p) = \min_{s \in \{0,1\}} \text{G-BoP}_P(h_0, h_p, s) = 0.$$

Therefore, $\text{BoP}_P(h_0, h_p) = 0$.

**Explanation (sufficiency).**   For $h_0$, the most important feature is $\mathbf{X}$, while for $h_p$, the most important feature is $\mathbf{S}$.

We now test sufficiency by evaluating the accuracy of classifiers using only the important feature.

- For $h_0$, keeping $\mathbf{X}$ results in the original predictor. Therefore, prediction does not change at all and the feature is maximally sufficient for both groups ($\text{G-BoP}_{\text{suff}} = 0$ for $s = 0$ and $s = 1$, hence $\gamma_X = 0$.

- For $h_p$, keeping $\mathbf{S}$ does not change the prediction output because $\frac{1}{2}\mathbf{X} > 0 = \mathbf{X} > 0$. Therefore, prediction does not change at all and the feature is maximally sufficient for both groups ($\text{G-BoP}_{\text{suff}} = 0$ for $s = 0$ and $s = 1$, hence $\gamma_X = 0$

Therefore, $\text{BoP}_X = 0$ for sufficiency.

**Explanation (incomprehensiveness)**   In this setting, we evaluate incomprehensiveness by measuring the degradation in model predictions when the most important feature is removed.

- **Removing X from $h_0$:** For $h_0$, incomprehensiveness is:

$$\Pr(h_0(\mathbf{X}) \neq h_p())) = \frac{1}{2},$$

as without $\mathbf{X}$, $h_0$ can only make a random guess. Hence, incomprehensiveness for each group is $\frac{1}{2}$ and $\gamma_X = \frac{1}{2}$.

- **Removing S from $h_p$:** For $h_p$, we compute:

$$\Pr\big(h_p(\mathbf{X}, \mathbf{S}) \neq h_p(\mathbf{X})\big) = \Pr(\mathbf{X} + \mathbf{S} \leq 0 \mid \mathbf{X} > 0) \Pr(\mathbf{X} > 0)$$
$$+ \Pr(\mathbf{X} + \mathbf{S} > 0 \mid \mathbf{X} \leq 0) \Pr(\mathbf{X} \leq 0)$$
$$= \tfrac{1}{4}. \tag{8}$$

where the computation per group also gives:

$$\Pr(h_p(\mathbf{X}, \mathbf{S}) \neq h_p(\mathbf{X})|s = 0) = \Pr(h_p(\mathbf{X}, \mathbf{S}) \neq h_p(\mathbf{X})|s = 1) = \frac{1}{4}.$$

Hence, $\gamma_X = \frac{1}{4}$.

Therefore, $\text{BoP-X} = -\frac{1}{4}$.

$\square$

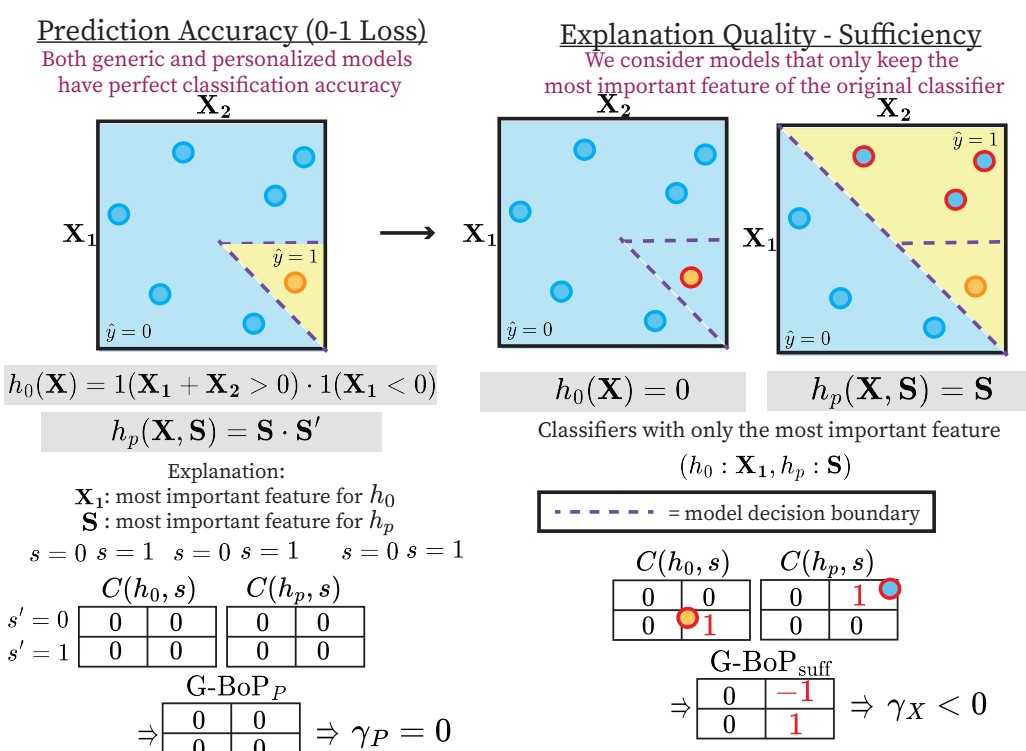

Figure 7: Comparing a generic model ($h_0$) and a personalized model ($h_p$) on prediction and explanation (sufficiency). Top-left: The generic model $h_0$ uses both $\mathbf{X_1}$ and $\mathbf{X_2}$ for predictions with its decision boundary defined by $1(\mathbf{X_1} + \mathbf{X_2} > 0) \cdot 1(\mathbf{X_1} < 0)$. The personalized model, $h_p$ instead predicts using the binary group attributes $s \in 0, 1$ and $s' \in 0, 1$ via the rule $s \cdot s'$. Bottom-left: Both classifiers achieve perfect accuracy across all four groups, hence $\gamma_P = 0$. Top-right: Sufficiency evaluation reveals a difference in explanation quality. For $h_0$, keeping only the top feature $\mathbf{X_1}$ results in a constant prediction $h_0(\mathbf{X_1}) = 0$, causing an error for the group $s = s' = 1$ (orange circle). For $h_p$, keeping only $\mathbf{S}$ yields $h_p(\mathbf{S}) = \mathbf{S}$, which fails to recover the true $\mathbf{Y}$ for the group $(s = 1, s' = 0)$ (blue circles). Bottom-right: Thus, the G-BoP is positive for $s = s' = 1$ but negative for $s = 1, s' = 0$, yielding $\gamma_X < 0$. This shows that even with identical predictive performance, the models rely on different features, and personalization can reduce sufficiency-based explainability for some groups.

## C.3 PROOF FOR THEOREM 4.3:

We provide the proof for Theorem 4.3, for two measures of explainability evaluation: sufficiency and incomprehensiveness, as illustrated in Figure 7 and Figure 8. Figure 7 illustrates the proof for sufficiency, where both generic $h_0$ and personalized $h_p$ models predict perfectly (left), yet only keeping the most important feature for each (right) shows that the personalized model is more explainable for the group ($s' = 1, s = 0$), and less explainable for group ($s' = 0, s = 1$). Figure 8 illustrates the proof for incomprehensiveness.

*Proof.* Let $\mathbf{X} = (\mathbf{X_1}, \mathbf{X_2})$ where $\mathbf{X_1}$ and $\mathbf{X_2}$ are independent and follow $\text{Unif}(-1, 1)$. Define two binary personal attributes $s \in \{0, 1\}$ and $s' \in \{0, 1\}$ such that the true label that we seek to predict is $\mathbf{Y} = \mathbf{S} \cdot \mathbf{S}'$. We define the classifiers of interest as:

$$h_0(\mathbf{X}) = \mathbb{1}(\mathbf{X_1} + \mathbf{X_2} > 0) \cdot \mathbb{1}(\mathbf{X_2} < 0), \quad h_p(\mathbf{X}, \mathbf{S}) = \mathbf{S} \cdot \mathbf{S}'.$$

**Prediction.** Both $h_0$ and $h_p$ are perfectly aligned with the ground truth and yield $\hat{y} = \mathbf{Y}$. Therefore, they achieve perfect accuracy. In particular, this holds for both values of $\mathbf{S}$ and $\mathbf{S}'$:

$$\text{G-BoP}_P$$

| $s'\backslash s$ | $s = 0$ | $s = 1$ |
|---|---|---|
| $s' = 0$ | 0 | 0 |
| $s' = 1$ | 0 | 0 |

Such that we get:

$$\gamma_P(h_0, h_p) = \min_{s,s' \in \{0,1\}} \text{G-BoP}_P(h_0, h_p, s) = 0.$$

**Explanation (sufficiency).** For $h_0$, the most important feature is $\mathbf{X_1}$, while for $h_p$, the most important feature is $\mathbf{S}$.

We now test sufficiency by evaluating the accuracy of classifiers using only the important feature.

- For $h_0$, keeping only $\mathbf{X_1}$ results in a constant predictor $h_0(\mathbf{X_1}) = 0$. This fails to recover $\hat{y}$ when $s = 1$ and $s' = 1$ (red orange dot), leading to an error for the subgroup $(s = 1, s' = 1)$, while the three other subgroups still enjoy perfect prediction.

- For $h_p$, keeping only $\mathbf{S}$ yields $h_p(\mathbf{S}) = \mathbf{S}$, which fails to recover $\hat{y}$ when $s = 1$ and $s' = 0$ (red blue circles) but still correctly predicts for the other three subgroups.

Combining per-group values gives: such that we get:

$$\text{G-BoP}_{\text{suff}}$$

| $s'\backslash s$ | $s = 0$ | $s = 1$ |
|---|---|---|
| $s' = 0$ | 0 | $-1$ |
| $s' = 1$ | 0 | 1 |

$$\gamma_X(h_0, h_p) = \min_{s \in \{0,1\}} \text{G-BoP}_{\text{suff}}(h_0, h_p, s) = -1.$$

**Explanation (incomprehensiveness)** In this setting, we evaluate incomprehensiveness by measuring the degradation in model predictions when the most important feature is removed.

The generic classifier is $h_0(\mathbf{X}) = \mathbb{1}(\mathbf{X_1} + \mathbf{X_2} > 0) \cdot \mathbb{1}(X_1 < 0)$ and the personalized classifier is $h_p(\mathbf{X}, \mathbf{S}) = \mathbf{S} \cdot \mathbf{S}'$. The most important feature for $h_0$ is $\mathbf{X_1}$ and for $h_p$ is $\mathbf{S}$.

- **Removing $\mathbf{X_1}$ from $h_0$:** Without $\mathbf{X_1}$, the classifier reduces to the constant function $h_0(\mathbf{X}_{\backslash \mathbf{X_1}}) = 0$. This leads to an incorrect prediction when $s = 1$ and $s' = 1$.

- **Removing $\mathbf{S}$ from $h_p$:** The personalized model becomes $h_p(\mathbf{X}, \mathbf{S}_{\backslash \mathbf{S}}) = \mathbf{S}'$, which ignores $\mathbf{S}$. This leads to an incorrect prediction when $s = 0$ and $s' = 1$, since the true label is $y = 0$ but $h_p = 1$.

All other combinations yield correct predictions even when the important feature is removed.

$$\text{G-BoP}_{\text{incomp}}$$

| $s'\backslash s$ | $s = 0$ | $s = 1$ |
|---|---|---|
| $s' = 0$ | 0 | 0 |
| $s' = 1$ | 1 | -1 |

This yields the minimum group benefit of personalization is:

$$\gamma_X^{\text{incomp}}(h_0, h_p) = \min_{s,s' \in \{0,1\}} \text{G-BoP}_{\text{incomp}}(h_0, h_p, s, s') = -1.$$

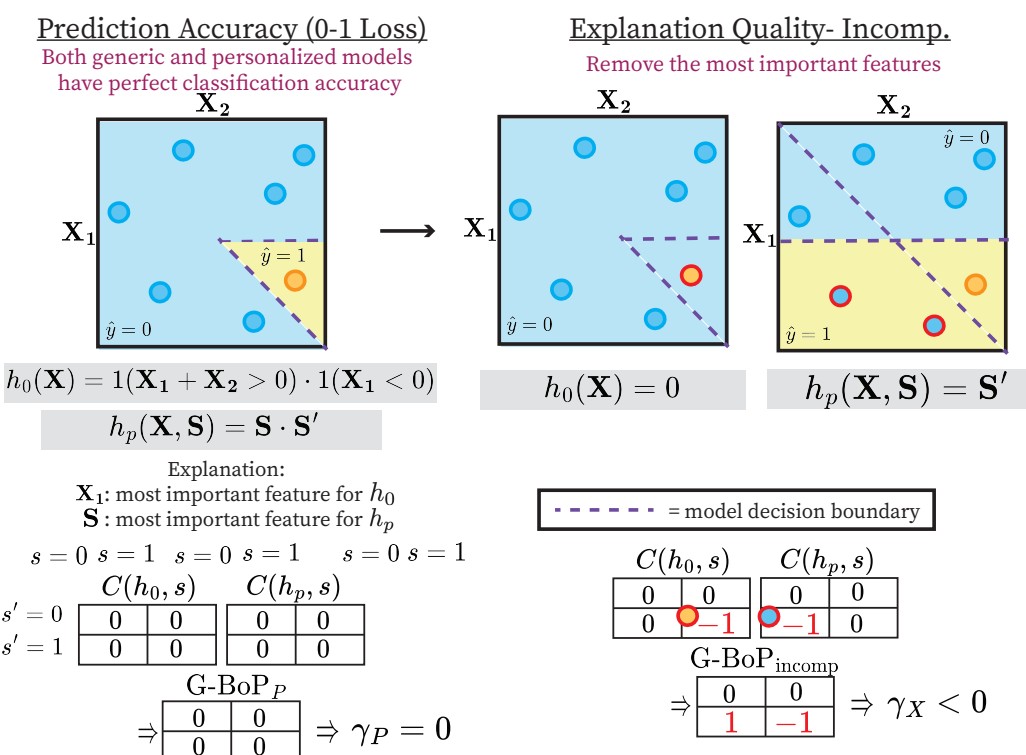

Figure 8: Comparing a generic model ($h_0$) and a personalized model ($h_p$) on prediction and explanation (incomprehensiveness). Both achieve perfect accuracy, but removing each most important features yields different prediction performances. We find that $\gamma_P = 0$ while $\gamma_X < 0$.

$\square$

### C.4 PROOF FOR THEOREM 4.4:

See Figure 9 for a visualization of Theorem 4.4 for a linear model with $h_0$ and $h_p$ Bayes optimal regressors.

*Proof.* A Bayes optimal regressor using a subset of variables from indices in $J \subseteq [1, \ldots, t + k]$ would be given as:

$$\hat{y} = h_J^*(\mathbf{X}_J, \mathbf{S}_J) = \sum_{\substack{j \in J, \\ j \leq t}} \alpha_j \mathbf{X}_j + \sum_{\substack{j \in J, \\ j \geq t+1}} \alpha_j \mathbf{S}_{j-t}, \tag{9}$$

where $h_J^*$ represents a Bayes optimal regressor for the given subset $J$, and $\mathbf{X}_J$ and $\mathbf{S}_J$ are sub-vectors of $\mathbf{X}$ and $\mathbf{S}$, using the indices in $J$.

In what follows, we denote $\backslash J$ as a shorthand notation for $[1, \ldots t + k] \backslash J$.

From equation 9 and the definition of the true response $\mathbf{Y} = \sum_{j \leq t} \alpha_j \mathbf{X}_j + \sum_{j \geq t+1} \alpha_j \mathbf{S}_{j-t}, +\epsilon$ we obtain:

$$\text{MSE}(h_0) = \sum_{j=t+1}^{t+k} \alpha_j^2 \text{Var}(\mathbf{S}_{t+j}) + \text{Var}(\epsilon), \tag{10}$$

$$\text{MSE}(h_p) = \text{Var}(\epsilon). \tag{11}$$

We define $J_0$ and $J_p$ as a set of important features for $h_0$ and $h_p$. Note that $J_0$ and $J_p$ are the same across all samples for the additive model. Then, the sufficiency of the explanation for $h_0$ and $h_p$ is

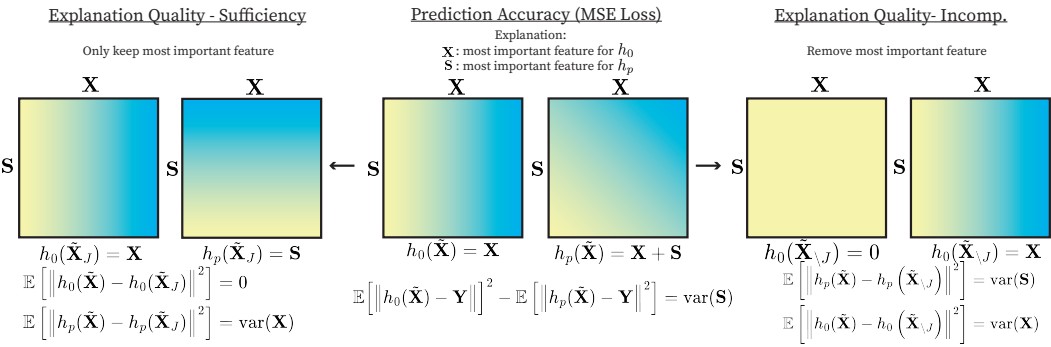

Figure 9: For a linear model, absence of benefit in explanation quality means that there is also an absence of benefit in prediction accuracy, as illustrated here (see Theorem 4.4). We consider a linear model $\mathbf{Y} = \mathbf{X} + \mathbf{S} + \epsilon$, with $h_0$ and $h_p$ Bayes optimal regressors. In this example, absence of benefit of personalization for the explanation quality, BoP-X$^{\text{suff}} = 0$ evaluated in terms of sufficiency (left column) means: $\mathbb{E}[\|h_0(\tilde{\mathbf{X}}) - h_0(\tilde{\mathbf{X}}_{\mathbf{J}})\|^2] = \mathbb{E}[\|h_p(\tilde{\mathbf{X}}) - h_p(\tilde{\mathbf{X}}_{\mathbf{J}})\|^2] \Rightarrow \text{var}(\mathbf{X}) = 0$. Then, absence of benefit of personalization for the explanation quality, BoP-X$^{\text{comp}} = 0$ evaluated in terms of comprehensiveness (right column) means: $\mathbb{E}[\|h_0(\tilde{\mathbf{X}}) - h_0(\tilde{\mathbf{X}}_{\setminus \mathbf{J}})\|^2] = \mathbb{E}[\|h_0(\tilde{\mathbf{X}}) - h_0(\tilde{\mathbf{X}}_{\setminus \mathbf{J}})\|^2] \Rightarrow \text{var}(\mathbf{S}) = \text{var}(\mathbf{X}) \Rightarrow \text{var}(\mathbf{S}) = 0$. This allows us to conclude that, in terms of prediction accuracy (middle column): $\text{MSE}_0 = \text{MSE}_p$ and hence there is also no benefit of personalization in prediction :BoP-P $= 0$.

written as:

$$\mathbb{E}[\|h_0(\tilde{\mathbf{X}}) - h_0(\tilde{\mathbf{X}}_{\mathbf{J_o}})\|^2] = \sum_{\substack{j \in \setminus J_0, \\ j \leq t}} \alpha_j^2 \text{Var}(\mathbf{X_t}) \tag{12}$$

$$\mathbb{E}[\|h_p(\tilde{\mathbf{X}}) - h_p(\tilde{\mathbf{X}}_{\mathbf{J_p}})\|^2] = \sum_{\substack{j \in \setminus J_p, \\ j \leq t}} \alpha_j^2 \text{Var}(\mathbf{X_t}) + \sum_{\substack{j \in \setminus J_p, \\ j \geq t+1}} \alpha_j^2 \text{Var}(\mathbf{S_{j-t}}). \tag{13}$$

Similarly, the comprehensiveness of the explanation for $h_0$ and $h_p$ is written as:

$$\mathbb{E}[\|h_0(\tilde{\mathbf{X}}) - h_0(\tilde{\mathbf{X}}_{\setminus J_0})\|^2] = \sum_{\substack{j \in J_0, \\ j \leq t}} \alpha_j^2 \text{Var}(\mathbf{X}_t) \tag{14}$$

$$\mathbb{E}[\|h_p(\tilde{\mathbf{X}}) - h_p(\tilde{\mathbf{X}}_{\setminus J_p})\|^2] = \sum_{\substack{j \in J_p, \\ j \leq t}} \alpha_j^2 \text{Var}(\mathbf{X}_t) + \sum_{\substack{j \in J_p, \\ j \geq t+1}} \alpha_j^2 \text{Var}(\mathbf{S}_{j-t}). \tag{15}$$

Then, our assumption of BoP-X $= 0$ for sufficiency becomes:

$$\mathbb{E}[\|h_0(\tilde{\mathbf{X}}) - h_0(\tilde{\mathbf{X}}_{J_0})\|^2] = \mathbb{E}[\|h_p(\tilde{\mathbf{X}}) - h_p(\tilde{\mathbf{X}}_{J_p})\|^2] \tag{16}$$

$$\Rightarrow \sum_{\substack{j \in \setminus J_0, \\ j \leq t}} \alpha_j^2 \text{Var}(\mathbf{X}_t) = \sum_{\substack{j \in \setminus J_p, \\ j \leq t}} \alpha_j^2 \text{Var}(\mathbf{X}_t) + \sum_{\substack{j \in \setminus J_p, \\ j \geq t+1}} \alpha_j^2 \text{Var}(\mathbf{S}_{j-t}) \tag{17}$$

Similarly, our assumption of BoP-X $= 0$ for comprehensiveness becomes:

$$\mathbb{E}[\|h_0(\tilde{\mathbf{X}}) - h_0(\tilde{\mathbf{X}}_{\setminus J_0})\|^2] = \mathbb{E}[\|h_p(\tilde{\mathbf{X}}) - h_p(\tilde{\mathbf{X}}_{\setminus J_p})\|^2] \tag{18}$$

$$\Rightarrow \sum_{\substack{j \in J_0, \\ j \leq t}} \alpha_j^2 \text{Var}(\mathbf{X}_t) = \sum_{\substack{j \in J_p, \\ j \leq t}} \alpha_j^2 \text{Var}(\mathbf{X}_t) + \sum_{\substack{j \in J_p, \\ j \geq t+1}} \alpha_j^2 \text{Var}(\mathbf{S}_{j-t}). \tag{19}$$

Summing both equations:

$$
\begin{aligned}
\sum_{\substack{j \in \backslash J_0 \\ j \leq t}} \alpha_j^2 \operatorname{Var}(\mathbf{X}_t) + \sum_{\substack{j \in J_0 \\ j \leq t}} \alpha_j^2 \operatorname{Var}(\mathbf{X}_t) &= \sum_{\substack{j \in \backslash J_p \\ j \leq t}} \alpha_j^2 \operatorname{Var}(\mathbf{X}_t) + \sum_{\substack{j \in \backslash J_p \\ j \geq t+1}} \alpha_j^2 \operatorname{Var}(\mathbf{S}_{j-t}) \\
&\quad + \sum_{\substack{j \in J_p \\ j \leq t}} \alpha_j^2 \operatorname{Var}(\mathbf{X}_t) + \sum_{\substack{j \in J_p \\ j \geq t+1}} \alpha_j^2 \operatorname{Var}(\mathbf{S}_{j-t}) \\
&\Rightarrow \operatorname{Var}(\mathbf{X}) = \operatorname{Var}(\mathbf{X}) + \operatorname{Var}(\mathbf{S}) \\
&\Rightarrow \operatorname{Var}(\mathbf{S}) = 0.
\end{aligned}
\tag{20}
$$

Since $\operatorname{Var}(\mathbf{S}) = 0$, we have that $\operatorname{MSE}(h_0) = \operatorname{MSE}(h_p)$ and thus: BoP-P $= 0$ which concludes the proof.

We can make the same claim with similar logic for a classifier where $\mathbf{Y}$ is given as:

$$
\mathbf{Y} = \mathbb{1}(\alpha_1 \mathbf{X}_1 + \cdots \alpha_t \mathbf{X}_t + \alpha_{t+1} \mathbf{S}_1 + \cdots + \alpha_{t+k} \mathbf{S}_k + \epsilon > 0).
\tag{21}
$$

The derivations above are made at the population level, i.e., without distinguishing subgroups in the data. However, the reasoning also applies for subgroups, where we define subgroups to be defined by $\mathbb{1}(\mathbf{S} \geq 0)$ taking values in $\{0, 1\}$. In other words, if G-BoP$_{\text{suff}}(h_0, h_p, s) = 0$ and G-BoP$_{\text{incomp}}(h_0, h_p, s) = 0$ then G-BoP$_P(h_0, h_p, s) = 0$ for any $s \in \{0, 1\}$. However, we note that we can only make a statement on $\gamma(h_0, h_p)$ (prediction accuracy) for the case where $\gamma_{\text{sufficiency}}(h_0, h_p) = 0$ and $\gamma_{\text{incomprehensiveness}}(h_0, h_p) = 0$ if the following is true: the group realizing the minima in the three $\gamma$'s is the same group. $\qquad \square$

## D  PROOF OF THEOREMS ON LOWER BOUNDS FOR THE PROBABILITY OF ERROR

As in (Monteiro Paes et al., 2022), we will prove every theorem for the flipped hypothesis test defined as:

$$
\begin{aligned}
H_0: & \quad \gamma(h_0, h_p; \mathcal{D}) \leq \epsilon \quad \Leftrightarrow \quad \text{Personalized } h_p \text{ performs worst: yields } \epsilon < 0 \text{ disadvantage} \\
H_1: & \quad \gamma(h_0, h_p; \mathcal{D}) \geq 0 \quad \Leftrightarrow \quad \text{Personalized } h_p \text{ performs at least as good as generic } h_0.
\end{aligned}
$$

where we emphasize that $\epsilon < 0$.

As shown in (Monteiro Paes et al., 2022), proving the bound for the original hypothesis test is equivalent to proving the bound for the flipped hypothesis test, since estimating $\gamma$ is as hard as estimating $-\gamma$. In every section that follows, $H_0, H_1$ refer to the flipped hypothesis test.

Here, we first prove a proposition that is valid for all of the cases that we consider in the next sections.

**Proposition D.1.** *Consider $H_0$ the set of distributions of data, for which the generic model $h_0$ performs better, i.e., the true $\gamma$ is such that $\gamma(h_0, h_p, \mathcal{D}) \leq \epsilon$, and $H_1$ the set of distributions of data points for which the personalized model performs better, i.e., the true $\gamma$ is such that $\gamma(h_0, h_p, \mathcal{D}) \geq 0$. Denote $H_0', H_1'$ the set of distributions that are mixtures of components in $H_0$ and $H_1$ respectively. Consider a decision rule $\Psi$ that represents any hypothesis test. We have the following bound on the probability of error $P_e$:*

$$
\min_{\Psi} \max_{\substack{P_0 \in H_0' \\ P_1 \in H_1'}} P_e \geq \frac{1}{2} \left(1 - TV(P \parallel Q)\right),
$$

*for any $P \in H_0'$ and any $Q \in H_1'$. Here $TV$ refers to the total variation between probability distributions $P$ and $Q$.*

*Proof.* Consider $h_0$ and $h_p$ fixed. Take one decision rule $\Psi$ that represents any hypothesis test. Consider a dataset such that $H_0$ is true, i.e., $\mathcal{D} \sim P_0'$ and a dataset such that $H_1$ is true, i.e., $\mathcal{D} \sim P_1'$.

It might seem counterintuitive to use two datasets to compute the same quantity $P_e$, i.e., one dataset to compute the first term in $P_e$, and one dataset to compute the second term in $P_e$. However, this

is just a reflection of the fact that $P_e$ is a composite measure, with its two terms coming from two different settings: $H_0$ true or $H_0$ false, which are disjoint events. In the same way that $H_0$ cannot be simultaneously true and false, each term in $P_e$ considers one or the other case, and we use one or the other dataset.

We have:

$$P_e = \frac{1}{2} \left( \Pr(\text{Rejecting } H_0 | H_0 \text{ true}) + \Pr(\text{Failing to reject } H_0 | H_1 \text{ true}) \right)$$

$$= \frac{1}{2} \left( \Pr(\Psi(h_0, h_p, \mathcal{D}, \epsilon) = 1 | \mathcal{D} \sim P_0) + \Pr(\Psi(h_0, h_p, \mathcal{D}, \epsilon) = 0 | \mathcal{D} \sim P_1) \right)$$

$$= \frac{1}{2} \left( \Pr(\Psi(\mathcal{D}) = 1 | \mathcal{D} \sim P_0) + \Pr(\Psi(\mathcal{D}) = 0 | \mathcal{D} \sim P_1) \right) \text{ simplifying notations}$$

$$= \frac{1}{2} \left( 1 - \Pr(\Psi(\mathcal{D}) = 0 | \mathcal{D} \sim P_0) + \Pr(\Psi(\mathcal{D}) = 0 | \mathcal{D} \sim P_1) \right) \text{ complementary event}$$

$$= \frac{1}{2} \left( 1 - P_0(E_\Psi) + P_1(E_\Psi) \right) \text{ writing } E_\Psi \text{ the event } \Psi(\mathcal{D}) = 0$$

$$= \frac{1}{2} \left( 1 - (P_0(E_\Psi) - P_1(E_\Psi)) \right).$$

Now, we show that this equality holds when considering mixtures of components in $H_0$ and mixtures of components in $H_1$.

Consider a dataset distributed as $P_0 \in H_0'$ that is a mixture $P_0 = \sum_{j=1}^{J_0} w_0^{(j)} P_0^{(j)}$, where $w_0^{(j)} \geq 0$ and $\sum_{j=1}^{J_0} w_0^{(j)} = 1$, such that, for each component $P_0^{(j)}$ of the mixture, $H_0$ is true.

Consider another dataset distributed as $P_1 \in H_1'$ that is a mixture $P_1 = \sum_{j=1}^{J_1} w_1^{(j)} P_1^{(j)}$, where $w_1^{(j)} \geq 0$ and $\sum_{j=1}^{J_1} w_1^{(j)} = 1$, such that, for each component $P_1^{(j)}$ of the mixture, $H_1$ is true.

For every $j, j'$, since $P_0^{(j)} \in H_0$ and $P_1^{(j')} \in H_1$ we have:

$$P_e = \frac{1}{2} \left( 1 - (P_0^{(j)}(E_\Psi) - P_1^{(j')}(E_\Psi)) \right).$$

Since $\sum_{j=1}^{J_0} w_0^{(j)} = 1$ and $\sum_{j'=1}^{J_1} w_1^{(j')} = 1$ we have:

$$P_e = \left( \sum_{j=1}^{J_0} w_0^{(j)} \right) \left( \sum_{j'=1}^{J_1} w_1^{(j')} \right) P_e$$

$$= \left( \sum_{j=1}^{J_0} w_0^{(j)} \right) \left( \sum_{j'=1}^{J_1} w_1^{(j')} P_e \right)$$

$$= \sum_{j=1}^{J_0} \sum_{j'=1}^{J_1} w_0^{(j)} w_1^{(j')} P_e$$

$$= \sum_{j=1}^{J_0} \sum_{j'=1}^{J_1} w_0^{(j)} w_1^{(j')} \frac{1}{2} \left( 1 - (P_0^{(j)}(E_\Psi) - P_1^{(j')}(E_\Psi)) \right) \text{ (choosing to replace } P_e \text{ its expression involving } j, j')$$

$$= \frac{1}{2} \sum_{j=1}^{J_0} \sum_{j'=1}^{J_1} w_0^{(j)} \left( w_1^{(j')} - w_1^{(j')} P_0^{(j)}(E_\Psi) + w_1^{(j')} P_1^{(j')}(E_\Psi) \right)$$

$$= \frac{1}{2} \sum_{j=1}^{J_0} w_0^{(j)} \sum_{j'=1}^{J_1} \left( w_1^{(j')} - w_1^{(j')} P_0^{(j)}(E_\Psi) + w_1^{(j')} P_1^{(j')}(E_\Psi) \right)$$

$$= \frac{1}{2} \sum_{j=1}^{J_0} w_0^{(j)} \left( \sum_{j'=1}^{J_1} w_1^{(j')} - \sum_{j'=1}^{J_1} w_1^{(j')} P_0^{(j)}(E_\Psi) + \sum_{j'=1}^{J_1} w_1^{(j')} P_1^{(j')}(E_\Psi) \right)$$

$$= \frac{1}{2} \sum_{j=1}^{J_0} w_0^{(j)} \left( 1 - P_0^{(j)}(E_\Psi) + \sum_{j'=1}^{J_1} w_1^{(j')} P_1^{(j')}(E_\Psi) \right) \quad \text{(using } \sum_{j'=1}^{J_1} w_1^{(j')} = 1)$$

$$= \frac{1}{2} \sum_{j=1}^{J_0} w_0^{(j)} \left( 1 - P_0^{(j)}(E_\Psi) + P_1(E_\Psi) \right) \quad \text{(definition of } P_1)$$

$$= \frac{1}{2} \left( \sum_{j=1}^{J_0} w_0^{(j)} - \sum_{j=1}^{J_0} w_0^{(j)} P_0^{(j)}(E_\Psi) + \sum_{j=1}^{J_0} w_0^{(j)} P_1(E_\Psi) \right)$$

$$= \frac{1}{2} \left( 1 - \sum_{j=1}^{J_0} w_0^{(j)} P_0^{(j)}(E_\Psi) + P_1(E_\Psi) \right) \quad \text{(using } \sum_{j=1}^{J_0} w_0^{(j)} = 1)$$

$$= \frac{1}{2} \left( 1 - P_0(E_\Psi) + P_1(E_\Psi) \right) \quad \text{(definition of } P_0)$$

Hence, the equality holds for mixtures of components in $H_0, H_1$ respectively. We denote $H_0', H_1'$ the respective sets of such mixtures.

Now, we will bound this quantity:

$$\min_{\substack{\Psi}} \max_{\substack{P_0 \in H_0' \\ P_1 \in H_1'}} P_e = \min_{\substack{\Psi}} \max_{\substack{P_0 \in H_0' \\ P_1 \in H_1'}} \frac{1}{2} \left( 1 - (P_0(E_\Psi) - P_1(E_\Psi)) \right)$$

$$\geq \max_{\substack{P_0 \in H_0' \\ P_1 \in H_1'}} \min_{\substack{\Psi}} \left[ \frac{1}{2} \left( 1 - (P_0(E_\Psi) - P_1(E_\Psi)) \right) \right] \quad \text{using minmax inequality}$$

$$= \max_{\substack{P_0 \in H_0' \\ P_1 \in H_1'}} \left[ \frac{1}{2} \left( 1 - \max_{\Psi}(P_0(E_\Psi) - P_1(E_\Psi)) \right) \right] \quad \text{to minimize over } \Psi, \text{ we maximize } (P_0(E_\Psi) - P_1(E_\Psi))$$

$$\geq \max_{\substack{P_0 \in H_0' \\ P_1 \in H_1'}} \left[ \frac{1}{2} \left( 1 - \max_{\text{events } A} (P_0(A) - P_1(A)) \right) \right] \quad \text{because the max is now over all possible events } A$$

The maximization is broadened to consider all possible events $A$. This increases the set over which the maximum is taken. Because $\Psi$ is only a subset of all possible events, maximizing over all events $A$ (which includes $\Psi$) will result in a value that is at least as large as the maximum over $\Psi$. In other words, extending the set of possible events can only increase the maximum.

$$= \max_{\substack{P_0 \in H_0' \\ P_1 \in H_1'}} \left[ \frac{1}{2} \left( 1 - TV(P_0 \parallel P_1) \right) \right] \quad \text{by definition of the total variation (TV)}$$

$$= \frac{1}{2} \left( 1 - \min_{\substack{P_0 \in H_0' \\ P_1 \in H_1'}} TV(P_0 \parallel P_1) \right)$$

$$\geq \frac{1}{2} \left( 1 - TV(P \parallel Q) \right) \quad \text{for any } P \in H_0' \text{ and } Q \in H_1'.$$

This is true because the total variation distance $TV(P \parallel Q)$ for any particular pair $P$ and $Q$ cannot be smaller than the minimum total variation distance across all pairs. We recall that, by definition, the total variation of two probability distributions $P, Q$ is the largest possible difference between the probabilities that the two probability distributions can assign to the same event $A$.

Since the total variation distance $TV$ gives values in the interval $[0, 1]$, the lower bound cannot exceed $1/2$. This is expected. In a binary hypothesis test with equal priors, a decision rule corresponding to random guessing would yield $P_e = 1/2$. Consequently, the optimal probability of error (the minimum over all decision rules $\Psi$) must lie in the interval $[0, 1/2]$. $\qquad \square$

We note that, through this derivation, we have recovered Le Cam's standard result (see (Le Cam, 2012; Yu, 1997) Lemma 1]).

Next, we prove a lemma that will be useful for the follow-up proofs.

**Lemma D.2.** *Consider a random variable $a$ such that $\mathbb{E}[a] = 1$. Then:*

$$\mathbb{E}[(a-1)^2] = \mathbb{E}[a^2] - 1 \tag{22}$$

*Proof.* We have that:

$$
\begin{aligned}
\mathbb{E}[(a-1)^2] &= \mathbb{E}[a^2 - 2a + 1] \\
&= \mathbb{E}[a^2] - 2\mathbb{E}[a] + 1 \text{ (linearity of the expectation)} \\
&= \mathbb{E}[a^2] - 2 + 1 (\mathbb{E}[a] = 1 \text{ by assumption}) \\
&= \mathbb{E}[a^2] - 1.
\end{aligned}
$$

$\square$

### D.1 PROOF FOR THEOREM 5.1: ANY PROBABILITY DISTRIBUTION AND ANY NUMBER OF SAMPLES IN EACH GROUP

Below, we find the lower bound for the probability of error for any probability distribution of the BoP, and any number of samples per group.

**Theorem** (Theorem 5.1, restated). *The lower bound writes:*

$$\min_{\Psi} \max_{\substack{P_0 \in H_0' \\ P_1 \in H_1'}} P_e \geq \frac{1}{2} \left( 1 - \frac{1}{2\sqrt{d}} \left[ \frac{1}{d} \sum_{j=1}^{d} \mathbb{E}_{p^\epsilon} \left[ \frac{p^\epsilon(B)}{p(B)} \right]^{m_j} - 1 \right]^{\frac{1}{2}} \right) \tag{23}$$

*where $P_0$ is a mixture of distributions of data, for which the generic model $h_0$ performs better, i.e., the true $\gamma$ is such that $\gamma(h_0, h_p, \mathcal{D}) < 0$, and $P_1$ is a mixture of distribution of data points for which the personalized model performs better, i.e., the true $\gamma$ is such that $\gamma(h_0, h_p, \mathcal{D}) \geq \epsilon$. Dataset $\mathcal{D}$ is drawn from an unknown distribution and has $d$ groups where $d = 2^k$, with each group having $m_j$ samples.*

*Proof.* By Proposition D.1, we have that:

$$\min_{\Psi} \max_{\substack{P_0 \in H_0' \\ P_1 \in H_1'}} P_e \geq \frac{1}{2} \left( 1 - TV(P \parallel Q) \right)$$

for any $P \in H_0'$ and any $Q \in H_1'$. We will design two probability distributions $P, Q$ defined on the $N$ data points $(\mathbf{X_1}, \mathbf{S_1}, \mathbf{Y_1}), ..., (\mathbf{X_N}, \mathbf{S_N}, \mathbf{Y_N})$ of the dataset $\mathcal{D}$ to compute an interesting right hand side term.

An "interesting" right hand side term is a term that makes the lower bound as tight as possible, i.e., it relies on distributions $P, Q$ for which $TV(P \parallel Q)$ is small, i.e., probability distributions that are similar. To achieve this, we will first design the distribution $Q \in H_1'$, and then propose $P$ as a very small modification of $Q$, just enough to allow it to verify $P \in H_0'$.

Mathematically, $P, Q$ are distributions on the dataset $\mathcal{D}$, i.e., on $N$ i.i.d. realizations of the random variables $\mathbf{X}, \mathbf{S}, \mathbf{Y}$. Thus, we wish to design probability distributions on $(\mathbf{X_1}, \mathbf{S_1}, \mathbf{Y_1}), ..., (\mathbf{X_N}, \mathbf{S_N}, \mathbf{Y_N})$.

However, we note that the dataset distribution is only meaningful in terms of how each triplet $(\mathbf{X_i}, \mathbf{S_i}, \mathbf{Y_i})$ impacts the value of the individual BOP $\mathbf{B_i}$. Indeed, we recall that $\mathbf{B_i}$ is a function of the data point $\mathbf{Z_i} = (\mathbf{X_i}, \mathbf{S_i}, \mathbf{Y_i})$, which we denote as $f$ such that $\mathbf{B_i} = f(\mathbf{Z_i}) = \text{cost}(h_0, \tilde{\mathbf{X}}_i, \mathbf{Y}_i) - \text{cost}(h_p, \tilde{\mathbf{X}}_i, \mathbf{Y}_i)$. Hence, any probability distribution on $\mathbf{Z_i}$ will yield a probability distribution on $\mathbf{B_i}$ and any distribution on the dataset $\mathbf{Z_1}, ..., \mathbf{Z_N}$ will yield a distribution on $\mathbf{B_1}, ..., \mathbf{B_N}$.

Conversely, let be given $\tilde{P}(b_1, ..., b_N) = \Pi_{i=1}^{N} \tilde{P}_i(b_i)$ a distribution on $\mathbf{B_1}, ..., \mathbf{B_N}$ defined by $N$ independent distributions $\tilde{P}_i$ for $i = 1, .., N$, such that the support of each $\tilde{P}_i$ is restricted to the image

of $f$. We propose to build a probability distribution $P(z_1, ..., z_N) = \Pi_{i=1}^N P_i(z_i)$ on $\mathbf{Z_1}, ..., \mathbf{Z_N}$ that will ensure that $f(\mathbf{Z_1}), ..., f(\mathbf{Z_N})$ is distributed as $\tilde{P}$.

First, for each $P_i$ we restrict $P_i$ so that, for every value $b_i$ that $\mathbf{B_i}$ can take according to $\tilde{P}_i$, there exists a unique $z_i$ with positive density, concentrated as a Dirac at $z_i$, such that we have $f(z_i) = b_i$. Existence is guaranteed since $\mathbf{B_i}$ takes values in the image of $f$. Uniqueness is guaranteed because we can assign 0 mass to the potential non-unique values. Equivalently, $f$ is a bijection from $\text{supp}(P_i)$ to the set of values taken by $\mathbf{B}_i$ for each $i$. We note that the construction of $P_i$ (as a collection of Dirac measures centered at points $z_i$ in the domain of $f$) makes $\text{supp}(P_i)$ a discrete set of points.

By this construction, we have restricted the support such that $f$ acts as a bijection between $\text{supp}(P_i) \subset \mathcal{Z}$ and $\text{supp}(\tilde{P}_i) \subset \mathcal{B}$ for all $i$ (and analogously for $Q_i$ and $\tilde{Q}_i$). Let $g = f^{-1}$ denote the inverse mapping defined on this restricted domain.

We define the distributions $P$ and $Q$ over the full dataset as the push-forward measures of $\tilde{P}$ and $\tilde{Q}$ under the component-wise mapping $g$, meaning $P = g_\# \tilde{P}$ and $Q = g_\# \tilde{Q}$. By the Data Processing Inequality (DPI), applying any measurable function to a pair of distributions cannot increase their Total Variation distance. Applying the forward mapping $f$ to the distributions $P$ and $Q$ yields: $TV(\tilde{P} \parallel \tilde{Q}) \leq TV(P \parallel Q)$. Conversely, applying the inverse mapping $g = f^{-1}$ to the distributions $\tilde{P}$ and $\tilde{Q}$ yields: $TV(P \parallel Q) \leq TV(\tilde{P} \parallel \tilde{Q})$. Thus, the Total Variation distance is preserved: $TV(P \parallel Q) = TV(\tilde{P} \parallel \tilde{Q})$.

Thus, we design probability distributions $\tilde{P}, \tilde{Q}$ on $N$ i.i.d. realizations of the "auxiliary random" variable $\mathbf{B}$, with values in $\mathbb{R}$, defined as:

$$\mathbf{B} = \ell(h_0(\mathbf{X}), \mathbf{Y}) - \ell(h_p(\mathbf{X}, \mathbf{S}), \mathbf{Y}). \tag{24}$$

Intuitively, $\mathbf{B}_i$ represents how much the triplet $(\mathbf{X_i}, \mathbf{S_i}, \mathbf{Y_i})$ contributes to the value of the BOP. $b_i > 0$ means that the personalized model provided a better prediction than the generic model on the triplet $(x_i, s_i, y_i)$ corresponding to the data point $i$. In what follows, we drop the "tilde" for notational convenience, such that we use $P, Q$ to refer to $\tilde{P}, \tilde{Q}$.

Consider the event $b = (b_1, ..., b_N) \in \mathbb{R}^N$ of $N$ realizations of $\mathbf{B}$. For simplicity in our computations, we divide this event into the $d$ groups, i.e., we write instead: $b_j = (b_j^{(1)}, ..., b_j^{(m)})$, since each group $j$ has $m_j$ samples. Thus, we have: $b = \{b_j^{(k)}\}_{j=1...d, k=1...m}$ indexed by $j, k$ where $j = 1...d$ is the group in which this element is, and $k = 1...m_j$ is the index of the element in that group.

**Design $Q$.** Next, we continue designing a distribution $Q$ (since we have justified that we can define them on $\mathbf{B}$) on this set of events that will (barely) verify $H_1$, i.e., such that the expectation of $B$ according to $Q$ will give $\gamma = 0$. We recall that $\gamma = 0$ means that the minimum benefit across groups is 0, implying that there might be some groups that have a $> 0$ benefit.

Given $p$ as a distribution with mean $\mu = 0$, we propose the following distribution for $Q$

$$Q_j(b_j) = \prod_{k=1}^m p(b_j^{(k)}), \text{ for every group } j = 1....d$$

$$Q(b) = \prod_{j=1}^d Q_j(b_j).$$

We verify that we have designed $Q$ correctly, i.e., we verify that $Q \in H_1'$. When the dataset is distributed according to $Q$, we have:

$\gamma = \min_{s \in S} C_s(h_0, s) - C_s(h_p, s)$

$\quad = \min_{s \in S} \mathbb{E}_Q[\ell(h_0(\mathbf{X}), \mathbf{Y}) \mid \mathbf{S} = s] - \mathbb{E}_Q[\ell(h_p(\mathbf{X}), \mathbf{Y}) \mid \mathbf{S} = s]$ (by definition of group cost)

$\quad = \min_{s \in S} \mathbb{E}_Q[\ell(h_0(\mathbf{X}), \mathbf{Y}) - \ell(h_p(\mathbf{X}), \mathbf{Y}) \mid \mathbf{S} = s]$ (by linearity of expectation)

$\quad = \min_{s \in S} \mathbb{E}_Q[B \mid \mathbf{S} = s]$ (by definition of random variable $\mathbf{B}$)

$$= \min_{s \in S} 0 \text{ (by definition of the probability distribution on } \mathbf{B})$$
$$= 0.$$

Thus, we find that $\gamma = 0$, which means that $\gamma \geq 0$, i.e., $Q \in H_1$, and hence $Q \in H_1'$.

**Design $P$.** Next, we design $P$ as a small modification of the distribution $Q$. We recall that $P \in H_0$ means that $\gamma \leq \epsilon$ where $\epsilon < 0$ in the flipped hypothesis test. This means that, under $H_0$, there is one group that suffers a decrease in performance of $|\epsilon|$ because of the personalized model.

Given $p$ as a distribution with $\mu = 0$, and $p^\epsilon$ a distribution with mean $\mu = \epsilon < 0$, we have:

$$P_j(b_j) = \prod_{k=1}^{m_j} p(b_j^{(k)}), \text{ for every group } j = 1....d,$$

$$P_j^\epsilon(b_j) = \prod_{k=1}^{m_j} p^\epsilon(b_j^{(k)}), \text{ for every group } j = 1....d,$$

$$P(b) = \frac{1}{d} \sum_{j=1}^{d} P_j'(b) \text{ where: } P_j'(b) = P_j^\epsilon(b_j) \prod_{j' \neq j} P_{j'}(b_{j'}).$$

Intuitively, the distribution $P$ represents the fact that the personalized model worsen performances by $|\epsilon|$ for one group. We construct the mixture by assuming that this group can be either group 1, or group 2, etc, or group $d$, and consider these to be disjoint, equiprobable events.

We verify that we have designed $P$ correctly, i.e., we check that each of its component is in $H_0$. When the dataset is distributed according to one component $P_j'$, we have:

$$\gamma = \min_{s \in S} C_s(h_0, s) - C_s(h_p, s)$$
$$= \min_{s \in S} \mathbb{E}_{P_j'}[\mathbf{B} \mid \mathbf{S} = s] \text{ (same computations as for } Q \in H_1)$$
$$= \min(\epsilon, 0, ..., 0) \text{ (since exactly group has mean } \epsilon)$$
$$= \epsilon \text{ (since } \epsilon < 0).$$

Thus, we find that $\gamma = \epsilon$ which means that $\gamma \leq \epsilon$, i.e., $P_j' \in H_0$ for every $j$, i.e., $P \in H_0$.

**Compute total variation $TV(P \parallel Q)$.** We have verified that $Q \in H_1$ and that $P \in H_0$. We use these probability distributions to compute the lower bound for $P_e$. First, we compute their total variation:

$$TV(P \parallel Q) = \frac{1}{2} \int_{b_1,...,b_j} |P(b_1, ..., b_j) - Q(b_1, ..., b_j)| \, db_1...db_j \text{ (TV for probability density functions)}$$

$$= \frac{1}{2} \int_{b_1,...,b_j} \left| \frac{1}{d} \sum_{j=1}^{d} P_j^\epsilon(b_j) \prod_{j' \neq j} P_{j'}(b_{j'}) - \prod_{j=1}^{d} Q_j(b_j) \right| db_1...db_j \text{ (definition of } P, Q)$$

$$= \frac{1}{2} \int_{b_1,...,b_j} \left| \frac{1}{d} \sum_{j=1}^{d} \frac{P_j^\epsilon(b_j)}{P_j(b_j)} \prod_{j'=1}^{d} P_{j'}(b_{j'}) - \prod_{j=1}^{d} Q_j(b_j) \right| db_1...db_j \text{ (adding missing } j' = j)$$

$$= \frac{1}{2} \int_{b_1,...,b_j} \left| \frac{1}{d} \sum_{j=1}^{d} \frac{P_j^\epsilon(b_j)}{P_j(b_j)} \prod_{j'=1}^{d} Q_{j'}(b_{j'}) - \prod_{j=1}^{d} Q_j(b_j) \right| db_1...db_j \text{ (} P_j = Q_j \text{ by construction)}$$

$$= \frac{1}{2} \int_{b_1,...,b_j} \prod_{j=1}^{d} Q_j(b_j) \left| \frac{1}{d} \sum_{j=1}^{d} \frac{P_j^\epsilon(b_j)}{P_j(b_j)} - 1 \right| db_1...db_j \text{ (extracting the product)}$$

$$= \frac{1}{2} \mathbb{E}_Q \left[ \left| \frac{1}{d} \sum_{j=1}^{d} \frac{P_j^\epsilon(b_j)}{P_j(b_j)} - 1 \right| \right] \text{ (recognizing an expectation with respect to } Q)$$

$$= \frac{1}{2}\mathbb{E}_Q\left[\left|\frac{1}{d}\sum_{j=1}^d \frac{\prod_{k=1}^{m_j} p^\epsilon(b_j^{(k)})}{\prod_{k=1}^{m_j} p(b_j^{(k)})} - 1\right|\right] \text{ (definition of } P_j \text{ and } P_j^{(\epsilon)})$$

$$\leq \frac{1}{2}\mathbb{E}_Q\left[\left|\frac{1}{d}\sum_{j=1}^d \frac{\prod_{k=1}^{m_j} p^\epsilon(b_j^{(k)})}{\prod_{k=1}^{m_j} p(b_j^{(k)})} - 1\right|^2\right]^{1/2} \text{ (Cauchy-Schwartz)}$$

**Auxiliary computation to apply Lemma D.2**   Next, we will apply Lemma D.2. For this, we need to prove that the expectation of the first term is 1. We have:

$$\mathbb{E}_Q\left[\frac{1}{d}\sum_{j=1}^d \frac{\prod_{k=1}^{m_j} p^\epsilon(b_j^{(k)})}{\prod_{k=1}^{m_j} p(b_j^{(k)})}\right]$$

$$= \frac{1}{d}\sum_{j=1}^d \mathbb{E}_Q\left[\frac{\prod_{k=1}^{m_j} p^\epsilon(b_j^{(k)})}{\prod_{k=1}^{m_j} p(b_j^{(k)})}\right] \text{ (linearity of expectation)}$$

$$= \frac{1}{d}\sum_{j=1}^d \mathbb{E}_Q\left[\prod_{k=1}^{m_j} \frac{p^\epsilon(b_j^{(k)})}{p(b_j^{(k)})}\right] \text{ (rearranging the product)}$$

$$= \frac{1}{d}\sum_{j=1}^d \prod_{k=1}^{m_j} \mathbb{E}_Q\left[\frac{p^\epsilon(b_j^{(k)})}{p(b_j^{(k)})}\right] \text{ (product of independent variables)}$$

$$= \frac{1}{d}\sum_{j=1}^d \prod_{k=1}^{m_j} \mathbb{E}_p\left[\frac{p^\epsilon(b_j^{(k)})}{p(b_j^{(k)})}\right] \text{ (definition of } Q)$$

$$= \frac{1}{d}\sum_{j=1}^d \prod_{k=1}^{m_j} \int_{-\infty}^{+\infty} \frac{p^\epsilon(b)}{p(b)}p(b)db \text{ (definition of expectation in } p)$$

$$= \frac{1}{d}\sum_{j=1}^d \prod_{k=1}^{m_j} \int_{-\infty}^{+\infty} p^\epsilon(b)db \text{ (simplify)}$$

$$= \frac{1}{d}\sum_{j=1}^d \prod_{k=1}^{m_j} 1 \text{ (probability density function integrates to 1)}$$

$$= \frac{1}{d}\sum_{j=1}^d 1 \text{ (term independent of } k)$$

$$= \frac{1}{d}d \text{ (term independent of } j)$$

$$= 1.$$

**Continue by applying Lemma D.2.**   This auxiliary computation shows that we meet the assumption of Lemma D.2. Thus, we continue the computation of the lower bound of the TV by applying Lemma D.2.

$$TV(P \parallel Q) \leq \frac{1}{2}\mathbb{E}_Q\left[\left(\frac{1}{d}\sum_{j=1}^d \frac{\prod_{k=1}^{m_j} p^\epsilon(b_j^{(k)})}{\prod_{k=1}^{m_j} p(b_j^{(k)})}\right)^2 - 1\right]^{\frac{1}{2}} \text{ Lemma D.2}$$

$$= \frac{1}{2}\mathbb{E}_Q\left[\left(\frac{1}{d}\sum_{j=1}^d z_j\right)^2 - 1\right]^{\frac{1}{2}} \text{ defining } z_j = \frac{\prod_{k=1}^{m_j} p^\epsilon(b_j^{(k)})}{\prod_{k=1}^{m_j} p(b_j^{(k)})} = \prod_{k=1}^{m_j} \frac{p^\epsilon(b_j^{(k)})}{p(b_j^{(k)})}$$

$$= \frac{1}{2} \mathbb{E}_Q \left[ \frac{1}{d^2} \sum_{j,j'=1}^{d} z_j z_{j'} - 1 \right]^{\frac{1}{2}} \quad \text{expanding the square of the sum}$$

$$= \frac{1}{2} \mathbb{E}_Q \left[ \frac{1}{d^2} \left( \sum_{j=1}^{d} z_j^2 + \sum_{j,j'=1,j\neq j'}^{d} z_j . z_{j'} \right) - 1 \right]^{\frac{1}{2}},$$

where we split the double sum to get independent variables in the second term.

We get by linearity of the expectation, $\mathbb{E}[aX + bY] = a\mathbb{E}[X] + b\mathbb{E}[Y]$:

$$TV(P \parallel Q) \leq \frac{1}{2} \mathbb{E}_Q \left[ \frac{1}{d^2} \left( \sum_{j=1}^{d} z_j^2 + \sum_{j,j'=1,j\neq j'}^{d} z_j . z_{j'} \right) - 1 \right]^{\frac{1}{2}}$$

$$= \frac{1}{2} \left[ \frac{1}{d^2} \left( \sum_{j=1}^{d} \mathbb{E}_Q[z_j^2] + \sum_{j,j'=1,j\neq j'}^{d} \mathbb{E}_Q[z_j . z_{j'}] \right) - 1 \right]^{\frac{1}{2}}$$

$$= \frac{1}{2} \left[ \frac{1}{d^2} \left( \sum_{j=1}^{d} \mathbb{E}_Q \left[ \left( \prod_{k=1}^{m_j} \frac{p^\epsilon(b_j^{(k)})}{p(b_j^{(k)})} \right)^2 \right] + \sum_{j,j'=1,j\neq j'}^{d} \mathbb{E}_Q \left[ \left( \prod_{k=1}^{m_j} \frac{p^\epsilon(b_j^{(k)})}{p(b_j^{(k)})} \right) \cdot \left( \prod_{k=1}^{m_j} \frac{p^\epsilon(b_{j'}^{(k)})}{p(b_{j'}^{(k)})} \right) \right] \right) - 1 \right]^{\frac{1}{2}}$$

$$= \frac{1}{2} \left[ \frac{1}{d^2} \left( \sum_{j=1}^{d} \mathbb{E}_Q \left[ \left( \prod_{k=1}^{m_j} \frac{p^\epsilon(b_j^{(k)})}{p(b_j^{(k)})} \right)^2 \right] \right. \right.$$

$$\left. \left. + \sum_{\substack{j,j'=1 \\ j\neq j'}}^{d} \mathbb{E}_Q \left[ \prod_{k=1}^{m_j} \frac{p^\epsilon(b_j^{(k)})}{p(b_j^{(k)})} \right] \mathbb{E}_Q \left[ \prod_{k=1}^{m_{j'}} \frac{p^\epsilon(b_{j'}^{(k)})}{p(b_{j'}^{(k)})} \right] \right) - 1 \right]^{\frac{1}{2}} \quad \text{(product of independent variables)}$$

$$= \frac{1}{2} \left[ \frac{1}{d^2} \left( \sum_{j=1}^{d} \prod_{k=1}^{m_j} \mathbb{E}_p \left[ \left( \frac{p^\epsilon(b_j^{(k)})}{p(b_j^{(k)})} \right)^2 \right] \right. \right.$$

$$\left. \left. + \sum_{\substack{j,j'=1 \\ j\neq j'}}^{d} \prod_{k=1}^{m_j} \mathbb{E}_p \left[ \frac{p^\epsilon(b_j^{(k)})}{p(b_j^{(k)})} \right] \prod_{k=1}^{m_{j'}} \mathbb{E}_p \left[ \frac{p^\epsilon(b_{j'}^{(k)})}{p(b_{j'}^{(k)})} \right] \right) - 1 \right]^{\frac{1}{2}} \quad \text{(product of independent variables and def. of } Q)$$

$$= \frac{1}{2} \left[ \frac{1}{d^2} \left( \sum_{j=1}^{d} \prod_{k=1}^{m_j} \mathbb{E}_p \left[ \left( \frac{p^\epsilon(b_j^{(k)})}{p(b_j^{(k)})} \right)^2 \right] + \sum_{j,j'=1,j\neq j'}^{d} \prod_{k=1}^{m_j} 1 \prod_{k=1}^{m_j} 1 \right) - 1 \right]^{\frac{1}{2}} \quad \text{(auxiliary computation below)}$$

$$= \frac{1}{2} \left[ \frac{1}{d^2} \left( \sum_{j=1}^{d} \prod_{k=1}^{m_j} \mathbb{E}_p \left[ \left( \frac{p^\epsilon(b_j^{(k)})}{p(b_j^{(k)})} \right)^2 \right] + \sum_{j,j'=1,j\neq j'}^{d} 1 \right) - 1 \right]^{\frac{1}{2}} \quad \text{(term independent of } k)$$

$$= \frac{1}{2} \left[ \frac{1}{d^2} \left( \sum_{j=1}^{d} \prod_{k=1}^{m_j} \mathbb{E}_p \left[ \left( \frac{p^\epsilon(b_j^{(k)})}{p(b_j^{(k)})} \right)^2 \right] + (d^2 - d) \right) - 1 \right]^{\frac{1}{2}} \quad \text{(term independent of } j)$$

$$= \frac{1}{2} \left[ \frac{1}{d^2} \left( \sum_{j=1}^{d} \mathbb{E}_p \left[ \left( \frac{p^\epsilon(B)}{p(B)} \right)^2 \right]^{m_j} + (d^2 - d) \right) - 1 \right]^{\frac{1}{2}} \quad \text{(term independent of } k)$$

$$= \frac{1}{2} \left[ \frac{1}{d^2} \sum_{j=1}^{d} \mathbb{E}_p \left[ \left( \frac{p^\epsilon(B)}{p(B)} \right)^2 \right]^{m_j} + 1 - \frac{1}{d} - 1 \right]^{\frac{1}{2}} \quad \text{(distribute } 1/d^2)$$

$$= \frac{1}{2} \left[ \frac{1}{d^2} \sum_{j=1}^{d} \mathbb{E}_p \left[ \left( \frac{p^\epsilon(B)}{p(B)} \right)^2 \right]^{m_j} - \frac{1}{d} \right]^{\frac{1}{2}} \quad \text{(simplify)}$$

$$= \frac{1}{2\sqrt{d}} \left[ \frac{1}{d} \sum_{j=1}^{d} \mathbb{E}_p \left[ \left( \frac{p^\epsilon(B)}{p(B)} \right)^2 \right]^{m_j} - 1 \right]^{\frac{1}{2}} \quad \text{(extract } 1/\sqrt{d})$$

$$= \frac{1}{2\sqrt{d}} \left[ \frac{1}{d} \sum_{j=1}^{d} \left( \int_{-\infty}^{+\infty} \left( \frac{p^\epsilon(b)}{p(b)} \right)^2 p(b) db \right)^{m_j} - 1 \right]^{\frac{1}{2}} \quad \text{(definition of expectation)}$$

$$= \frac{1}{2\sqrt{d}} \left[ \frac{1}{d} \sum_{j=1}^{d} \left( \int_{-\infty}^{+\infty} \frac{p^\epsilon(b)^2}{p(b)} db \right)^{m_j} - 1 \right]^{\frac{1}{2}} \quad \text{(simplify } p(b))$$

$$= \frac{1}{2\sqrt{d}} \left[ \frac{1}{d} \sum_{j=1}^{d} \mathbb{E}_{p^\epsilon} \left[ \frac{p^\epsilon(B)}{p(B)} \right]^{m_j} - 1 \right]^{\frac{1}{2}} \quad \text{(def of expectation)}$$

**Auxiliary computation in** 1    We show that:

$$\mathbb{E}_p \left[ \frac{p^\epsilon(b_{j'}^{(k)})}{p(b_{j'}^{(k)})} \right]$$

$$= \int_{-\infty}^{+\infty} \frac{p^\epsilon(b)}{p(b)} p(b) db$$

$$= \int_{-\infty}^{+\infty} p^\epsilon(b) db \text{ simplify } p(b)$$

$$= 1 \text{ probability density function } p^\epsilon \text{ integrates to 1.}$$

**Final result:**    This gives the final result:

$$\min_{\Psi} \max_{\substack{P_0 \in H_0' \\ P_1 \in H_1'}} P_e \geq \frac{1}{2} \left( 1 - TV(P \parallel Q) \right)$$

$$\Rightarrow \min_{\Psi} \max_{\substack{P_0 \in H_0' \\ P_1 \in H_1'}} P_e \geq \frac{1}{2} \left( 1 - \frac{1}{2\sqrt{d}} \left[ \frac{1}{d} \sum_{j=1}^{d} \mathbb{E}_{p^\epsilon} \left[ \frac{p^\epsilon(B)}{p(B)} \right]^{m_j} - 1 \right]^{\frac{1}{2}} \right)$$

$\square$

### D.2   PROOF FOR COROLLARY 5.2: ANY DISTRIBUTION IN AN EXPONENTIAL FAMILY

We consider a fixed exponential family in it natural parameterization, i.e., probability distributions of the form:

$$f_X(x \mid \boldsymbol{\theta}) = h(x) \exp(\theta \cdot \mathbf{T}(x) - A(\boldsymbol{\theta})), \tag{25}$$

where $\theta$ is the only parameter varying between two distributions from that family, i.e., the functions $\eta$, $T$ and $A$ are fixed. We recall a few properties of any exponential family (EF) that will be useful in our computations.

First, the moment generating function (MGF) for the natural sufficient statistic $T(x)$ is equal to:

$$M^T(t) = \exp\left( A(\theta + t) - A(\theta) \right).$$

Then, the moments for $T(x)$, when $\theta$ is a scalar parameter, are given by:

$$\mathrm{E}[T] = A'(\theta)$$

$$V[T] = A''(\theta).$$

Since the variance is non-negative $V[T] \geq 0$, this means that we have $A''(\theta) > 0$ and thus $A'$ is monotonic and bijective. We will use that fact in the later computations.

In the following, we recall that the categorical distribution and the Gaussian distribution with fixed variance $\sigma^2$ are members of the exponential family.

**Example: Categorical distributions as a EF**   The categorical variable has probability density function:

$$p(x \mid \pi) = \exp\left(\sum_{k=1}^{K} x_k \log \pi_k\right)$$

$$= \exp\left(\sum_{k=1}^{K-1} x_k \log \pi_k + \left(1 - \sum_{k=1}^{K-1} x_k\right) \log\left(1 - \sum_{k=1}^{K-1} \pi_k\right)\right)$$

$$= \exp\left(\sum_{k=1}^{K-1} \log\left(\frac{\pi_k}{1 - \sum_{k=1}^{K-1} \pi_k}\right) x_k + \log\left(1 - \sum_{k=1}^{K-1} \pi_k\right)\right)$$

where we have used the fact that $\pi_K = 1 - \sum_{k=1}^{K-1} \pi_k$.

We note that we need to use the PDF of the categorical that uses a minimal (i.e., $K-1$) set of parameters. We define $h(x), T(x), \theta \in \mathbb{R}^{K-1}$ and $A(\theta)$ as:

$h(x) = 1$

$T(x) = x,$

$$\theta_k = \log\left(\frac{\pi_k}{1 - \sum_{k=1}^{K-1} \pi_k}\right) = \log\left(\frac{\pi_k}{\pi_K}\right), \text{ for } k = 1, ..., K-1$$

$$A(\theta) = -\log\left(1 - \sum_{k=1}^{K-1} \pi_k\right) = \log\left(\frac{1}{1 - \sum_{k=1}^{K-1} \pi_k}\right) = \log\left(\frac{\sum_{k=1}^{K} \pi_k}{1 - \sum_{k=1}^{K-1} \pi_k}\right) = \log\left(\sum_{k=1}^{K} e^{\theta_k}\right),$$

which shows that the categorical distribution is within the EF. For convenience we have defined $\theta_K$ setting it to $0$ as per the Equation above.

Now, we adapt these expressions for the case of a Categorical variable with only $K = 3$ values $x_1 = -1, x_2 = 1$ and $x_3 = 0$ such that $\pi_3 = 0$, i.e., there is no mass on the $x_3 = 0$, and we denote $\pi_1 = p_1$ and $\pi_2 = p_2$ and $\pi_3 = 1 - p_1 - p_2 = 0$. We get:

$h(x) = 1$

$T(x) = x,$

$$\theta_1 = \log\left(\frac{p_1}{p_2}\right), \text{ and } \theta_2 = 0 \text{ by convention, as above, } \theta_3 = \log\left(\frac{\pi_3}{p_2}\right) = -\infty$$

$$A(\theta_1) = \log\left(e^{\theta_1} + e^{\theta_2} + e^{\theta_3}\right) = \log\left(e^{\theta_1} + 1 + 0\right) = \log\left(e^{\log\left(\frac{p_1}{p_2}\right)} + 1\right) = \log\left(\frac{p_1}{p_2} + 1\right).$$

**Example: Gaussian distribution with fixed variance as a EF**   The Gaussian distribution with fixed variance has probability density function:

$$p(x \mid \mu) = \frac{1}{\sqrt{2\pi\sigma^2}} \exp\left(-\frac{(x-\mu)^2}{2\sigma^2}\right)$$

$$= \frac{1}{\sqrt{2\pi\sigma^2}} \exp\left(-\frac{x^2 - 2x\mu + \mu^2}{2\sigma^2}\right)$$

$$= \frac{1}{\sqrt{2\pi\sigma^2}} \exp\left(-\frac{x^2}{2\sigma^2}\right) \exp\left(\frac{2x\mu - \mu^2}{2\sigma^2}\right)$$

$$= \frac{1}{\sqrt{2\pi\sigma^2}} \exp\left(-\frac{x^2}{2\sigma^2}\right) \exp\left(\frac{x\mu}{\sigma^2} - \frac{\mu^2}{2\sigma^2}\right).$$

We define $h(x), T(x), \theta \in \mathbb{R}$ and $A(\theta)$ as:

$$h(x) = \frac{1}{\sqrt{2\pi\sigma^2}} \exp\left(-\frac{x^2}{2\sigma^2}\right)$$
$$T(x) = x,$$
$$\theta = \frac{\mu}{\sigma^2}$$
$$A(\theta) = \frac{\mu^2}{2\sigma^2} = \frac{\sigma^2\theta^2}{2}.$$

which shows that the Gaussian distribution with fixed variance $\sigma^2$ is within the EF.

**Corollary** (Corollary 5.2 (restated))**.** *The lower bound for the exponential family with any number of samples in each group writes:*

$$\min_{\Psi} \max_{\substack{P_0 \in H'_0 \\ P_1 \in H'_1}} P_e \geq \frac{1}{2}\left(1 - \frac{1}{2\sqrt{d}}\left[\frac{1}{d}\sum_{j=1}^{d}\left(\frac{M_p(2\Delta\theta)}{M_p(\Delta\theta)^2}\right)^{m_j} - 1\right]^{\frac{1}{2}}\right).$$

*Proof.* By Theorem 5.1, we have:

$$\min_{\Psi} \max_{\substack{P_0 \in H'_0 \\ P_1 \in H'_1}} P_e \geq \frac{1}{2}\left(1 - \frac{1}{2\sqrt{d}}\left[\frac{1}{d}\sum_{j=1}^{d}\mathbb{E}_{p^\epsilon}\left[\frac{p^\epsilon(B)}{p(B)}\right]^{m_j} - 1\right]^{\frac{1}{2}}\right),$$

where $p, p^\epsilon$ are distributions with means $0$ and $\epsilon$ respectively.

**Plug in the exponential family** Under the assumption of an exponential family distribution for the random variable $B$, we have:

$$\min_{\Psi} \max_{\substack{P_0 \in H'_0 \\ P_1 \in H'_1}} P_e$$

$$\geq \frac{1}{2}\left(1 - \frac{1}{2\sqrt{d}}\left[\frac{1}{d}\sum_{j=1}^{d}\mathbb{E}_{p^\epsilon}\left[\frac{h(B)\exp(\theta^\epsilon.T(B) - A(\theta^\epsilon))}{h(B)\exp(\theta^0.T(B) - A(\theta^0))}\right]^{m_j} - 1\right]^{\frac{1}{2}}\right)$$

$$= \frac{1}{2}\left(1 - \frac{1}{2\sqrt{d}}\left[\frac{1}{d}\sum_{j=1}^{d}\mathbb{E}_{p^\epsilon}\left[\frac{\exp(\theta^\epsilon.T(B) - A(\theta^\epsilon))}{\exp(\theta^0.T(B) - A(\theta^0))}\right]^{m_j} - 1\right]^{\frac{1}{2}}\right) \text{ simplifying } h$$

$$= \frac{1}{2}\left(1 - \frac{1}{2\sqrt{d}}\left[\frac{1}{d}\sum_{j=1}^{d}\mathbb{E}_{p^\epsilon}\left[\exp(\theta^\epsilon.T(B) - A(\theta^\epsilon))\exp(-\theta^0.T(B) + A(\theta^0))\right]^{m_j} - 1\right]^{\frac{1}{2}}\right) \text{ properties of exp}$$

$$= \frac{1}{2}\left(1 - \frac{1}{2\sqrt{d}}\left[\frac{1}{d}\sum_{j=1}^{d}\mathbb{E}_{p^\epsilon}\left[\exp(A(\theta^0) - A(\theta^\epsilon))\right.\right.\right.$$

$$\left.\left.\left.\cdot \exp\left((\theta^\epsilon - \theta^0)\cdot T(B)\right)\right]^{m_j} - 1\right]^{\frac{1}{2}}\right) \text{(properties of exp and rearranging terms)}$$

$$= \frac{1}{2}\left(1 - \frac{1}{2\sqrt{d}}\left[\frac{1}{d}\sum_{j=1}^{d}\exp(A(\theta^0) - A(\theta^\epsilon))^{m_j}\mathbb{E}_{p^\epsilon}\left[\exp((\theta^\epsilon - \theta^0)T(B))\right]^{m_j} - 1\right]^{\frac{1}{2}}\right)$$

$$= \frac{1}{2} \left( 1 - \frac{1}{2\sqrt{d}} \left[ \frac{1}{d} \sum_{j=1}^{d} \exp(A(\theta^0) - A(\theta^\epsilon))^{m_j} M_{p^\epsilon}(\Delta\theta)^{m_j} - 1 \right]^{\frac{1}{2}} \right)$$

(def. of MGF of $T(B)$: $M_{p^\epsilon}(t) = \mathbb{E}_{p^\epsilon}[\exp(t \cdot T(B))]$ with $\Delta\theta = \theta^\epsilon - \theta^0$)

We define $\Delta\theta = \theta_\epsilon - \theta_0$. Here, we will apply the properties of EF regarding moment generating functions, i.e., for the $p^\epsilon$ with natural parameter $\theta_\epsilon$:

$$M_{p^\epsilon}(t) = \exp\left(A(\theta_\epsilon + t) - A(\theta_\epsilon)\right) \Rightarrow M_{p^\epsilon}(-\Delta\theta) = \exp\left(A(\theta_0) - A(\theta_\epsilon)\right),$$
$$\Rightarrow M_{p^\epsilon}(\Delta\theta) = \exp\left(A(2\theta_\epsilon - \theta_0) - A(\theta_\epsilon)\right),$$

And, for $p$ associated with natural parameter $\theta_0$:

$$M_p(t) = \exp\left(A(\theta_0 + t) - A(\theta_0)\right) \Rightarrow M_p(-\Delta\theta) = \exp\left(A(2\theta_0 - \theta_\epsilon) - A(\theta_0)\right),$$
$$\Rightarrow M_p(\Delta\theta) = \exp\left(A(\theta_\epsilon) - A(\theta_0)\right),$$
$$\Rightarrow M_p(\Delta\theta)^2 = \exp\left(2A(\theta_\epsilon) - 2A(\theta_0)\right)$$
$$\Rightarrow M_p(2\Delta\theta) = \exp\left(A(2\theta_\epsilon - \theta_0) - A(\theta_0)\right)$$

So, that we have on the one hand:

$$M_{p^\epsilon}(-\Delta\theta)M_{p^\epsilon}(\Delta\theta) = \exp\left(A(\theta_0) - A(\theta_\epsilon)\right) . \exp\left(A(2\theta_\epsilon - \theta_0) - A(\theta_\epsilon)\right)$$

and on the other hand:

$$\frac{M_p(2\Delta\theta)}{M_p(\Delta\theta)^2} = \frac{\exp\left(A(2\theta_\epsilon - \theta_0) - A(\theta_0)\right)}{\exp\left(2A(\theta_\epsilon) - 2A(\theta_0)\right)}$$
$$= \frac{\exp\left(A(2\theta_\epsilon - \theta_0)\right)}{\exp\left(2A(\theta_\epsilon) - 2A(\theta_0)\right)} \cdot \frac{1}{\exp\left(A(\theta_0)\right)}$$
$$= \frac{\exp\left(A(2\theta_\epsilon - \theta_0)\right)}{\exp\left(2A(\theta_\epsilon) - A(\theta_0)\right)}$$
$$= \exp\left(A(2\theta_\epsilon - \theta_0) + A(\theta_0) - A(\theta_\epsilon) - A(\theta_\epsilon)\right)$$
$$= \exp\left(A(\theta_0) - A(\theta_\epsilon) + A(2\theta_\epsilon - \theta_0) - A(\theta_\epsilon)\right)$$
$$= \exp\left(A(\theta_0) - A(\theta_\epsilon)\right) . \exp\left(A(2\theta_\epsilon - \theta_0) - A(\theta_\epsilon)\right)$$

Consequently, we get two equivalent expressions for our final result:

$$= \frac{1}{2} \left( 1 - \frac{1}{2\sqrt{d}} \left[ \frac{1}{d} \sum_{j=1}^{d} \exp(A(\theta^0) - A(\theta^\epsilon))^{m_j} \exp\left(A(2\theta_\epsilon - \theta_0) - A(\theta_\epsilon)\right)^{m_j} - 1 \right]^{\frac{1}{2}} \right)$$

$$= \frac{1}{2} \left( 1 - \frac{1}{2\sqrt{d}} \left[ \frac{1}{d} \sum_{j=1}^{d} \left(M_{p^\epsilon}(-\Delta\theta)M_{p^\epsilon}(\Delta\theta)\right)^{m_j} - 1 \right]^{\frac{1}{2}} \right) \quad \text{(first expression)}$$

$$= \frac{1}{2} \left( 1 - \frac{1}{2\sqrt{d}} \left[ \frac{1}{d} \sum_{j=1}^{d} \left(\frac{M_p(2\Delta\theta)}{M_p(\Delta\theta)^2}\right)^{m_j} - 1 \right]^{\frac{1}{2}} \right) \quad \text{(second expression)}$$

We will use the second expression.

$\square$

## D.3 PROOF FOR CATEGORICAL BoP

Here, we apply the exponential family result found in D.2 to find the lower bound for a categorical distribution.

**Corollary D.3.** *[Lower bound for categorical individual BoP for any number of samples in each group (Monteiro Paes et al., 2022)] The lower bound writes:*

$$\min_{\Psi} \max_{\substack{P_0 \in H_0' \\ P_1 \in H_1'}} P_e \geq \frac{1}{2} \left( 1 - \frac{1}{2\sqrt{d}} \left[ \frac{1}{d} \sum_{j=1}^{d} \left( 1 + 4\epsilon^2 \right)^{m_j} - 1 \right]^{\frac{1}{2}} \right)$$

*where $P_{\mathbf{X},\mathbf{S},Y}$ is a distribution of data, for which the generic model $h_0$ performs better, i.e., the true $\gamma$ is such that $\gamma(h_0, h_p, \mathcal{D}) < 0$, and $Q_{\mathbf{X},\mathbf{S},Y}$ is a distribution of data points for which the personalized model performs better, i.e., the true $\gamma$ is such that $\gamma(h_0, h_p, \mathcal{D}) \geq \epsilon$.*

*Proof.* By Corollary 5.2, we have:

$$\min_{\Psi} \max_{\substack{P_0 \in H_0' \\ P_1 \in H_1'}} P_e \geq \frac{1}{2} \left( 1 - \frac{1}{2\sqrt{d}} \left[ \frac{1}{d} \sum_{j=1}^{d} \left( \frac{M_p(2\Delta\theta)}{M_p(\Delta\theta)^2} \right)^{m_j} - 1 \right]^{\frac{1}{2}} \right)$$

**Plug in Categorical assumption**   We find the bound for the categorical case.

We need to choose $\theta^\epsilon$ such that $\mathbb{E}_{p^\epsilon}[B] = \epsilon$. We have:

$$\mathbb{E}_{p^\epsilon}[B] = -p_1^\epsilon + p_2^\epsilon.$$

We choose $p_1^\epsilon = \frac{1-\epsilon}{2}$ and $p_2^\epsilon = \frac{1+\epsilon}{2}$.

For the categorical, we have $\theta = \theta_1$ and:

$$\theta_0 = \log \left( \frac{p_1}{p_2} \right) = \log \frac{1/2}{1/2} = 0$$

$$\theta_\epsilon = \log \left( \frac{p_1^\epsilon}{p_2^\epsilon} \right) = \log \left( \frac{1-\epsilon}{1+\epsilon} \right)$$

$$A(\theta_0) = \log \left( e^{\theta_0} + 1 \right) = \log(2)$$

$$A(\theta_\epsilon) = \log \left( e^{\theta_\epsilon} + 1 \right) = \log \left( \frac{1-\epsilon}{1+\epsilon} + 1 \right) = \log \left( \frac{1-\epsilon+1+\epsilon}{1+\epsilon} \right) = \log \left( \frac{2}{1+\epsilon} \right)$$

$$A(2\theta_\epsilon) = \log \left( e^{2\theta_\epsilon} + 1 \right)$$
$$= \log \left( (e^{\theta_\epsilon})^2 + 1 \right)$$
$$= \log \left( \left( \frac{1-\epsilon}{1+\epsilon} \right)^2 + 1 \right)$$
$$= \log \left( \frac{1 - 2\epsilon + \epsilon^2}{1 + 2\epsilon + \epsilon^2} + 1 \right)$$
$$= \log \left( \frac{1 - 2\epsilon + \epsilon^2 + 1 + 2\epsilon + \epsilon^2}{1 + 2\epsilon + \epsilon^2} \right)$$
$$= \log \left( \frac{2 + 2\epsilon^2}{1 + 2\epsilon + \epsilon^2} \right)$$

We also have: $\Delta\theta = \theta_\epsilon$.

Accordingly, we have:

$$M_p(\Delta\theta) = \exp \left( A(\theta_0 + \Delta\theta) - A(\theta_0) \right)$$
$$= \exp \left( A(\theta_\epsilon) - A(\theta_0) \right)$$
$$= \exp \left( \log \left( \frac{2}{1+\epsilon} \right) - \log(2) \right)$$

$$= \exp \log \left( \frac{1}{2} \left( \frac{2}{1+\epsilon} \right) \right)$$

$$= \frac{1}{1+\epsilon}$$

$$M_p(2\Delta\theta) = \exp \left( A(2\theta_\epsilon - \theta_0) - A(\theta_0) \right)$$

$$= \exp \left( A(2\theta_\epsilon) - A(\theta_0) \right)$$

$$= \exp \left( \log \left( \frac{2 + 2\epsilon^2}{1 + 2\epsilon + \epsilon^2} \right) - \log(2) \right)$$

$$= \exp \log \left( \frac{1}{2} \frac{2 + 2\epsilon^2}{1 + 2\epsilon + \epsilon^2} \right)$$

$$= \frac{1 + \epsilon^2}{1 + 2\epsilon + \epsilon^2}$$

And the lower bound becomes:

$$\min_{\Psi} \max_{\substack{P_0 \in H_0' \\ P_1 \in H_1'}} P_e \geq \frac{1}{2} \left( 1 - TV(P \parallel Q) \right)$$

$$\Rightarrow \min_{\Psi} \max_{\substack{P_0 \in H_0' \\ P_1 \in H_1'}} P_e \geq \frac{1}{2} \left( 1 - \frac{1}{2\sqrt{d}} \left[ \frac{1}{d} \sum_{j=1}^{d} \left( \frac{M_p(2\Delta\theta)}{M_p(\Delta\theta)^2} \right)^{m_j} - 1 \right]^{\frac{1}{2}} \right)$$

$$= \frac{1}{2} \left( 1 - \frac{1}{2\sqrt{d}} \left[ \frac{1}{d} \sum_{j=1}^{d} \left( \frac{\frac{1+\epsilon^2}{1+2\epsilon+\epsilon^2}}{\left( \frac{1}{1+\epsilon} \right)^2} \right)^{m_j} - 1 \right]^{\frac{1}{2}} \right)$$

$$= \frac{1}{2} \left( 1 - \frac{1}{2\sqrt{d}} \left[ \frac{1}{d} \sum_{j=1}^{d} \left( \frac{\frac{1+\epsilon^2}{1+2\epsilon+\epsilon^2}}{\frac{1}{1+2\epsilon+\epsilon^2}} \right)^{m_j} - 1 \right]^{\frac{1}{2}} \right)$$

$$= \frac{1}{2} \left( 1 - \frac{1}{2\sqrt{d}} \left[ \frac{1}{d} \sum_{j=1}^{d} \left( 1 + \epsilon^2 \right)^{m_j} - 1 \right]^{\frac{1}{2}} \right)$$

$\square$

## D.4 MAXIMUM ATTRIBUTES (CATEGORICAL BoP) FOR ALL PEOPLE

In the case where dataset $\mathcal{D}$ is drawn from an unknown distribution and has $d$ groups where $d = 2^k$, with each group having $m = \lfloor N/d \rfloor$ samples, Corollary D.3 becomes:

$$\min_{\Psi} \max_{\substack{P_0 \in H_0' \\ P_1 \in H_1'}} P_e \geq \frac{1}{2} \left( 1 - \frac{1}{2\sqrt{d}} \left[ \left( 1 + \epsilon^2 \right)^m - 1 \right]^{\frac{1}{2}} \right)$$

**Corollary D.4** (Maximum attributes (categorical) for all people). *Consider auditing a personalized classifier $h_p$ to verify whether it provides a gain of $\epsilon = 0.01$ to each group on an auditing dataset $D$. Consider an auditing dataset with $N = 8 \times 10^9$ samples, or one sample for each person on earth. If $h_p$ uses more than $k \geq 17$ binary group attributes, then for any hypothesis test, there will exist a pair of probability distributions $P_{X,S,Y} \in H_0, \quad Q_{X,G,Y} \in H_1$ for which the test results in a probability of error that exceeds $25\%$.*

$$k \geq 17 \implies \min_{\Psi} \max_{\substack{P_{X,S,Y} \in H_0 \\ Q_{X,G,Y} \in H_1}} P_e \geq \frac{1}{4}. \tag{26}$$

### D.5 PROOF FOR GAUSSIAN BOP

Here, we do the proof assuming that the BoP is a normal variable with a second moment bounded by $\sigma^2$.

**Corollary D.5.** *[Lower bound for Gaussian BoP for any number of samples in each group] The lower bound writes:*

$$\min_{\Psi} \max_{\substack{P_0 \in H_0' \\ P_1 \in H_1'}} P_e \geq \frac{1}{2} \left( 1 - \frac{1}{2\sqrt{d}} \left[ \frac{1}{d} \sum_{j=1}^{d} \exp \left( \frac{m_j \epsilon^2}{\sigma^2} \right) - 1 \right]^{\frac{1}{2}} \right)$$

*where $P_{\mathbf{X},\mathbf{S},Y}$ is a distribution of data, for which the generic model $h_0$ performs better, i.e., the true $\gamma$ is such that $\gamma(h_0, h_p, \mathcal{D}) < 0$, and $Q_{\mathbf{X},\mathbf{S},Y}$ is a distribution of data points for which the personalized model performs better, i.e., the true $\gamma$ is such that $\gamma(h_0, h_p, \mathcal{D}) > 0$.*

*Proof.* By Corollary 5.2, we have:

$$\min_{\Psi} \max_{\substack{P_0 \in H_0' \\ P_1 \in H_1'}} P_e \geq \frac{1}{2} \left( 1 - \frac{1}{2\sqrt{d}} \left[ \frac{1}{d} \sum_{j=1}^{d} \left( \frac{M_p(2\Delta\theta)}{M_p(\Delta\theta)^2} \right)^{m_j} - 1 \right]^{\frac{1}{2}} \right)$$

**Plug in Gaussian assumption** We find the bound for the Gaussian case. For the Gaussian, we have:

$$\theta_0 = \frac{\mu_0}{\sigma^2} = 0$$

$$\theta_\epsilon = \frac{\mu_\epsilon}{\sigma^2} = \frac{\epsilon}{\sigma^2}$$

$$A(\theta_0) = \frac{\sigma^2 \theta_0^2}{2} = 0$$

$$A(\theta_\epsilon) = \frac{\sigma^2 \theta_\epsilon^2}{2} = \frac{\epsilon^2}{2\sigma^2}$$

$$A(2\theta_\epsilon) = \frac{\sigma^2 4\theta_\epsilon^2}{2} = \frac{2\epsilon^2}{\sigma^2}$$

because $\mu_0 = 0$ and $\mu_\epsilon = \epsilon$ by construction. Thus, we also have: $\Delta\theta = \theta_\epsilon$.

Accordingly, we have:

$$M_p(\Delta\theta) = \exp\left( A(\theta_0 + \Delta\theta) - A(\theta_0) \right) = \exp\left( A(\theta_\epsilon) - A(\theta_0) \right) = \exp\left( \frac{\epsilon^2}{2\sigma^2} \right)$$

$$M_p(2\Delta\theta) = \exp\left( A(\theta_0 + 2\Delta\theta) - A(\theta_0) \right) = \exp\left( A(2\theta_\epsilon - \theta_0) \right) = \exp\left( A(2\theta_\epsilon) \right) = \exp\left( \frac{2\epsilon^2}{\sigma^2} \right)$$

And the lower bound becomes:

$$\min_{\Psi} \max_{\substack{P_0 \in H_0' \\ P_1 \in H_1'}} P_e \geq \frac{1}{2} \left( 1 - TV(P \parallel Q) \right)$$

$$\Rightarrow \min_{\Psi} \max_{\substack{P_0 \in H_0' \\ P_1 \in H_1'}} P_e \geq \frac{1}{2} \left( 1 - \frac{1}{2\sqrt{d}} \left[ \frac{1}{d} \sum_{j=1}^{d} \left( \frac{M_p(2\Delta\theta)}{M_p(\Delta\theta)^2} \right)^{m_j} - 1 \right]^{\frac{1}{2}} \right)$$

$$= \frac{1}{2} \left( 1 - \frac{1}{2\sqrt{d}} \left[ \frac{1}{d} \sum_{j=1}^{d} \left( \frac{\exp\left( \frac{2\epsilon^2}{\sigma^2} \right)}{\exp\left( \frac{\epsilon^2}{2\sigma^2} \right)^2} \right)^{m_j} - 1 \right]^{\frac{1}{2}} \right)$$

$$= \frac{1}{2} \left( 1 - \frac{1}{2\sqrt{d}} \left[ \frac{1}{d} \sum_{j=1}^{d} \left( \frac{\exp\left( \frac{2\epsilon^2}{\sigma^2} \right)}{\exp\left( \frac{2\epsilon^2}{2\sigma^2} \right)} \right)^{m_j} - 1 \right]^{\frac{1}{2}} \right)$$

$$= \frac{1}{2} \left( 1 - \frac{1}{2\sqrt{d}} \left[ \frac{1}{d} \sum_{j=1}^{d} \left( \frac{\exp\left(\frac{2\epsilon^2}{\sigma^2}\right)}{\exp\left(\frac{\epsilon^2}{\sigma^2}\right)} \right)^{m_j} - 1 \right]^{\frac{1}{2}} \right)$$

$$= \frac{1}{2} \left( 1 - \frac{1}{2\sqrt{d}} \left[ \frac{1}{d} \sum_{j=1}^{d} \exp\left(\frac{\epsilon^2}{\sigma^2}\right)^{m_j} - 1 \right]^{\frac{1}{2}} \right)$$

$$= \frac{1}{2} \left( 1 - \frac{1}{2\sqrt{d}} \left[ \frac{1}{d} \sum_{j=1}^{d} \exp\left(\frac{m_j\epsilon^2}{\sigma^2}\right) - 1 \right]^{\frac{1}{2}} \right)$$

In the case where each group has a different standard deviation of their BoP distribution, this becomes:

$$= \frac{1}{2} \left( 1 - \frac{1}{2\sqrt{d}} \left[ \frac{1}{d} \sum_{j=1}^{d} \exp\left(\frac{m_j\epsilon^2}{\sigma_j^2}\right) - 1 \right]^{\frac{1}{2}} \right).$$

$\square$

### D.6 Maximum attributes (Gaussian BoP) for all people

In the case where dataset $\mathcal{D}$ is drawn from an unknown distribution and has $d$ groups where $d = 2^k$, with each group having $m = \lfloor N/d \rfloor$ samples, Corollary D.5 becomes:

$$\min_{\Psi} \max_{\substack{P_0 \in H_0' \\ P_1 \in H_1'}} P_e \geq \frac{1}{2} \left( 1 - \frac{1}{2\sqrt{d}} \left[ \exp\left(\frac{m\epsilon^2}{\sigma^2}\right) - 1 \right]^{\frac{1}{2}} \right)$$

**Corollary D.6** (Maximum attributes (Gaussian BoP) for all people)**.** *Consider auditing a personalized classifier $h_p$ to verify if it provides a gain of $\epsilon = 0.01$ to each group on an auditing dataset D. Consider an auditing dataset with $\sigma = 0.1$ and $N = 8 \times 10^9$ samples, or one sample for each person on earth. If $h_p$ uses more than $k \geq 23$ binary group attributes, then for any hypothesis test there will exist a pair of probability distributions $P_{X,S,Y} \in H_0$, $\quad Q_{X,S,Y} \in H_1$ for which the test results in a probability of error that exceeds 25%.*

$$k \geq 23 \implies \min_{\Psi} \max_{\substack{P_{X,S,Y} \in H_0 \\ Q_{X,G,Y} \in H_1}} P_e \geq \frac{1}{4}. \tag{27}$$

### D.7 Proof for the symmetric generalized normal distribution

We solve for the bound assuming the BoP is a symmetric generalized Gaussian distribution.

**Symmetric Generalized Gaussian** The symmetric generalized Gaussian distribution, also known as the exponential power distribution, is a generalization of the Gaussian distributions that includes the Laplace distribution. A probability distribution in this family has probability density function:

$$p(x|\mu, \alpha, \beta) = \frac{\beta}{2\alpha\Gamma(1/\beta)} \exp\left( - \left( \frac{|x - \mu|}{\alpha} \right)^{\beta} \right), \tag{28}$$

with mean and variance:

$$\mathbb{E}[X] = \mu, \quad V[X] = \frac{\alpha^2 \Gamma(3/\beta)}{\Gamma(1/\beta)}. \tag{29}$$

We can write the standard deviation $\sigma = \alpha\sqrt{\frac{\Gamma(1/\beta)}{\Gamma(3/\beta)}} = \alpha\gamma(\beta)$ where we introduce the notation $\gamma(\beta) = \sqrt{\frac{\Gamma(1/\beta)}{\Gamma(3/\beta)}}$. This notation will become convenient in our computations.

**Example: Laplace**    The Laplace probability density function is given by:

$$f(x \mid \mu, b) = \frac{1}{2b} \exp\left(-\frac{|x - \mu|}{b}\right) \tag{30}$$

which is in the family for $\alpha = b$ and $\beta = 1$, since the gamma function verifies $\Gamma(1) = (1 - 1)! = 0! = 1$.

**Proposition D.7.** *[Lower bound for symmetric generalized Gaussian BoP for any number of samples in each group] The lower bound writes:*

$$\min_{\Psi} \max_{\substack{P_0 \in H_0' \\ P_1 \in H_1'}} P_e \geq \frac{1}{2} \left( 1 - \frac{1}{2\sqrt{d}} \left[ \frac{1}{d} \sum_{j=1}^{d} \mathbb{E}_{p^\epsilon} \left[ \exp\left( -\frac{|B - \epsilon|^\beta - |B|^\beta}{\alpha^\beta} \right) \right]^{m_j} - 1 \right]^{\frac{1}{2}} \right)$$

*where $P_{\mathbf{X}, \mathbf{S}, Y}$ is a distribution of data, for which the generic model $h_0$ performs better, i.e., the true $\gamma$ is such that $\gamma(h_0, h_p, \mathcal{D}) < 0$, and $Q_{\mathbf{X}, \mathbf{S}, Y}$ is a distribution of data points for which the personalized model performs better, i.e., the true $\gamma$ is such that $\gamma(h_0, h_p, \mathcal{D}) > 0$.*

*Proof.* By Theorem 5.1, we have:

$$\min_{\Psi} \max_{\substack{P_0 \in H_0' \\ P_1 \in H_1'}} P_e \geq \frac{1}{2} \left( 1 - \frac{1}{2\sqrt{d}} \left[ \frac{1}{d} \sum_{j=1}^{d} \mathbb{E}_{p^\epsilon} \left[ \frac{p^\epsilon(B)}{p(B)} \right]^{m_j} - 1 \right]^{\frac{1}{2}} \right)$$

**Plug in the symmetric generalized Gaussian distribution**    Under the assumption that the random variable $B$ follows an exponential power distribution, we continue the computations as:

$$\min_{\Psi} \max_{\substack{P_0 \in H_0' \\ P_1 \in H_1'}} P_e \geq \frac{1}{2} \left( 1 - \frac{1}{2\sqrt{d}} \left[ \frac{1}{d} \sum_{j=1}^{d} \mathbb{E}_{p^\epsilon} \left[ \frac{\exp\left( -\left( \frac{|B - \epsilon|}{\alpha} \right)^\beta \right)}{\exp\left( -\left( \frac{|B|}{\alpha} \right)^\beta \right)} \right]^{m_j} - 1 \right]^{\frac{1}{2}} \right)$$

$$= \frac{1}{2} \left( 1 - \frac{1}{2\sqrt{d}} \left[ \frac{1}{d} \sum_{j=1}^{d} \mathbb{E}_{p^\epsilon} \left[ \exp\left( -\left( \frac{|B - \epsilon|}{\alpha} \right)^\beta \right) \cdot \exp\left( \left( \frac{|B|}{\alpha} \right)^\beta \right) \right]^{m_j} - 1 \right]^{\frac{1}{2}} \right)$$

(property of exp)

$$= \frac{1}{2} \left( 1 - \frac{1}{2\sqrt{d}} \left[ \frac{1}{d} \sum_{j=1}^{d} \mathbb{E}_{p^\epsilon} \left[ \exp\left( -\left( \frac{|B - \epsilon|}{\alpha} \right)^\beta + \left( \frac{|B|}{\alpha} \right)^\beta \right) \right]^{m_j} - 1 \right]^{\frac{1}{2}} \right) \quad \text{(property of exp)}$$

$$= \frac{1}{2} \left( 1 - \frac{1}{2\sqrt{d}} \left[ \frac{1}{d} \sum_{j=1}^{d} \mathbb{E}_{p^\epsilon} \left[ \exp\left( -\frac{|B - \epsilon|^\beta - |B|^\beta}{\alpha^\beta} \right) \right]^{m_j} - 1 \right]^{\frac{1}{2}} \right) \quad \text{(property of exp)}$$

$\square$

### D.8    Proof for Laplace BoP

Here, we do the proof assuming that the BoP is a Laplace distribution (for more peaked than the normal variable).

**Corollary D.8.** *[Lower bound for a Laplace BoP for any number of samples in each group] The lower bound writes:*

$$\min_{\Psi} \max_{\substack{P_0 \in H_0' \\ P_1 \in H_1'}} P_e \geq \frac{1}{2} \left( 1 - \frac{1}{2\sqrt{d}} \left[ \frac{1}{d} \sum_{j=1}^{d} \exp\left( \frac{m_j \epsilon}{b} \right) - 1 \right]^{\frac{1}{2}} \right)$$

*where $P_{\mathbf{X},\mathbf{S},Y}$ is a distribution of data, for which the generic model $h_0$ performs better, i.e., the true $\gamma$ is such that $\gamma(h_0, h_p, \mathcal{D}) < 0$, and $Q_{\mathbf{X},\mathbf{S},Y}$ is a distribution of data points for which the personalized model performs better, i.e., the true $\gamma$ is such that $\gamma(h_0, h_p, \mathcal{D}) > 0$.*

*Proof.* By Proposition D.7, we have:

$$\min_{\substack{\Psi}} \max_{\substack{P_0 \in H_0' \\ P_1 \in H_1'}} P_e \geq \frac{1}{2} \left( 1 - \frac{1}{2\sqrt{d}} \left[ \frac{1}{d} \sum_{j=1}^{d} \mathbb{E}_{p^\epsilon} \left[ \exp\left( -\frac{|B-\epsilon|^\beta - |B|^\beta}{\alpha^\beta} \right) \right]^{m_j} - 1 \right]^{\frac{1}{2}} \right)$$

Plugging in our values of $\alpha$ and $\beta$ shown to satisfy the Laplace probability density function we get:

$$= \frac{1}{2} \left( 1 - \frac{1}{2\sqrt{d}} \left[ \frac{1}{d} \sum_{j=1}^{d} \mathbb{E}_{p^\epsilon} \left[ \exp\left( -\frac{|B-\epsilon| - |B|}{b} \right) \right]^{m_j} - 1 \right]^{\frac{1}{2}} \right)$$

**Using bounds** Since we are finding the worst case lower bound, we will find functions that upper and lower bound $|B - \epsilon| - |B|$. This function is lower bounded by $\epsilon$ and upper bounded by $-\epsilon$ since $\epsilon < 0$. Indeed, since $\epsilon < 0$, there are three cases:

- $0 < B < B - \epsilon$: this gives $|B - \epsilon| - |B| = B - \epsilon - B = -\epsilon$
- $B < 0 < B - \epsilon$ : this gives $|B - \epsilon| - |B| = B - \epsilon + B = 2B - \epsilon > 2\epsilon - \epsilon = \epsilon$ since $0 < B - \epsilon$.
- $B < B - \epsilon < 0$: this gives $|B - \epsilon| - |B| = -B + \epsilon + B = \epsilon$.

Thus, we have: $\epsilon \leq |B - \epsilon| - |B| \leq -\epsilon$ and:

$$\epsilon \leq |B - \epsilon| - B \leq -\epsilon$$
$$\Rightarrow \frac{\epsilon}{b} \leq \frac{|B-\epsilon| - |B|}{b} \leq -\frac{\epsilon}{b}$$
$$\Rightarrow -\frac{\epsilon}{b} \geq -\frac{|B-\epsilon| - |B|}{b} \geq \frac{\epsilon}{b}$$
$$\Rightarrow \exp\left( -\frac{\epsilon}{b} \right) \geq \exp\left( -\frac{|B-\epsilon| - |B|}{b} \right) \geq \exp\left( \frac{\epsilon}{b} \right)$$

Thus, applying the expectation gives:

$$\mathbb{E}_{p^\epsilon} \left[ \exp\left( -\frac{\epsilon}{b} \right) \right] \geq \mathbb{E}_{p^\epsilon} \left[ \exp\left( -\frac{|B-\epsilon| - |B|}{b} \right) \right] \geq \mathbb{E}_{p^\epsilon} \left[ \exp\left( \frac{\epsilon}{b} \right) \right]$$
$$\Rightarrow \exp\left( -\frac{\epsilon}{b} \right) \geq \mathbb{E}_{p^\epsilon} \left[ \exp\left( -\frac{|B-\epsilon| - |B|}{b} \right) \right] \geq \exp\left( \frac{\epsilon}{b} \right)$$

because the lower and upper bounds do not depend on $B$.

All the terms in these inequalities are positive, and the power function is increasing on positive numbers. Thus, we get:

$$\exp\left( -\frac{\epsilon}{b} \right)^{m_j} \geq \mathbb{E}_{p^\epsilon} \left[ \exp\left( -\frac{|B-\epsilon| - |B|}{b} \right) \right]^{m_j} \geq \exp\left( \frac{\epsilon}{b} \right)^{m_j}$$
$$\Rightarrow \frac{1}{d} \sum_{j=1}^{d} \exp\left( -\frac{\epsilon}{b} \right)^{m_j} \geq \frac{1}{d} \sum_{j=1}^{d} \mathbb{E}_{p^\epsilon} \left[ \exp\left( -\frac{|B-\epsilon| - |B|}{b} \right) \right]^{m_j} \geq \frac{1}{d} \sum_{j=1}^{d} \exp\left( \frac{\epsilon}{b} \right)^{m_j}$$
$$\Rightarrow \frac{1}{d} \sum_{j=1}^{d} \exp\left( -\frac{m_j \epsilon}{b} \right) \geq \frac{1}{d} \sum_{j=1}^{d} \mathbb{E}_{p^\epsilon} \left[ \exp\left( -\frac{|B-\epsilon| - |B|}{b} \right) \right]^{m_j} \geq \frac{1}{d} \sum_{j=1}^{d} \exp\left( \frac{m_j \epsilon}{b} \right)$$

$$\Rightarrow \frac{1}{d}\sum_{j=1}^{d}\exp\left(-\frac{m_j\epsilon}{b}\right) - 1 \geq \frac{1}{d}\sum_{j=1}^{d}\mathbb{E}_{p^\epsilon}\left[\exp\left(-\frac{|B-\epsilon|-|B|}{b}\right)\right]^{m_j} - 1 \geq \frac{1}{d}\sum_{j=1}^{d}\exp\left(\frac{m_j\epsilon}{b}\right) - 1$$

$$\Rightarrow \left(\frac{1}{d}\sum_{j=1}^{d}\exp\left(-\frac{m_j\epsilon}{b}\right) - 1\right)^{\frac{1}{2}}$$

$$\geq \left(\frac{1}{d}\sum_{j=1}^{d}\mathbb{E}_{p^\epsilon}\left[\exp\left(-\frac{|B-\epsilon|-|B|}{b}\right)\right]^{m_j} - 1\right)^{\frac{1}{2}}$$

$$\geq \left(\frac{1}{d}\sum_{j=1}^{d}\exp\left(\frac{m_j\epsilon}{b}\right) - 1\right)^{\frac{1}{2}}$$

$$\Rightarrow -\frac{1}{2\sqrt{d}}\left(\frac{1}{d}\sum_{j=1}^{d}\exp\left(-\frac{m_j\epsilon}{b}\right) - 1\right)^{\frac{1}{2}}$$

$$\leq -\frac{1}{2\sqrt{d}}\left(\frac{1}{d}\sum_{j=1}^{d}\mathbb{E}_{p^\epsilon}\left[\exp\left(-\frac{|B-\epsilon|-|B|}{b}\right)\right]^{m_j} - 1\right)^{\frac{1}{2}}$$

$$\leq -\frac{1}{2\sqrt{d}}\left(\frac{1}{d}\sum_{j=1}^{d}\exp\left(\frac{m_j\epsilon}{b}\right) - 1\right)^{\frac{1}{2}}$$

**Back to Probability of error** To maximize $P_e$, we take the function that gives us the lower bound. Plugging this upper bound back into our equation for $P_e$:

$$\min_{\Psi}\max_{\substack{P_0\in H_0' \\ P_1\in H_1'}} P_e \geq \frac{1}{2}\left(1 - \frac{1}{2\sqrt{d}}\left[\frac{1}{d}\sum_{j=1}^{d}\exp\left(-\frac{m_j\epsilon}{b}\right) - 1\right]^{\frac{1}{2}}\right)$$

In the case where each group has a different scale parameter of their BoP distribution, this becomes:

$$\min_{\Psi}\max_{\substack{P_0\in H_0' \\ P_1\in H_1'}} P_e \geq \frac{1}{2}\left(1 - \frac{1}{2\sqrt{d}}\left[\frac{1}{d}\sum_{j=1}^{d}\exp\left(-\frac{m_j\epsilon}{b_j}\right) - 1\right]^{\frac{1}{2}}\right),$$

such that for the unflipped hypothesis testing with $\epsilon > 0$ we get:

$$\min_{\Psi}\max_{\substack{P_0\in H_0' \\ P_1\in H_1'}} P_e \geq \frac{1}{2}\left(1 - \frac{1}{2\sqrt{d}}\left[\frac{1}{d}\sum_{j=1}^{d}\exp\left(\frac{m_j\epsilon}{b_j}\right) - 1\right]^{\frac{1}{2}}\right)$$

$\square$

## D.9 MAXIMUM ATTRIBUTES (LAPLACE BOP) FOR ALL PEOPLE

In the case where dataset $\mathcal{D}$ is drawn from an unknown distribution and has $d$ groups where $d = 2^k$, with each group having $m = \lfloor N/d \rfloor$ samples, Corollary D.8 becomes:

$$\min_{\Psi}\max_{\substack{P_0\in H_0' \\ P_1\in H_1'}} P_e \geq \frac{1}{2}\left(1 - \frac{1}{2\sqrt{d}}\left[\exp\left(\frac{m\epsilon}{b}\right) - 1\right]^{\frac{1}{2}}\right).$$

**Corollary D.9** (Maximum attributes (Laplace) for all people)**.** *Consider auditing a personalized classifier $h_p$ to verify if it provides a gain of $\epsilon = 0.01$ to each group on an auditing dataset D.*

*Consider an auditing dataset with $\sigma = 0.1$ and $N = 8 \times 10^9$ samples, or one sample for each person on earth. If $h_p$ uses more than $k \geq 26$ binary group attributes, then for any hypothesis test there will exist a pair of probability distributions $P_{X,S,Y} \in H_0, \quad Q_{X,G,Y} \in H_1$ for which the test results in a probability of error that exceeds $25\%$.*

$$k \geq 26 \implies \min_{\Psi} \max_{\substack{P_{X,S,Y} \in H_0 \\ Q_{X,G,Y} \in H_1}} P_e \geq \frac{1}{4}. \tag{31}$$

## E    LIMITS ON ATTRIBUTES AND SAMPLE SIZE

This section derives theoretical limits on the number of personal attributes and the sample size required per group to ensure that the probability of error remains below a practitioner-specified threshold.

**Corollary E.1.** *Let $N$ be the number of participants, and assume that each group $j = 1, \ldots, d$ has $m_j = m = \left\lfloor \frac{N}{d} \right\rfloor$ samples. To ensure that the probability of error verifies $\min \max P_e \leq \frac{1}{4}$, the number of binary attributes $k$ must be chosen such that $k \leq k_{\max}$, where:*

$$k_{max} = \begin{cases} \frac{1}{\ln 2} W \left( N \log(\epsilon^2 + 1) \right) & \text{(Categorical BoP)} \\ \frac{1}{\ln 2} W \left( \frac{\epsilon^2 N}{\sigma^2} \right) & \text{(Gaussian BoP, variance } \sigma^2) \\ \frac{1}{\ln 2} W \left( \frac{\epsilon N}{b} \right) & \text{(Laplace BoP, scale } b), \end{cases}$$

*where $W$ is the Lambert W function.*

## F    MIMIC-III EXPERIMENT RESULTS

Below is all supplementary material for the MIMIC-III experiment. This includes G-BoP distribution plots and plots showing how incomprehensiveness and sufficiency change over the number of features removed.

### F.1    EXPERIMENT PLOTS

**Experiment Setup.** We assume that the practitioner uses a 70/30 train-test split for both tasks and compare two neural network models: a personalized model with one-hot encoded group attributes ($h_p$) and a generic model without them ($h_0$). Regression outputs are normalized to zero mean and unit variance.

**Explanation Method and Explanation Evaluation metric.** We assume that the practitioner generates the most important features of our models using Integrated Gradients from Captum as our explanation method (Sundararajan et al., 2017). We assume that they use sufficiency and incomprehensiveness as our explanation evaluation metrics, where $50\%$ of features are either kept or removed.

Integrated Gradients extracts the most important features of each model by computing input-feature attributions by integrating gradients along a path from a baseline to the input. To evaluate $\text{BoP}_X$ using sufficiency and incomprehensiveness, we set $r$ such that $50\%$ of features are kept or removed. Plots below depict how sufficiency and incomprehensiveness change for different values of $r$, as well as show the individual BoP distributions. We use Integrated Gradients for its efficiency, interpretability, and broad adoption, though our framework supports any attribution method.

In the following section, we show supplementary plots for the regression task on the auditing dataset. We show the distribution of the BoP across participants for all three metrics we evaluate. We overlay Laplace and Gaussian distributions to see which fit the individual BoP distribution best, illustrating that prediction and incomprehensiveness are best fit by Laplace distributions and sufficiency by a Gaussian distribution. Additionally, we show how incomprehensiveness and sufficiency change for the number of important attributes $r$ that are kept are removed.

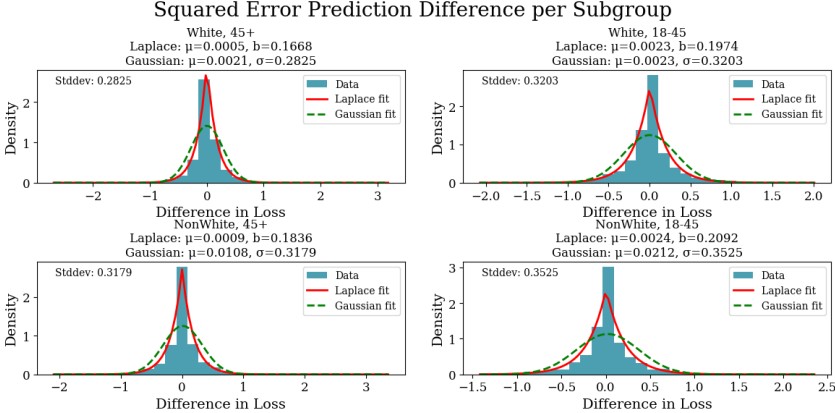

Figure 10: Individual prediction cost for all groups using the square error loss function.

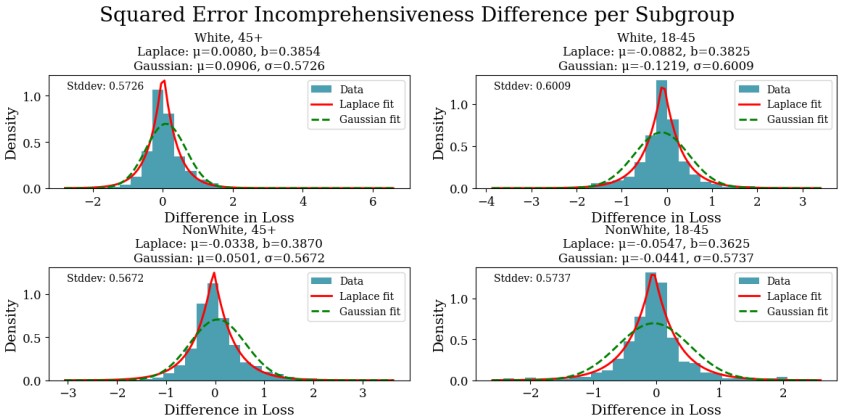

Figure 11: Individual incomprehensiveness cost for all groups using the square error loss function.

# G  ADDITIONAL DATASET RESULTS

The following are the experimental results for G-BoP$_P$ and G-BoP$_X$ on the UCI Heart (Janosi et al., 1989) and MIMIC-III Kidney injury dataset (Johnson et al., 2016) utilizing three explainer methods through Captum: Integrated Gradients Sundararajan et al. (2017), Shapley Value Sampling (Štrumbelj & Kononenko, 2010), and DeepLIFT (Shrikumar et al., 2017). Interestingly, we see a large amount of agreement across these explainer methods: in nearly all cases, groups that benefited or were harmed remain consistent across methods, although the amount by which this occurs varies. We compute $\epsilon_{lim}$, the value of $\epsilon$ for which the lower bound of $P_e$ surpasses 25% for the Shapley Value Sampling Method on the UCI Heart dataset to illustrate the full pipeline.

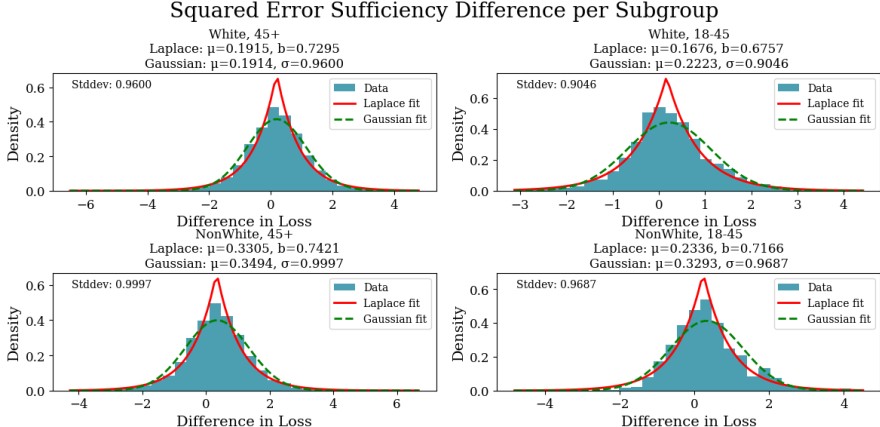

Figure 12: Individual sufficiency cost for all groups using the square error loss function.

Table 4: Experimental results on the UCI Heart test set, with columns for DeepLift (D.L.), Integrated Gradients (I.G.), and Shapley Value Sampling (S.V.S.). The classification task is predicting heart disease presence and the regression task is predicting ST depression induced by exercise. All available features are used, and negative entries appear in red. Using our framework, we computed $\epsilon_{\text{lim}}$ (for the S.V.S. explainer method) where the lower bound on $P_e$ surpasses 25%. In classification, $\epsilon_{\text{lim}} = 0.1156$ for all metrics; in regression, $\epsilon_{\text{lim}} = 0.0163$ for prediction (Laplace), $0.02$ for incomprehensiveness (Laplace), and $0.153$ for sufficiency (Gaussian). Given an $\epsilon = 0.002$, none of these tests are reliable.

| **Classification Results** | | | | | | | |
|---|---|---|---|---|---|---|---|
| **Group** | **Prediction** | **Incomp. D.L** | **Suff. D.L** | **Incomp. I.G.** | **Suff. I.G.** | **Incomp. S.V.S.** | **Suff. S.V.S.** |
| **Female, 45+** | 0.0000 | 0.0000 | -0.0435 | 0.0000 | -0.0435 | 0.0000 | -0.0870 |
| **Female, 18–45** | 0.0000 | -0.1429 | 0.0000 | -0.1429 | 0.0000 | 0.0000 | 0.0000 |
| **Male, 45+** | 0.0588 | -0.0588 | -0.0784 | -0.0588 | -0.1373 | -0.0588 | -0.1176 |
| **Male, 18–45** | 0.1000 | 0.1000 | 0.1000 | 0.1000 | 0.1000 | 0.1000 | 0.1000 |
| **All Pop.** | 0.0440 | -0.0330 | -0.0440 | -0.0330 | -0.0750 | -0.0220 | -0.0769 |
| **Minimal BoP** | 0.0000 | -0.1429 | -0.0784 | -0.1429 | -0.1373 | -0.0588 | -0.1176 |

| **Regression Results** | | | | | | | |
|---|---|---|---|---|---|---|---|
| **Group** | **Prediction** | **Incomp. D.L** | **Suff. D.L** | **Incomp. I.G** | **Suff. I.G** | **Incomp. S.V.S.** | **Suff. S.V.S.** |
| **Female, 45+** | -0.3077 | 0.3528 | 0.1385 | 0.0980 | 0.2040 | 0.1747 | 0.3332 |
| **Female, 18–45** | 0.0521 | -0.0004 | 0.1067 | -0.0438 | 0.1774 | -0.0207 | 0.0222 |
| **Male, 45+** | 0.0914 | 0.0286 | 0.0531 | 0.0173 | 0.1381 | 0.0315 | 0.1617 |
| **Male, 18–45** | -0.1410 | 0.1239 | 0.4293 | 0.1384 | 0.4365 | 0.1360 | 0.3592 |
| **All Pop.** | -0.0363 | 0.0791 | 0.1833 | 0.0523 | 0.2035 | 0.0779 | 0.2258 |
| **Minimal BoP** | -0.3077 | -0.0004 | 0.0531 | -0.0438 | 0.1381 | -0.0207 | 0.0222 |

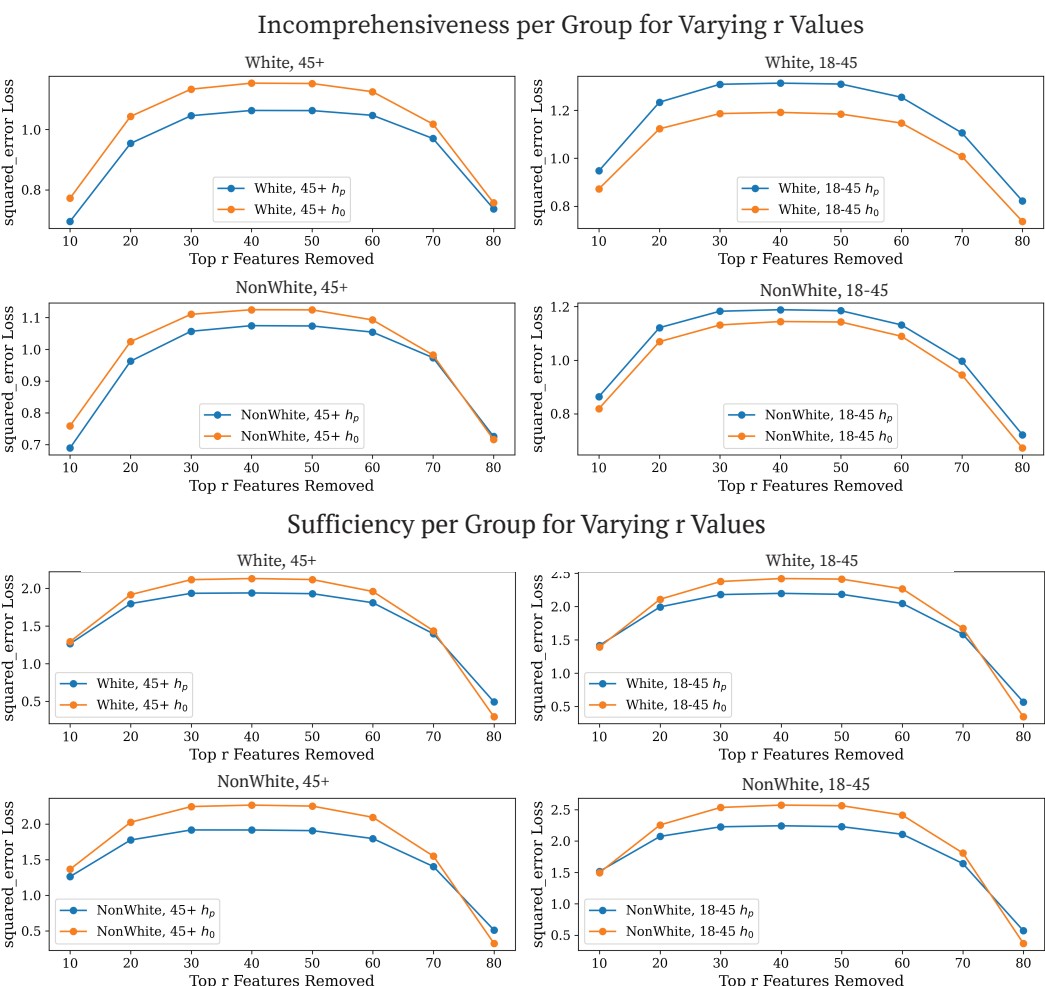

Figure 13: Values of Sufficiency and Incomprehensiveness across varying $r$ top features selected using the square error loss function. Values are found for $h_0$ and $h_p$.

Table 5: Experimental results on the MIMIC-III Kidney test set, with columns for DeepLift (D.L.), Integrated Gradients (I.G.), and Shapley Value Sampling (S.V.S.); negative values appear in red. The regression task predicts hours to the next continuous renal replacement therapy (CRRT). For classification, the target is patient mortality during the same hospital admission. Features include recent lab measurements (e.g., sodium, potassium, creatinine) prior to CRRT, along with patient age, hours in the ICU at CRRT administration, and the Sequential Organ Failure Assessment (SOFA) score at admission.

| Classification Results | | | | | | | |
|---|---|---|---|---|---|---|---|
| **Group** | **Prediction** | **Incomp. D.L** | **Suff. D.L** | **Incomp. I.G** | **Suff. I.G** | **Incomp. S.V.S.** | **Suff. S.V.S.** |
| **Female, 45+** | 0.0392 | 0.0392 | -0.0784 | 0.0392 | -0.0784 | 0.0392 | -0.0196 |
| **Female, 18–45** | 0.0000 | 0.0000 | 0.3636 | 0.0000 | 0.3636 | 0.0000 | 0.3636 |
| **Male, 45+** | 0.0164 | -0.0164 | 0.0820 | -0.0164 | 0.0984 | -0.0164 | 0.0000 |
| **Male, 18–45** | 0.0000 | 0.0000 | -0.0833 | 0.0000 | -0.0833 | 0.0000 | 0.1667 |
| **All Pop.** | 0.0224 | 0.0074 | 0.0296 | 0.0074 | 0.0370 | 0.0074 | 0.0370 |
| **Minimal BoP** | 0.0000 | -0.0164 | -0.0833 | -0.0164 | -0.0833 | -0.0164 | -0.0196 |

| Regression Results | | | | | | | |
|---|---|---|---|---|---|---|---|
| **Group** | **Prediction** | **Incomp. D.L** | **Suff. D.L** | **Incomp. I.G** | **Suff. I.G** | **Incomp. S.V.S.** | **Suff. S.V.S.** |
| **Female, 45+** | 0.7582 | 0.1440 | -0.5722 | 0.1322 | -0.6185 | 0.1380 | -0.5414 |
| **Female, 18–45** | 0.5639 | 0.0177 | -0.3325 | 0.0404 | -0.2543 | 0.0649 | -0.3107 |
| **Male, 45+** | 0.3449 | 0.0258 | -0.1180 | 0.0299 | -0.1368 | 0.0310 | -0.1518 |
| **Male, 18–45** | 0.4869 | -0.1016 | -0.1639 | -0.0997 | -0.1571 | -0.0892 | -0.2124 |
| **All Pop.** | -0.0093 | 0.0595 | -0.3097 | 0.0584 | -0.3311 | 0.0635 | -0.3167 |
| **Minimal BoP** | -0.0093 | -0.1016 | -0.5722 | -0.0997 | -0.6185 | -0.0892 | -0.5414 |

