# OpenReview forum: "When Machine Learning Gets Personal: Evaluating Prediction and Explanation"
_ICLR.cc/2026/Conference — ICLR 2026 Poster_

### Official Review · Reviewer_k2Uy · 2025-10-31

**Soundness:** 3
**Presentation:** 3
**Contribution:** 4
**Rating:** 8
**Confidence:** 3

**Summary:**

The paper introduces a unified framework to evaluate how *personalization* in machine learning affects both **prediction accuracy** and **explanation quality** across groups.
It extends the *Benefit of Personalization (BoP)* from binary classification to regression and explanation metrics (sufficiency, incomprehensiveness), proving that personalization can improve or harm interpretability even when accuracy is unchanged.
The authors derive finite-sample limits on the reliability of hypothesis tests for personalization and apply their framework to healthcare datasets (MIMIC-III, UCI Heart), showing that personalization effects are often untestable given dataset size and attribute structure.

**Strengths:**

- Extends BoP theory to continuous and explanation-based settings, bridging personalization, fairness, and explainability.
- Offers clear theoretical results (Thms. 4.1–4.3) showing prediction and explanation gains can diverge; provides a rare link between them under additive models.
- Derives practical finite-sample error bounds that guide when personalization tests are statistically valid.
- Empirical study (MIMIC-III) effectively illustrates the theoretical limits, highlighting cases where high empirical gains are statistically meaningless.
- Well-motivated and relevant to fairness, interpretability, and trustworthy AI.

**Weaknesses:**

- The explanation method is left vague in the main text; it becomes clear only in Appendix A. Stating early that the framework is demonstrated with Integrated Gradients, DeepLIFT, and SHAP would help.
- Heavy notation and scattered definitions (Tables 1 & 3) slow reading; concise examples or visual summaries could clarify.
- Some theorems (e.g., 4.3) claim necessity (“always”) rather than possibility (“can”)—the text should match this (compare lines 232 vs 242)
- Too much of the intuition (e.g., Fig. 6) is tucked into appendices, reducing accessibility. You have some nice plots!
- Limited discussion of connections to fairness of recourse (implying/not implying fairness of prediction) literature, despite thematic overlap.

**Questions:**

1. Does the framework apply beyond feature-attribution explainers (e.g., to counterfactual or surrogate explanations)?
2. Could the additive-model alignment in Thm. 4.3 generalize to linear or kernel settings?
3. How robust are the lower-bound results to distributional misspecification (e.g., Laplace vs Gaussian)?
4. Can the framework test *absence of harm* rather than *presence of benefit*?


### Comment on Table 3 (Appendix B):
Table 3 could be made more consistent with Table 1, particularly regarding the inclusion of s∖Js_{\setminus J}s∖J​ in the definition of incomprehensiveness. Clarifying this alignment would improve coherence across the theoretical and empirical sections.
It would also help readers if the caption or surrounding text explicitly stated that higher BoP values in each row indicate greater benefit for subgroup sss, reinforcing the interpretability of the metrics.

---

> ### Author Response · Authors · 2025-11-21
> **Response to Reviewer k2Uy (1/3)**
>
> Thank you for your positive evaluation and for your insightful questions, which allow us to further clarify the practical impact and scope of our contributions.
>
> **Weaknesses:**
>
> **Early Clarification of Explanation Methods.** Thanks for this suggestion. We have added a sentence state the explainers we use in Section 3.
>
> **Clarifying BoP Notation and Definitions.** Thank you for this feedback. To draw a link between the different locations of these tables, we have added a couple concrete examples from the table to the main text on L178-180.
>
> **Thm 4.3 and Text Matching.** Thanks for this catch, we have edited this in the manuscript and now state: “We show that this is true, for a simple additive model…”
>
> **Intuitive Figures in Appendix.** We appreciate this compliment! Unfortunately, due to the added revisions to the manuscript, we don’t have room to move the proof figures into the main body. We have added more explicit references to the figures for each proof in Section 4.
>
> **Incorporating Fairness of Recourse.** Good point. We have added the following paragraph in the extended related works:
> A related line of work examines fairness of algorithmic recourse, which studies whether different demographic groups face unequal effort to obtain favorable outcomes from a predictive model. Ustun et al. (2019) ([1] below) show that recourse burden can vary sharply across groups, even when recommended actions look formally identical, either because the recourse itself differs or because the real-world effort required to carry it out is unequal. This line of work demonstrates that fair prediction does not guarantee fair recourse. Our framework offers a complementary perspective: instead of analyzing post-hoc interventions, we study when personalization produces unequal benefits or harms across groups in prediction and explanation. Like the recourse literature, our results highlight that different desiderata, here, prediction benefit and explanation benefit, can diverge and therefore must be evaluated jointly.
>
> [1] Ustun, Berk, Alexander Spangher, and Yang Liu. "Actionable recourse in linear classification." In Proceedings of the conference on fairness, accountability, and transparency,
>
> (1/3)

---

> > ### Author Response · Authors · 2025-11-21
> > **Response to Reviewer k2Uy (2/3)**
> >
> > **Questions:**
> >
> > **Using Different Explainers.** Great question- the hypothesis testing machinery introduced in Section 5 is designed to operate on any cost function (across classification and regression tasks). Therefore, any explainability method can be used as long as it has an evaluation metric to evaluate the quality of the explanations. This allows practitioners to substitute the explainability or fairness metrics or methods most relevant to their specific application, not necessarily constraining them to feature-based explainers. We have added this comment as a remark in Section 5.
> >
> > **Theorem 4.3 Generalization** For the linear case, if you are referring to a linear model without assuming independence among the $X_j$’s or among the $S_j$’s, then the alignment in Theorem 4.3 does not hold in general. Independence plays an essential role in the proof: it ensures that both the prediction error and the sufficiency/incomprehensiveness scores decompose into sums of the form: $\alpha_j^2 \operatorname{Var}(Z_j)$, without cross-covariance terms. This decomposition is what allows the two explanation equalities to isolate the same contributions that appear in the prediction MSE, ultimately forcing $\operatorname{Var}(S)=0$. When $X$ and $S$ are dependent, these expressions pick up covariance terms, and the key algebraic cancellation no longer occurs. In an extended version of the proof, we show explicitly that zero BoP for explanation does not imply zero BoP for prediction unless a specific covariance condition holds. We have created an updated version of Thm. 4.3 by adding a corollary with the results from the extended proof to the end of Section 4:
> >
> > *Cor. 4.5. Under the same assumptions as Theorem 4.3, except that $\mathbf{X}$ and $\mathbf{S}$ are not assumed to be independent, if the following covariance condition holds: $\sum_{i \in J_p, i \le t} \sum_{j \in J_p, j \ge t+1} 2 \alpha_i \alpha_j \text{Cov}(X_i, S_{j-t}) + \sum_{i \in \backslash J_p, i \le t} \sum_{j \in \backslash J_p, j \ge t+1} 2 \alpha_i \alpha_j \text{Cov}(X_i, S_{j-t}) = 0$, then having $\operatorname{G\text{-}BoP}_X$ positive for sufficiency and incomprehensiveness implies that $\operatorname{G\text{-}BoP}_P = 0$.*
> >
> > For kernel methods, the situation is similar: the explanation and prediction terms do not reduce to coordinate-wise variance contributions, so the argument does not trivially generalize.
> >
> > (2/3)

---

> > > ### Author Response · Authors · 2025-11-21
> > > **Response to Reviewer k2Uy (3/3)**
> > >
> > > **Robustness of distribution classification analysis** This is a great question. To empirically understand what happens if the distributions are misclassified (ie a Gaussian distribution as a Laplace), we recalculated our lower bound on $P_e$ for our regression experimental results. Previously, prediction and incomprehensiveness used the Laplace form. Using the Gaussian form, they now cannot be reliably tested, with a lower bound of $P_e$ of greater than 80% for both at $\epsilon = 0.002$. Using the Laplace form for sufficiency, the lower bound is below 50% and it now shows no barriers to testing where it previously did. This underscores the importance of correctly classifying your distribution, as the lower bound of the $P_e$ can change significantly.
> > >
> > > To understand more general relationships between Gaussian and Laplace bounds, we made the following figure: https://imgur.com/a/MTANhMc. The figure compares the lower bound on the probability of error $P_e$ under Gaussian and Laplace distributions with matched scale parameters (\sigma for Gaussian and b for Laplace), shown for three dataset sizes $N$. For small datasets, the Gaussian curves yield a lower $P_e$ for the same $\epsilon$, so Gaussian assumptions give tighter guarantees when data are limited. As N increases, this ordering changes: the Laplace curves fall more quickly, and for large N they give a smaller lower bound on $P_e$ than the Gaussian curves. This illustrates that the relative tightness of the Gaussian and Laplace bounds depends on dataset size, with Laplace seemingly becoming more favorable once N is sufficiently large.
> > >
> > > We would also like to note that if none of these standard distributions are a good match for the BoP distribution, Theorem 5.1 remains valid for **any** distribution, as long as its probability density function is known. In such cases, practitioners may use flexible density estimation tools, such as normalizing flows, to approximate the PDF from data and apply Theorem 5.1 directly. We have added this insight into Section 5.
> > >
> > > **Evaluating Absence of Harm** Yes, the framework can evaluate absence of harm as well as presence of benefit. In our setup, the null hypothesis $H_0:\gamma \le 0 $ covers both harm and no improvement for at least one group. Rejecting H_0 certifies that no group is harmed, since doing so requires $\gamma \ge \varepsilon > 0$.
> > >
> > > The parameter $\epsilon$ sets the minimum benefit the practitioner expects from personalization. When the empirical $\hat{\gamma}$ is positive and the lower bound on the probability of error is below 50%, there is no statistical barrier to rejecting $H_0$; in that case, the framework provides a clear absence-of-harm guarantee.
> > >
> > > The same structure can also test for presence of harm by choosing a negative $\epsilon$ and reversing the hypotheses, giving a symmetric procedure for detecting degradation as well.
> > >
> > > **Clarifying Tables 1 and 3.** We absolutely understand this perceived misalignment: Table 1 defines our costs in terms of a general model (either personalized or not) and Table 3 uses the necessary forms of h_0 and h_p in the BoP definition. We have added a sentence, and your suggested clarifying sentence in the caption of Table 3.
> > >
> > > (3/3)

---

### Official Review · Reviewer_BgiC · 2025-10-31

**Soundness:** 3
**Presentation:** 2
**Contribution:** 3
**Rating:** 4
**Confidence:** 3

**Summary:**

The paper assesses the impact of personal information on explanation fidelity, fairness and model performance claiming that impact on predictions and explanations can be different.

**Strengths:**

* XAI is a hot topic with many open questions and investigating personalization and faithfulness is an interesting angle
* The paper aims at a strong mathematical underpinning paired with real-world examples

**Weaknesses:**

* The paper should early on state that they rely on importance scores, which covers only a subset of XAI methods though arguably the most important one (as the authors claim). Also the emphasis is on tabular data, which is a further narrowing down. Both should be clear from the abstract.

* The setting is not fully clear. On the one hand, the title should contain fairness rather than prediction (pointing towards overall prediction accuracy) as fairness seems to be the focus and not prediction accuary as also stated "We propose a framework to evaluate whether this weaker fairness criterion is satisfied, both theoretically and empirically, rather than proposing corrective algorithms."   . In turn, also some of the abstract and introduction might require rephrasing. On the other hand, H1 points towards an improvement of all groups. It is better to make one clear case on either of the two.

* There is no passage in the appendix (or main part) for theorm 5.1. called like "proof of theorem 5.1" ( it might have been called proof for theorem D.3 / 4.3 (which exist) or so), but I could not find the proof. While this might be a minor issue, it essentially makes the work non-verifyable.

* Given the inaccessibility of the proof (prior comment), I focused on the intution provided, which could be expanded. Essentially, the paper appears to say that if there is colinearity among features (i.e., the personalization ones and others) , faithfulness can be jeopardized.  While this sounds reasonable, it is not clear, why that must be really the case. It seems not really specific to personalization but a rather general issue with colinearity. Do you agree?

* The case-study style evaluation is not appreciated. It seems to jeopardize generalization and also makes an assessment clear given by all the practioners advice - a reference to an Appendix is of little help here, but some kind of convincing overview would be better. The promise of providing clear datasets characteristics (mentioned in abstract) to see when personalization hurts seems to be missing - or where is it?

Detail: use faithfulness/fidelity in the main part of the paper, not just in the Appendix, as it is a well-known term and clarifies some of our complex arguign, ie.  the sentence "this is by design:...." is obsolete if you clearly speak of fidelity.

**Questions:**

see above.

* Why do you need your framework and cannot just assess fairness using one of the many existing ways, ie.:
1) Add personalization features -> compute fairness metrics
2) Remove personalization features -> compute fairness metrics
3) Compare

---

> ### Author Response · Authors · 2025-11-21
> **Response to Reviewer BgiC (1/2)**
>
> Thank you for your thoughtful review. We address your concerns below.
>
> **Scope of Importance Scores and Explanation Methods** To clarify our setting, we have now added explicit mention of the use of tabular datasets and feature-attribution methods in the abstract (L23). We would like to clarify that while the theoretical results of Section 4 explicitly use Sufficiency and Incomprehensiveness, Section 5’s results are not limited to feature-attribution explanation methods: they can be applied to any explanation method that has a continuous or categorical quality metric. We have added a remark in Section 5 to note this. Additionally, while we illustrate results on tabular data, all of our utilized explanation methods can be applied to images, text, graphs, time series, or audio data.
>
> **Clarifying the Fairness Goal of Our Evaluation** We appreciate and understand this point. Our goal in this work is to evaluate whether personalizing a model improves prediction and explanation accuracy for every group, and bound the probability of error for these evaluations. Standard techniques for empirical risk minimization use personal characteristics to improve performance at a population level. However, these gains are not uniformly distributed across the groups who provide personal data. This is why H1 asserts an improvement of $\epsilon$ for all groups, because the practitioner decides a desired improvement gain for each group that makes personalization justified. Hence, we seek to evaluate prediction accuracy and explanation quality using fairness; we have modified the abstract and Introduction accordingly.
>
> **Correcting the Labeling of Proof of Thm 5.1.** We sincerely apologize for the initial confusing labeling, Thm 5.1 was labeled as Thm D.3 in Appendix D.1. We have edited the document to make all of the proof locations and labels explicitly clear, both in the main body and in the Appendices. Once again, we are very sorry this hindered the verifiability of some theoretical results.
>
> **Addressing the Role of Feature Collinearity.** Thank you for this question. This intuition is correct for the proofs in Section 4, where we illustrate how explanation and prediction gains can diverge or align. However, we would like to emphasize this doesn't relate with the aforementioned Thm 5.1 and the other results in Section 5. That being said, we agree that feature collinearity can affect explanations in general, but it plays a particularly relevant role in a personalized setting. Personalization introduces new features that might be correlated to existing ones, which may create redundant pathways in the model, not just in the data. If this happens, generic and personalized models can make the same predictions, but the personalized model’s added pathway changes how explanation methods distribute importance. This effect is specific to personalization, because it arises from the model update with the new feature rather than pre-existing feature correlations. We have added a Remark in Section 4 expanding on this insightful discussion.
>
> **Explaining Dataset Characteristics.** We are not quite clear on the first part of your feedback here. We would love clarification if possible so we can better understand how to improve Section 6. As for the second part, the core of our promised "dataset characteristics" is captured by the bounds we derive, which specify the maximum number of attributes ($k_{\max}$) and the minimum sample size ($m_{\min}$) needed to maintain a target lower bound on the probability of error. This information is critical practitioner advice on when and how evaluating personalization is theoretically limited. We refer you to the theoretical limits derived in Corollaries E.1 and 5.3:
> * **Corollary E.1 ($k_{\max}$):** This corollary provides the maximum number of binary attributes ($k_{\max}$) that can be included in a dataset of a given total size $N$ and a number of groups $d$ (where $m=\lfloor N/d\rfloor$) before the lower bound on the probability of error exceeds $50\%$. This directly answers the question of when having too many attributes  makes it impossible to reliably estimate the benefit of personalization.
> * **Corollary 5.3 ($m_{\min}$):** This corollary specifies the minimum required group sample size ($m_{\min}$) needed for a dataset of the same structure (fixed $N, d$) to keep the lower bound on the probability of error below a desired threshold $v$. This addresses the conditions (minimum data per group) required to avoid unreliable estimation.
>
> We empirically illustrate how the lower bound on the probability of error varies for different dataset characteristics in Figure 2. In Figure 2, we plot how the lower bound on the probability of error grows as the number of personal attributes increases. We show this for classification and regression metrics (Laplace and Gaussian distributions with different scale factors $b$ and $\sigma$), as well as varying dataset sizes.
>
> (1/2)

---

> > ### Author Response · Authors · 2025-11-21
> > **Response to Reviewer BgiC (2/2)**
> >
> > **Motivation for a Framework to Assess Fairness.** Thank you for this question, which strongly connects with the two-fold goal of our work:
> >
> > * First, we agree that practitioners often compare a model with and without personalization features. Our goal is not to replace that empirical approach, but to show when such comparisons cannot be relied upon, due to dataset statistics. We characterize the statistical limits under which any hypothesis test of “benefit vs. harm” is invalid for continuous metrics, and tighten the existing bound for categorical metrics.  Our results show that, depending on group sizes (Cor. 5.3) , the number of personal attributes (Cor. E.1), and the distribution of individual effects, the comparison you describe can have a lower bound error probability above 50%, meaning that even large observed differences cannot be trusted.
> >
> > * The second goal of the paper (Section 4) is to offer the first formal analysis demonstrating how personalization may differentially affect prediction and explanation. In particular, Theorems 4.1 and 4.2 prove that evaluating prediction and explanation jointly is necessary, while Theorem 4.3 initiates the important research program of understanding when these two objectives might align. The significance of these proofs lies in their conceptual contribution: to highlight that gains in prediction performance (as measured by $BoP_P$) and gains in explanation quality (as measured by $BoP_X$) need not align. Prior to our work, the relationship between predictive and explanatory benefits under personalization had not been formally analyzed, and it was unclear whether optimizing for one might implicitly guarantee improvement in the other. Our theorems explicitly show that this is not necessarily the case. These results lay the groundwork for a nuanced understanding of personalization: they motivate the need to assess both prediction and explanation benefits independently, rather than assuming alignment.

---

> > > ### Comment · Reviewer_BgiC · 2025-11-22
> > >
> > > >> Clarifying the Fairness Goal of Our Evaluation We appreciate and understand this point. Our goal in this work is to evaluate whether personalizing a model improves prediction and explanation accuracy for every group...
> > > When you have a (weak) fairness constraint, the work is about fairness. personalization often means each sample (-> think like your personalized ChatGPT instance based on your prior interactions). It is very tough in your setting to improve predictions for each sample -  I think your idea of group size is much larger than 1. Anyway, that is not a key point.
> > >
> > > >> Personalization introduces new features that might be correlated to existing ones, which may create redundant pathways in the model, not just in the data. If this happens, generic and personalized models can make the same predictions, but the personalized model’s added pathway changes how explanation methods distribute importance.
> > >
> > > Yes, no doubt, but this also happens to linear regression. My point being that if you add collinear features to a table, you already get that the trust in the coefficient is reduced (confidence interval). Now, I fully understand that for a neural network feature learning in such a case might pick any collinear feature and rely on it - which is well known and regularization techniques like dropout/L2 want to force the network to rely on all of them. I think the overall message is very interesting in a personalization context because I don't think people have thought about it, but if it is no surprise, i.e., if I did linear regression without your work and I would see that gender or so is collinear with other features, it is clear that this feature reduces confidence for others and might change explanations. Also as linear expression is very well understood theoretically, I am not sure if there are not any similar results to yours (though not phrased in that context).
> > >
> > > Nevertheless, the response is good and I am left with a better impression of the paper, but I don't consider update to 6 (from  the current 4).

---

> > > > ### Author Response · Authors · 2025-11-26
> > > >
> > > > While the observation that feature collinearity affects explanations may not be surprising to some in the context of linear regression, our work provides a formal confirmation of this intuition. This is valuable in itself and is also not obvious to the broader community. Explanation scores fundamentally differ from standard linear regression coefficients, and to our knowledge, no result on them has been formally established.
> > > >
> > > > Thank you for saying that our clarification improved your impression of the work and that the message is interesting. Given this positive feedback on the substance of our response, we find it difficult to reconcile with the unchanged evaluation score. To ensure a constructive final assessment, we would appreciate it if you could explicitly identify any remaining specific criteria preventing your score increase—especially regarding Section 5, which contains the core contribution of the paper.
> > > >
> > > > This targeted feedback would be very helpful, allowing us to make any remaining refinements and present the work as clearly and effectively as possible.

---

> > > > > ### Comment · Reviewer_BgiC · 2025-11-26
> > > > >
> > > > > I am not willing to increase to 6 given the contribution. I changed maybe from a 3.6 to a 4.4 mentally - which leaves it at a 4. I would expect a deeper discussion on the linear regression aspect and linear models more generally to deepen your results and relate them to existing knowledge, e.g.,
> > > > >
> > > > > For a linear model f(x) = beta^T x + b, interventional SHAP values under independent features reduce to:
> > > > > phi_i = beta_i * (x_i - E[X_i]).
> > > > > With correlated features, the Linear explainer shares credit among correlated variables via a closed-form, efficient computation that depends on the feature covariance.
> > > > >
> > > > > So given that your results are no big surprise in the XAI community. I recommend working out the details more.
> > > > >
> > > > > I won't have time to engage further at this point in time and hope that the feedback helps.
> > > > >
> > > > >
> > > > > 2: reject, not good enough
> > > > > 4: marginally below the acceptance threshold. But would not mind if paper is accepted
> > > > > 6: marginally above the acceptance threshold. But would not mind if paper is rejected

---

### Official Review · Reviewer_dhvG · 2025-11-03

**Soundness:** 2
**Presentation:** 3
**Contribution:** 2
**Rating:** 2
**Confidence:** 4

**Summary:**

The paper aims to measure how personalisation in machine learning impacts model performance and explainability. The authors show that, for the Bayes optimal classifier, accuracy and explainability are not necessarily aligned. However, when the data distribution follows an additive model, they show that there is a strong relationship between accuracy and explainability. The paper presents a statistical test to determine whether personlisation leads to improvements in accuracy and/or explainability.

**Strengths:**

- Paper is well written and clear for the most part
- Figure 3 and example scenario in Section 6 may be helpful for practitioners who might use the framework

**Weaknesses:**

**Weak Problem Statement**

There are two facets here:
1. Does this actually occur in practice?
2. If yes, what are the ramifications?

The authors don't really address these questions. I'd imagine the most salient case here is when we see a benefit in predictive accuracy but not in explainability (which kind of seems to be the case in Table 2?). For example, Monteiro Paes et al. (2022), demonstrate that personalisation may lead to harm on the second page (Table 1). Even still, we need to **infer** why a decrease in incomprehensiveness (or sufficiency) is undesirable. The authors should explicitly discuss the harm here.

I know that the authors state that the BoP framework is metric agnostic, but they should note that they have chosen these metrics to use throughout the paper, including in theoretical results.

The main motivation in this paper is:
> A common intuition is that if personalization improves prediction, it must also improve explanations (Del Giudice, 2024) (lines 189-190).

However, this doesn't seem to be the point of Del Giudice (2024). "Explanation quality" in this paper is different. Del Giudice is talking about the **model's** ability to represent real-world phenomena:
> The cost is that maximizing the predictive accuracy of a model tends to sacrifice its ability to represent the underlying phenomenon in an accurate and interpretable fashion (page 24).

This differs from the explanation metrics one might use in XAI, which aims to measure whether explanations faithfully explain **the model** (not nature).

**Results**
> Insight: The choice of improvement threshold $\varepsilon$ is key.

The paragraph here is just repeating the theoretical result in words; its quite trivial. The authors need to elaborate on this further.

Moreover, I can't seem to find the results that accompany the first and last insight.

**Overall Takeaway**

The authors conclude that there are limits to testing whether there is a benefit to personlisation. However, Monteiro Paes et al. have already demonstrated this in their work. Instead, if the authors have shown a tighter bound, they should frame the takeaway in light of those results.

**Questions:**

- Is it easy to construct distributions that show Theorem 4.1 and 4.2? I would be interested in how often this arises in practice.

---

> ### Author Response · Authors · 2025-11-21
> **Response to Reviewer dhvG (1/3)**
>
> Thank you for your thoughtful review. We address your concerns below.
>
> **Occurrence in Practice.** You are correct to notice that Table 2 illustrates how personalization differently affects predictive accuracy and explanation quality (for sufficiency and incomprehensiveness). Indeed, similar to Monteiro Paes Table 1, you can see in our Table 2 how some groups seem to experience an empirical gain in predictive accuracy and drop in explanation quality (for example on the classification task: White, 45+) and vice versa (for example on the classification task: NonWhite, 18-45). In the Appendix G, Tables 4 and 5 show the same diverging results on two additional datasets. We now emphasize the occurrence of this phenomenon in the revised manuscript (Introduction and Section 6).
>
> **On Making the Ramifications more Explicit.** These empirical patterns matter because changes in explanation quality have real consequences for users. Prior work (Dai et al., 2022) shows that explanation quality matters because users rely on explanations to judge when to trust a model, notice spurious reasoning, and correct mistakes. When fidelity is low for a particular group, the explanation gives an inaccurate picture of the model’s reasoning, which can lead to worse decisions for that group. To illustrate, Dai et al. give a healthcare example where the model relies on a spurious feature. Explanations for men accurately reveal this, so doctors mistrust the model and correct its mistakes. Explanations for women hide the spurious reasoning and instead highlight clinically relevant features, causing doctors to trust incorrect predictions and leading to unequal misdiagnosis rates. In our work, we use the standard notion of faithfulness (Lyu et al., 2024; Dasgupta et al., 2022; Jacovi and Goldberg, 2020). Sufficiency and comprehensiveness measure how well an explanation reflects the model’s reasoning. A drop in these scores means the explanation is less faithful, and the concerns above apply. We have added text to state this in the manuscript (L51-56).
>
> **On the Use of Particular Explanation Metrics.** Thank you for this feedback. To make this explicitly clear we have added a sentence noting this on L74 in the Introduction, and another on L419 in Section 6. We also clarify this aspect In Section 3 L162, and each of our Theorems 4.1-4.3 do mention that they are proven for these specific metrics.  Lastly, the validation framework presented in Section 5 is not tied to any particular explainability measure. We are happy to provide further details supporting the framework's metric agnosticism if you would like.
>
> **Motivation: User Expectations & Del Giudice (2024)** Thank you for the clarification. You are right that Del Giudice uses “explanation” to mean a model’s ability to represent the underlying phenomenon, which is different from faithfulness based XAI metrics. We cited this work not to claim an exact equivalence, but because it reflects a broader belief that better prediction should naturally lead to better explanations, either of real world structure as in Del Giudice or of a model’s internal reasoning in high stakes XAI settings. This belief also implies that the explanations produced by a higher performing model will be of higher quality (and can be trusted to understand underlying phenomena), and our results show that this is not always true. Our contribution is to formalize this risk in the context of personalization, independent of the specific notion of explanation used in Del Giudice, and we have revised the beginning of Section 4 to clarify this distinction.
>
> **Results:**
> **$\epsilon$ Insight:** We appreciate this feedback, and have added the following text to insight 2 to clarify the real-world significance of empirical results:
> _In real-world application, $\epsilon$ reflects the minimum performance gain a practitioner needs to justify collecting costly personal data (e.g., genetic markers in healthcare). For instance, $\epsilon=0.002$ in length-of-stay prediction accuracy may be worthwhile. Converting this back to original units is approximately 0.06 days more accurate. When scaled across just 100 patients daily, this seemingly small individual gain translates into a significant cumulative benefit: 6 days of hospital stay better estimated per day across those 100 patients. This illustrates that $\epsilon$ should be set based on practical value, not statistical convenience, even though higher ϵ values tend to make hypothesis tests more reliable. Our framework's impact is not just to suggest practitioners pick a threshold, but to provide the tool to determine if their data can even support a statistically meaningful conclusion at that threshold, moving beyond intuition to a quantitative, evidence-based decision._
>
> (1/3)

---

> ### Author Response · Authors · 2025-11-21
> **Response to Reviewer dhvG (2/3)**
>
> **Supporting Results for First and Last Insights.** Thanks for this remark.
>
>
> * For the **first insight**, it is referring to the results in Table 2 and analysis in paragraph 4. In Table 2, we show the empirical results of the benefit of personalization. In paragraph 4, we find the lower bound on the probability of error given our BoP distribution, dataset characteristics, and selected $\epsilon$. We illustrate that for regression, sufficiency has a lower bound exceeding 80%, whereas prediction and incomprehensiveness have a lower bound of less than 10% and therefore show no barriers to testing. The empirical benefit for sufficiency is 0.1914 as opposed to 0.0021 for prediction, so despite being larger in magnitude it indicates nothing about the reliability of evaluation. This addresses a critical, real-world failure mode: a practitioner observes an empirical gain and deploys a personalized model, unaware that a perceived accuracy gain (or loss) from personalization may be the result of statistical fluctuations that depend on dataset size and the number of groups. Our framework provides a necessary guardrail, preventing dangerously misleading conclusions that can arise from data sparsity. In high-stakes applications like healthcare, this could prevent a practitioner from deploying a personalized model based on a seemingly large but statistically meaningless benefit, thereby avoiding potential harm or wasted resources.
>
> * For the **final insight,** we are referring to the additional dataset results in Appendix G, specifically Table 4. We use $\epsilon = 0.002$ and find that no results (spanning classification or regression) will have a lower bound on $P_e$ below 50% with this selected $\epsilon$ (stated in the caption for Table 4 and now added to), therefore rendering the test unreliable for this dataset. Given the limited size of most medical datasets, it may seem nearly impossible to build truly personalized predictive models: there is not enough data to reliably test whether personalization benefits all groups. Hence, our framework leads to a striking conclusion: personalized medicine is likely to fail unless we can gather much larger datasets.
>
> We have made these connections explicit in both paragraphs of Section 6.
>
> **Novelty: Beyond Monteiro Paes et al. (2022).** You suggested our contribution might be limited given prior work by Monteiro Paes et al. While our framework builds upon their proposed approach, our contribution is a significant and necessary extension. The key novelty lies not in tightening their bound, but generalizing the analysis from binary costs to the continuous-valued metrics that are essential for evaluating regression tasks and, crucially, for quantifying explanation quality in regression. This extension is what unlocks the ability to audit the fairness and reliability of explanations for all models, bridging personalization, fairness, and explainability. We also extend the lower bound to accommodate a differing number of samples in each group to allow for more realistic dataset evaluation. In the case of Monteiro Paes, their bound can only be used when each group has the same number of samples, whereas our formulation allows varying sizes.
>
> (2/3)

---

> ### Author Response · Authors · 2025-11-21
> **Response to Reviewer dhvG (3/3)**
>
> **Question about Thm 4.1 and 4.2:**
>
> * **Distribution Construction:** Yes, it is easy to construct distributions that illustrate Theorems 4.1 and 4.2. In both proofs we use minimal synthetic settings that isolate the core effect: personalization can leave prediction unchanged while changing explanation quality.
> For Theorem 4.1, we take two independent uniform features and define the personalized attribute as a coarse summary of them, the indicator of $X_1 + X_2 > 0$. This makes the personalized model’s important feature more concentrated than the generic model’s. Under sufficiency or incomprehensiveness, this difference causes the explanation scores to diverge even though both models have perfect accuracy. For Theorem 4.2, we use a single uniform feature with the personalized attribute equal to that feature. Both models again achieve perfect prediction, but removing the important feature harms the personalized model’s explanation more than the generic model’s. This shows the opposite direction of the effect.
>
>
> * **Examples in practice:** Although these distributions are deliberately minimal to permit closed-form proofs, the significance of these proofs lies not in their technical complexity but in their conceptual contribution: to show that gains in prediction performance (as measured by $\gamma_P$) and gains in explanation quality (as measured by $\gamma_X$) need not align. The same mechanism appears in practice whenever features contain overlapping or redundant information, or when a personalized attribute gives a simpler path to a prediction without changing accuracy. These conditions are common in real datasets, especially in clinical and biological settings with correlated measurements (please refer to detailed Examples 1 and 2 in Section 4). In our case studies on MIMIC-III and two additional datasets we observe this pattern directly, with prediction gains and explanation gains separating across several demographic groups.
>
> (3/3)

---

> > ### Comment · Reviewer_dhvG · 2025-11-22
> >
> > I appreciate the authors for their response. Concerns still remain:
> >
> > **Del Giudice (2024)**
> >
> > I still believe that this reference is ill-placed. I strongly disagree with the authors on the use of this reference:
> > > it reflects a broader belief that better prediction should naturally lead to better explanations, either of real world structure as in Del Giudice or of a model’s internal reasoning in high stakes XAI settings
> >
> > The paper needs a reference to back up their claim pertaining to **model explanations**.
> >
> > ---
> > **Dai et al. 2022**
> >
> > This is not a real-life example; they mention that it is a "stylized" example. The main question with this example is:
> >
> > "Are explanations used this way in practice?"
> >
> > can the authors provide references that suggest that XAI methods are used by doctors to make consequential medical decisions?
> >
> > ---
> > **Metrics**
> > - Even if take this use case for granted, it is hard to contextualise these test results. For example, what does a 0.1914 gain in sufficiency from personalisation mean in this task? Does it actually help the doctor make a better decision?
> > - Note that this is an inherent limitation of these quantitative explainability metrics
> >
> > ---
> > **Examples**
> > - "pre-approved by another bank": have the authors come across a feature like this in credit datasets?
> > - Also, I am confused to why the features added in the examples are features that may be used to make personlised predictions

---

> > > ### Author Response · Authors · 2025-11-27
> > > **Response by Authors (1/2)**
> > >
> > > Thank you for your continued engagement, we address the remaining concerns below:
> > >
> > > **Del Giudice (2024): Practitioner Expectations.**
> > >
> > > We understand your point, and agree that Del Giudice (2024) might not be the ideal reference for motivating the assumption we discuss; consequently, we have removed it in the new revised manuscript. We now ground this motivation directly in the XAI literature, which explicitly describes the persistent tendency to conflate model predictive accuracy with explanation correctness or faithfulness.
> > >
> > > For example, the survey passage we now cite [1] documents that researchers routinely assume that plausible-looking explanations from a high-performing model are faithful, and that evaluating whether an explanation “looks reasonable” in practice amounts to implicitly evaluating the model’s predictive accuracy rather than the explanation itself. In particular, Zhang et al. (2020) emphasize that assessing whether an explanation “looks reasonable” evaluates “the accuracy of the black box model and not the faithfulness of the explanation,” and Adebayo et al. (2018), Jacovi & Goldberg (2020), and Leavitt & Morcos (2020) all critique the widespread assumption that explanations that accompany a high-accuracy model must accurately reflect its internal reasoning. As the survey summarizes, these commentaries “**relate to the inherent coupling of evaluating the black box’s predictive accuracy with explanation quality**,” even though—as Robnik-Šikonja & Bohanec (2018) state—these quantities can be orthogonal.
> > >
> > > This literature reflects precisely the assumption we formalize in our work: that improved predictive accuracy is expected to yield higher-quality, more trustworthy explanations. Our contribution is to show that, under personalization, this expectation can fail even when explanation quality is defined in terms of model-faithful XAI metrics. The revised version of Section 4 now cites this literature directly to support the claim.
> > >
> > > **Dai et al. 2022: Explanations in Practice.**
> > >
> > > We agree that the Dai et al. (2022) example is stylized. However, XAI methods are already widely deployed in the medical domain, and clinicians routinely interact with explanations when interpreting AI outputs. This is not hypothetical: a recent review identified 454 medical AI articles published between 2018–2022, with 93 analyzed in depth, showing extensive use of XAI techniques across diagnostic and clinical decision-support applications [1] [2].
> > >
> > > A growing body of work shows that XAI is already shaping consequential medical decisions across multiple clinical domains. In obstetrics, explainable decision-support systems for gestational diabetes significantly influence clinicians’ choices and advice-taking behavior, demonstrating that explanations directly affect medical judgment [3]. In dermatology, domain-specific XAI explanations increase diagnostic accuracy, confidence, and trust, highlighting clinicians’ willingness to adopt XAI systems in practice [4]. In radiology, physicians achieve their highest diagnostic accuracy when receiving AI advice paired with visual explanatory annotations, with non-experts benefiting most from explainable guidance [5]. In cardiology, XAI methods are used to select and justify heart-failure survival prediction models, with explainability explicitly enabling clinicians to understand model reasoning and make more informed treatment decisions [6]. Together, these studies demonstrate that explanations influence diagnosis, trust, and decision pathways in real clinical environments—underscoring the importance of evaluating whether explanations faithfully reflect model behavior.
> > >
> > > We have included this discussion in the Appendix A of the revised manuscript.
> > >
> > > (1/2)

---

> > > > ### Author Response · Authors · 2025-11-27
> > > > **Response by Authors (2/2)**
> > > >
> > > > **Metrics: Clarifying their Utility**
> > > >
> > > > We agree that metrics like sufficiency can be hard to interpret in absolute terms, and this is a general limitation of quantitative XAI evaluation. These metrics are not designed to measure clinical utility directly. Instead, they assess how well an explanation reflects the model’s actual reasoning before the explanation is used in downstream tasks such as debugging, model comparison, or scientific analysis [8], [9]. In this sense, a 0.19 increase in sufficiency does not imply a specific improvement in a doctor’s decision, but it does indicate that the explanation more accurately captures the features the model relies on. These are not end-user clinical metrics, they are validation tools that help determine whether an explanation is trustworthy enough to be used in subsequent workflows.
> > > >
> > > > **Examples: Sensitive Financial Attributes**
> > > >
> > > > A personalized feature is an attribute specific to an individual—often protected, sensitive, or costly to obtain—used to tailor a prediction or explanation to that person or subgroup. In this sense, “pre-approved by another bank” is an illustrative example of an individualized financial attribute that is strongly correlated with existing predictors.
> > > >
> > > > While this exact feature may not appear verbatim in public datasets, real credit datasets contain closely related individualized, bureau-derived attributes. For example, the widely used FICO HELOC dataset [10] contains personal credit-history indicators such as RiskPerformance, ExternalRiskEstimate, PercentTradesWBalance, and other attributes derived from credit-bureau records of homeowners applying for home-equity lines of credit (HELOCs). These variables encode prior loan performance, credit utilization, and personal financial behavior—precisely the kind of individualized information our example is meant to represent.
> > > >
> > > > We hope our responses successfully contextualized and validated the real-world necessity of our framework, providing the necessary grounds to reconsider the paper’s overall assessment.
> > > >
> > > > **References**
> > > >
> > > > [1] Nauta, Meike, Jan Trienes, Shreyasi Pathak, Elisa Nguyen, Michelle Peters, Yasmin Schmitt, Jörg Schlötterer, Maurice Van Keulen, and Christin Seifert. "From anecdotal evidence to quantitative evaluation methods: A systematic review on evaluating explainable ai." ACM Computing Surveys 55, no. 13s (2023): 1-42.
> > > >
> > > > [2] Ali, Subhan, Filza Akhlaq, Ali Shariq Imran, Zenun Kastrati, Sher Muhammad Daudpota, and Muhammad Moosa. "The enlightening role of explainable artificial intelligence in medical & healthcare domains: A systematic literature review." Computers in Biology and Medicine 166 (2023): 107555.
> > > >
> > > > [3] Salih, Ahmed M., Ilaria Boscolo Galazzo, Polyxeni Gkontra, Elisa Rauseo, Aaron Mark Lee, Karim Lekadir, Petia Radeva, Steffen E. Petersen, and Gloria Menegaz. "A review of evaluation approaches for explainable AI with applications in cardiology." Artificial Intelligence Review 57, no. 9 (2024): 240.
> > > >
> > > > [4] Du, Yuhan, Anna Markella Antoniadi, Catherine McNestry, Fionnuala M. McAuliffe, and Catherine Mooney. "The role of XAI in advice-taking from a clinical decision support system: a comparative user study of feature contribution-based and example-based explanations." Applied Sciences 12, no. 20 (2022): 10323.
> > > >
> > > > [5] Chanda, Tirtha, Katja Hauser, Sarah Hobelsberger, Tabea-Clara Bucher, Carina Nogueira Garcia, Christoph Wies, Harald Kittler et al. "Dermatologist-like explainable AI enhances trust and confidence in diagnosing melanoma." Nature Communications 15, no. 1 (2024): 524.
> > > >
> > > > [6] Gaube, Susanne, Harini Suresh, Martina Raue, Eva Lermer, Timo K. Koch, Matthias FC Hudecek, Alun D. Ackery et al. "Non-task expert physicians benefit from correct explainable AI advice when reviewing X-rays." Scientific reports 13, no. 1 (2023): 1383.
> > > >
> > > > [7] Moreno-Sánchez, Pedro A. "Improvement of a prediction model for heart failure survival through explainable artificial intelligence." Frontiers in cardiovascular medicine 10 (2023): 1219586.
> > > >
> > > > [8] Liu, Yang, Sujay Khandagale, Colin White, and Willie Neiswanger. "Synthetic Benchmarks for Scientific Research in Explainable Machine Learning.” Thirty-fifth Conference on Neural Information Processing Systems Datasets and Benchmarks Track.
> > > >
> > > > [9] Zhou, Pei-Yuan, Amane Takeuchi, Fernando Martinez-Lopez, Malikeh Ehghaghi, Andrew KC Wong, and En-Shiun Annie Lee. "Benchmarking Interpretability in Healthcare Using Pattern Discovery and Disentanglement." Bioengineering 12, no. 3 (2025): 308.
> > > >
> > > > [10] Brown, Kyle, Derek Doran, Ryan Kramer, and Brad Reynolds. "Heloc applicant risk performance evaluation by topological hierarchical decomposition." arXiv preprint arXiv:1811.10658 (2018).
> > > >
> > > > (2/2)

---

### Official Review · Reviewer_bsq1 · 2025-11-06

**Soundness:** 3
**Presentation:** 3
**Contribution:** 3
**Rating:** 6
**Confidence:** 2

**Summary:**

The paper discusses personalization and explainability of machine learning models and prove the relationship between these two. The paper then discusses hypothesis testing, lower bound, practical aspects of the theory and concludes with experiments.

**Strengths:**

I am not really an expert in this field so just offering my best evaluation here.

The paper is well written and provides important insights on the personalization and explainability. The sections on their relationship are well-thought and the theoretical aspects are sound.

**Weaknesses:**

I am not familiar with the relevant literature but the paper is quite original and contains insightful theories.

**Questions:**

None

---

> ### Author Response · Authors · 2025-11-21
> **Response to Reviewer bsq1**
>
> Thank you for your complementary and honest review, we appreciate your positive feedback. Please let us know if there is anything we can do to help clarify the work further.

---

### Author Response · Authors · 2025-11-21
**General Response to All Reviewers**

We are genuinely grateful for the quality and depth of feedback provided by all reviewers. Their constructive criticism and insightful suggestions have improved our paper's clarity, rigor, and impact. Based on the common concerns raised, we have made revisions to the manuscript (highlighted in blue for easy identification) that we hope strengthen our contribution.

We welcome continued discussion and remain committed to further revisions that strengthen our contribution to the field. Detailed responses to individual reviewer comments follow below.

---

### Author Response · Authors · 2025-12-03
**General Response to AC (1/2)**

We summarize our rebuttal to facilitate the new Area Chair’s evaluation. Below we provide a list of the primary reviewer concerns and a concise summary of our response to each. Readers seeking full details or responses to minor points should refer to our complete comments to the reviewers.

**Reviewer bsq1**
The reviewer offered a broadly positive assessment, describing the paper as well written, sound, and insightful, but explicitly noted that they are “not really an expert in this field,” did not engage with the technical details, and raised no concrete objections. We thanked them and offered to clarify any parts of the work; there were no specific issues to resolve.

**Reviewer dhvG**
We enumerate as follows their main questions and concerns:


* _Whether the divergence between prediction and explanation benefits occurs in practice, and why it matters_ $\rightarrow$ We pointed to Tables 2, 4, and 5 and additional real datasets where prediction and explanation benefits diverge across demographic groups, and clarified why lower explanation fidelity has real consequences (drawing on Dai et al. 2022 and related work).
* _The use of Del Giudice (2024) as a motivating citation_ $\rightarrow$ We removed Del Giudice and instead grounded the motivation in XAI surveys that explicitly document the common tendency to conflate model accuracy with explanation quality.
* _How to interpret our quantitative XAI metrics (e.g., sufficiency) and ε-thresholds_ $\rightarrow$ We added text explaining how ε corresponds to practically meaningful accuracy gains and clarified that sufficiency/incomprehensiveness are model-faithfulness metrics rather than direct clinical outcomes.
* _Whether there is evidence that XAI is actually used in consequential medical decisions_ $\rightarrow$ We added extensive references and discussion showing that XAI is already influencing real clinical decisions across multiple specialties.
* _How our results extend prior work by Monteiro Paes et al._ $\rightarrow$ We further reiterated our novelty beyond Monteiro Paes (in particular, our work extends it from binary to continuous metrics, supporting explanation quality and accuracy in regression and handling unequal group sizes).

Unfortunately, many of these points reflect a broader skepticism about XAI evaluation metrics and the extent to which XAI is deployed in practice, which are important field-level debates but largely orthogonal to the technical questions our framework addresses.

**Reviewer BgiC**
This reviewer found the topic interesting and the message “very interesting in a personalization context,” but raised higher-level concerns:
* _Confusion about whether the focus is fairness vs. prediction accuracy_ $\rightarrow$ We clarified that our goal is to test whether personalization improves prediction and explanation for all groups and to bound the probability of error for that fairness-style criterion, revising the abstract and introduction accordingly.
* _Difficulty locating the proof of Theorem 5.1_ $\rightarrow$ We clarified it did appear in Appendix as Theorem D.3 (both statement and proof), but fixed the labelling for enhanced identifiability.
* _Skepticism that some of our phenomena are specific to personalization rather than generic feature collinearity_ $\rightarrow$ We explained why personalization is not just generic collinearity, but introduces new model pathways that can change explanation allocations even when predictions for all groups stay the same.
* _Value of our framework relative to simply comparing fairness metrics with and without personalization_ $\rightarrow$ We emphasized that our framework augments the simple “with vs. without personalization” comparison by showing when any such empirical comparison is statistically unreliable due to group sizes, number of attributes, and BoP distribution.

We note that some aspects of the review suggest a misunderstanding of the paper’s main contribution. In particular, the question “why not simply compare fairness metrics with and without personalization” overlooks that our framework provides a lower bound on the probability of error for exactly that comparison, and the continued focus on Section 4 (even after we clarified the location of the proof of Theorem 5.1) reflects limited engagement with the statistical testing framework and empirical analysis in Sections 5–6.

(1/2)

---

> ### Author Response · Authors · 2025-12-03
> **General Response to AC (2/2)**
>
> **Reviewer k2Uy**
> This reviewer provided a strongly positive evaluation, highlighting our extension of BoP to continuous and explanation metrics, the conceptual link between personalization, fairness, and explainability, and the practical finite-sample bounds. Their concerns were primarily about clarity and scope:
> * _Minor clarifications and revisions_ $\rightarrow$ _Minor clarifications and revisions_ $\rightarrow$ We modified the text to name the explanation methods earlier in the paper, streamlined notation and definitions introduced in Tables 1 and 3, and provided more intuition in the main text and clearer wording around Theorem 4.3 (“can” vs “always”), and expand related works to include fairness of recourse.
> * _Applicability beyond feature-attribution explainers_ $\rightarrow$ We emphasized that Section 5’s testing framework is explainer-agnostic.
> * _Generalization of Thm. 4.3 beyond additive models_ $\rightarrow$ We revised Thm. 4.3’s statement and added a corollary that spells out the covariance conditions under which alignment holds without independence.
> * _Robustness of our bounds to misspecified BoP distributions_ $\rightarrow$ We analyzed robustness to Laplace vs. Gaussian BoP distributions (including a figure) and noted that Theorem 5.1 applies to any distribution with a known pdf.
> * _Testing absence of harm_ $\rightarrow$ We explained how the framework can test both absence of harm and presence of benefit by choosing ε appropriately.
>
> Reviewer k2Uy offered a strongly positive assessment, with suggestions that focused on clarity, scope, and communicating applicability rather than on changing the core framework or results.
>
> **Our conclusion**:
> Overall, the revised manuscript substantively addresses the reviewers’ scientific concerns: we clarify the real-world relevance and empirical occurrence of divergent prediction vs. explanation benefits, ground the motivation in the XAI literature, strengthen connections to fairness and recourse, and make the novelty and generality of our statistical framework more explicit. The remaining reservations do not point to flaws in the framework or results, but instead reflect skepticism about XAI and limited engagement with the broader theoretical and statistical contributions in Sections 4–5. The paper combines the first formal analysis of how personalization can differentially affect prediction and explanation, and a general hypothesis-testing framework with finite-sample bounds that tells practitioners when empirical personalization gains (for prediction or explanation) are statistically untrustworthy across groups.
>
> If there are any remaining questions or specific points where our revisions are unclear, we would be very grateful for guidance and are happy to provide any further clarification the Area Chair might find helpful.
>
> (2/2)

---

### Meta-Review · Area_Chair_Yt5Z · 2026-01-10

**Summary:**

The reviewers acknowledge that the topic of the paper is interesting and novel, with little close related work. However, they also raised questions mainly about its contributions compared to Monteiro Paes et al. (2022), and its practical limitations. I believe that the clarification with regards to Monteiro Paes et al. (2022) provide the necessary details and actually show that this paper is moving one step forward to a more practical, yet theoretically grounded approach. On the second line, while I agree with reviewer BgiC on several of the concerns about the practical limitations, I also agree with the authors that most of these points are limitations of the field itself and not directly a limitation of the current paper. While the most negative reviewer, dhvG, did not engage much with the discussion, I believe their main concerns about the novelty of the work, its motivation and practical limitations have been mostly addressed.

In summary, I believe that the authors have done a very good job during the rebuttal and, while I believe they should further revise the paper to incorporate the main points of discussion with the reviewers,  the contributions of the paper are interesting and its quality s above the bar,

**Reviewer Concerns:**

As I comment above, I believe that those points relevant to the main contributions of the paper have been addressed and the remaining points are wither minor or not directly a limitation of the present paper. Reviewer BgiC moved from a "3.6 to a 4.4" score, which I read as a weak reject based on their comments, and  reviewer dhvG did not state anything about their score but the remaining concerns  seem related to XAI evaluation metrics and practical limitations of the XAI field, being thus in my opinion orthogonal to the contributions of the paper.

**Reviewer Scores:**

I do not fully understand why the most negative reviewers  have been inflexible on their assessments despite of the authors efforts to address, in my opinion, satisfactorily their concerns.

---

### Decision · Program_Chairs · 2026-01-26

Accept (Poster)